# Provable Benefit of Multitask Representation Learning in Reinforcement Learning

**Yuan Cheng**
University of Science and Technology of China
`cy16@mail.ustc.edu.cn`

**Songtao Feng**
The Ohio State University
`feng.1359@osu.edu`

**Jing Yang**
The Pennsylvania State University
`yangjing@psu.edu`

**Hong Zhang**
University of Science and Technology of China
`zhangh@ustc.edu.cn`

**Yingbin Liang**
The Ohio State University
`liang.889@osu.edu`

## Abstract

As representation learning becomes a powerful technique to reduce sample complexity in reinforcement learning (RL) in practice, theoretical understanding of its advantage is still limited. In this paper, we theoretically characterize the benefit of representation learning under the low-rank Markov decision process (MDP) model. We first study multitask low-rank RL (as upstream training), where all tasks share a common representation, and propose a new multitask reward-free algorithm called REFUEL. REFUEL learns both the transition kernel and the near-optimal policy for each task, and outputs a well-learned representation for downstream tasks. Our result demonstrates that multitask representation learning is provably more sample-efficient than learning each task individually, as long as the total number of tasks is above a certain threshold. We then study the downstream RL in both online and offline settings, where the agent is assigned with a new task sharing the same representation as the upstream tasks. For both online and offline settings, we develop a sample-efficient algorithm, and show that it finds a near-optimal policy with the suboptimality gap bounded by the sum of the estimation error of the learned representation in upstream and a vanishing term as the number of downstream samples becomes large. Our downstream results of online and offline RL further capture the benefit of employing the learned representation from upstream as opposed to learning the representation of the low-rank model directly. To the best of our knowledge, this is the first theoretical study that characterizes the benefit of representation learning in exploration-based reward-free multitask RL for both upstream and downstream tasks.

## 1  Introduction

Multitask representation learning has arisen as a popular and powerful framework to deal with data-scarce environment in supervised learning, bandits and reinforcement learning (RL) recently (Maurer et al., 2016; Du et al., 2020; Tripuraneni et al., 2021; Yang et al., 2021; Cella et al., 2022; Lu et al., 2021; Arora et al., 2020; Hu et al., 2021). The general multitask representation learning framework consists of both upstream multiple source tasks learning and downstream target task learning. In RL applications, in upstream, the agent is assigned with $T$ source tasks sharing a

common representation and the goal is to learn a near-optimal policy for each task via learning the representation. In downstream, with the help of the learned representation, the agent aims to find a near-optimal policy of a new task that shares the same representation as the source tasks. While representation learning has achieved great success in supervised learning (Du et al., 2020; Tripuraneni et al., 2021; Maurer et al., 2016; Kong et al., 2020) and multi-armed bandits (MAB) problems (Yang et al., 2021; Qin et al., 2022; Cella et al., 2022), most works in multitask RL mainly focus on empirical algorithms (Sodhani et al., 2021; Arulkumaran et al., 2022; Teh et al., 2017) with limited theoretical works (Arora et al., 2020; Hu et al., 2021; Brunskill and Li, 2013; Müller and Pacchiano, 2022; Calandriello et al., 2015; Lu et al., 2021; D'Eramo et al., 2020). Generally speaking, there are two main challenges for multitask representation learning in RL. First, differently from supervised learning with static datasets, in MAB and RL, representation learning and data collection are intertwined with each other. The agent has to choose an exploration policy in each iteration based on the representation learned in the past. Such iterative process introduces temporal dependence to the collected data. Note that none of the aforementioned theoretical multitask RL studies took the influence of sequential exploration and temporal dependence in data into consideration. Second, data distribution mismatch between the target task and the source tasks may occur. Additional assumption is usually required in order to ensure the representation learned from sources task will benefit the target task (Lu et al., 2021).

In this work, we aim to unveil the fundamental impact of multitask representation learning on RL by answering the following two open questions:

- First, in the upstream, is it possible to design exploration-based multitask representation learning algorithms for RL to achieve efficiency gain compared with single-task RL after taking the interactive data collection process and temporal data dependency into account?

- Second, can the learned representation from upstream bring efficiency gain to the downstream learning?

**Main contributions:** We focus on multitask representation learning under low-rank MDPs (Agarwal et al., 2020), where the transition kernel admits a low-rank decomposition that naturally involves representation learning. It answers the aforementioned questions affirmatively, as summarized below.

- First, we propose a new multitask representation learning algorithm called REFUEL under low-rank MDPs in upstream. REFUEL features joint modeling learning, pseudo cumulative value function construction for exploration, joint exploration policy learning and a unique termination criterion. These novel designs enable REFUEL to leverage the shared representation across source tasks to improve the learning efficiency.

- We show that REFUEL can 1) find a near-optimal policy and 2) learn a near-accurate model for each of $T$ source tasks to certain average accuracy $\epsilon_u$, with a sample complexity of $\widetilde{O}\left(\frac{H^3 d^2 K^2}{T\epsilon_u^2} + \frac{(H^3 d^4 K + H^5 dK^2 + H^5 d^3 K)}{\epsilon_u^2} + \frac{H^5 K^3}{dT\epsilon_u^2}\right)$, where $H, K, d$ denote the horizon steps, the cardinality of action space and the low-rank dimension, respectively. Compared with state-of-the-art single-task representation in low-rank MDPs (Uehara et al., 2022), REFUEL achieves reduced sample complexity as long as $T = \Omega\left(\frac{K}{d^5}\right)$, which shows the benefit of multitask learning provably. To the best of our knowledge, this is the first theoretical result demonstrating the benefit of representation learning in exploration-based reward-free multitask RL.

- Finally, we apply the learned representation from REFUEL to downstream online and offline learning problems. In **offline** RL, we propose a new sample efficient algorithm with the suboptimality gap bounded by $\widetilde{O}(H\sqrt{d}\xi_{\text{down}} + H\max\{d, \sqrt{p}\}/\sqrt{N_{\text{off}}})$, where $N_{\text{off}}$ is the number of samples of offline dataset and $p$ measures the model complexity of reward. In **online** RL, we propose an online algorithm with the suboptimality gap bounded by $\widetilde{O}\left(Hd\xi_{\text{down}} + Hd^{1/2}N_{on}^{-1/2}\max\{d, \sqrt{p}\}\right)$, where $N_{\text{on}}$ is the number of iterations in the online algorithm. In both the offline and online settings, the first term captures the **approximation error** where $\xi_{\text{down}}$ measures the quality of the learned representation from upstream, and is negligible when the upstream learning is sufficiently accurate, and the second term improves the dependency on $H$ and $d$ compared with the state of the art (Uehara et al., 2022) for low-rank MDPs without the knowledge of the representation, showing the benefit of employing the learned representation from upstream.

**Notations:** Throughout the paper, we use $[N]$ to denote set $\{1, \ldots, N\}$ for $N \in \mathbb{N}$, use $\|x\|_2$ to denote the $\ell_2$ norm of vector $x$, use $\|X\|_{\mathrm{F}}$ to denote the Frobenius norm of matrix $X$, use $\triangle(\mathcal{A})$ to denote the probability simplex over set $\mathcal{A}$, and use $\mathcal{U}(\mathcal{A})$ to denote uniform sampling over $\mathcal{A}$, given $|\mathcal{A}| < \infty$. Furthermore, for any symmetric positive definite matrix $\Sigma$, we let $\|x\|_{\Sigma} := \sqrt{x^\top \Sigma x}$.

## 2 Related Work

**Multitask bandits, multitask RL and meta-RL:** The benefit of multitask learning in linear bandits has been investigated in Yang et al. (2021); Qin et al. (2022); Cella et al. (2022); Azizi et al. (2022); Deshmukh et al. (2017); Cella and Pontil (2021); Hu et al. (2021). For multitask RL, Arora et al. (2020) showed that representation learning can reduce sample complexity for imitation learning. Calandriello et al. (2015) studied multitask learning under linear MDPs with known representation. Hu et al. (2021) studied multi-task RL with low inherent Bellman error (Zanette et al., 2020) with known representation. Assuming all tasks are sampled from an unknown distribution, Brunskill and Li (2013) studied the benefit of multitask RL when each task is sampled independently from a distribution over a finite set of MDPs while Müller and Pacchiano (2022) focused on meta-RL for linear mixture model (Yang and Wang, 2020). Considering all tasks share a common representation, D'Eramo et al. (2020) showed the benefit in the convergence rate on value iteration and Lu et al. (2021) proved the sample efficiency gain of multitask RL under low-rank MDPs. None of the above works took the influence of sequential exploration and temporal dependence in data into consideration, where Lu et al. (2021) assumed a generative model is accessible and D'Eramo et al. (2020) assumed that the dataset is given in prior. In our study, we design reward-free exploration for multitask RL.

**Single task RL under low-rank MDPs:** Low-rank MDPs are first studied byAgarwal et al. (2020), and Uehara et al. (2022) studied low-rank MDPs for both online and offline RL. Modi et al. (2021) proposed a model-free algorithm MOFFLE under low-nonnegative-rank MDPs. Du et al. (2019); Misra et al. (2020); Zhang et al. (2022) studied *block MDP* which is a special case of low-rank MDPs.

**Offline RL under linear and low-rank MDPs:** Most of existing works on offline RL focus on single-task learning, among which Jin et al. (2021) studied offline RL under linear MDPs and proposed an algorithm with provable suboptimality gap, Uehara et al. (2022) studied offline RL under low-rank MDPs and proposed an algorithm REP-LCB that leverages pessimism, and Yin et al. (2022); Xiong et al. (2022) studied offline RL leveraging variance information.

## 3 Problem formulation

### 3.1 Episodic MDP

Consider an episodic non-stationary Markov Decision Process (MDP) with finite horizon $H$, denoted by $\mathcal{M} = (\mathcal{S}, \mathcal{A}, H, P, r)$, where $\mathcal{S}$ is the state space (possibly infinite), $\mathcal{A}$ is the finite action space with cardinality $K$, $H$ is the number of steps in each episode, $P = \{P_h\}_{h \in [H]}$ is the collection of transition measures with $P_h : \mathcal{S} \times \mathcal{A} \mapsto \triangle(\mathcal{S})$, $r = \{r_h\}_{h \in [H]}$ is the collection of deterministic reward functions with $r_h : \mathcal{S} \times \mathcal{A} \mapsto [0, 1]$. The initial state $s_1$ is assumed to be fixed for each episode and the sum of the reward is normalized with $\sum_{h=1}^{H} r_h \in [0, 1]$.

Denote policy $\pi = \{\pi_h\}_{h \in [H]}$ as a sequence of mappings where $\pi_h : \mathcal{S} \mapsto \triangle(\mathcal{A})$. We use $\pi_h(a|s)$ to denote the probability of selecting action $a$ in state $s$ at step $h$. Given a starting state $s_h$, we use $s_{h'} \sim (P, \pi)$ to denote a state sampled by executing policy $\pi$ under the transition model $P$ for $h' - h$ steps and $\mathbb{E}_{(s_h, a_h) \sim (P, \pi)}[\cdot]$ to denote the expectation over states $s_h \sim (P, \pi)$ and actions $a_h \sim \pi$. Given policy $\pi$ and a state $s$ at step $h$, the (state) value function is defined as $V_{h,P,r}^\pi(s) := \mathbb{E}_{(s_{h'}, a_{h'}) \sim (P, \pi)}\left[\sum_{h'=h}^{H} r_{h'}(s_{h'}, a_{h'})\Big| s_h = s\right]$. Similarly, the (state-)action value function for a given state-action pair $(s, a)$ under policy $\pi$ at step $h$ is defined as $Q_{h,P,r}^\pi(s, a) := r_h(s, a) + (P_h V_{h+1,P,r}^\pi)(s, a)$, where we define $(P_h f)(s, a) := \mathbb{E}_{s' \sim P_h(\cdot|s,a)}[f(s')]$ for any function $f : \mathcal{S} \mapsto \mathbb{R}$. For simplicity, we omit subscript $h$ for $h = 1$. Since the MDP begins with the same initial state $s_1$, to simplify the notation, we use $V_{P,r}^\pi$ to denote $V_{1,P,r}^\pi(s_1)$ without any ambiguity. Let $P^*$ denote the transition kernel of the underlying environment. Given a reward function $r$, there always exists an optimal policy $\pi^*$ that yields the optimal value function $V_{P^*,r}^{\pi^*} = \sup_\pi V_{P^*,r}^\pi$, abbreviated as $V_{P^*,r}^*$.

This paper focuses on low-rank MDPs (Jiang et al., 2017; Agarwal et al., 2020) defined as follows.

**Definition 1** (Low-rank MDPs). A transition kernel $P_h^* : \mathcal{S} \times \mathcal{A} \to \triangle(\mathcal{A})$ admits a low-rank decomposition with dimension $d \in \mathbb{N}$ if there exists two embedding functions $\phi_h^* : \mathcal{S} \times \mathcal{A} \to \mathbb{R}^d$ and $\mu_h^* : \mathcal{S} \to \mathbb{R}^d$ such that

$$P_h^*(s'|s,a) = \langle \phi_h^*(s,a), \mu_h^*(s') \rangle, \quad \forall s, s' \in \mathcal{S}, a \in \mathcal{A}.$$

Without loss of generality, we assume $\|\phi_h^*(s,a)\|_2 \le 1$ for all $(s,a) \in \mathcal{S} \times \mathcal{A}$ and for any function $g : \mathcal{S} \mapsto [0,1]$, $\left\| \int \mu_h^*(s)g(s)ds \right\|_2 \le \sqrt{d}$. An MDP is a low-rank MDP with dimension $d$ if for $h \in [H]$, its transition kernel $P_h^*$ admits a low rank decomposition with dimension $d$. Let $\phi^* = \{\phi_h^*\}_{h \in [H]}$ and $\mu^* = \{\mu_h^*\}_{h \in [H]}$ be the embeddings for $P^*$, where $\phi^*$ is also called representation in RL literature.

### 3.2 Reward-free multitask RL and its downstream learning

We consider a multitask RL consisting of reward-free upstream and reward-known downstream.

In **reward-free multitask** upstream learning, the agent is expected to conduct **reward-free exploration** over $T$ source tasks, where all reward functions $\{r^t\}_{t \in [T]}$ are unknown. Each task $t \in [T]$ is associated with a low-rank MDP $\mathcal{M}^t = (\mathcal{S}, \mathcal{A}, H, P^t, r^t)$. All $T$ tasks are identical except for (a) their transition model $P^t$, which admits a low-rank decomposition with dimension $d$: $P^t(s'|s,a) = \langle \phi^*(s'), \mu^{(*,t)}(s,a) \rangle$ for all $t \in [T]$ and (b) their reward $r^t$. Namely, **all tasks share the same feature function** $\phi^*$, but may differ in $\mu^{(*,t)}$ as well as reward $r^t$. The goal of upstream learning is to find a near-optimal policy and a near-accurate model for any task $t \in [T]$ and any reward function $r^t, t \in [T]$ via sufficient reward-free exploration, and output a well-learned representation for downstream task. For each $t \in [T]$, we use $P^{(*,t)}$ to denote the true transition kernel of task $t$.

In downstream learning, the agent is assigned with a new target task $T+1$ associated with a low-rank MDP $\mathcal{M}^{T+1} = (\mathcal{S}, \mathcal{A}, H, P^{T+1}, r^{T+1})$. The task shares the same $\mathcal{S}, \mathcal{A}, H$ and $\phi^*$ with the $T$ upstream tasks, but has a task-specific $\mu^{(*,T+1)}$. We also assume that the reward for the new task $r^{T+1}$ is given to the agent. Here, the agent is assumed to take the learned representation function $\widehat{\phi}$ during the upstream training, and then interacts with the new task environment $\mathcal{M}^{T+1}$ in an offline or online manner to find a near-optimal policy for the new task. By utilizing the representation learned in the upstream, the agent is expected to achieve better sample efficiency in the downstream. Finally, we use $P^{(*,T+1)}$ to denote the true transition kernel of task $T+1$.

## 4 Upstream reward-free multitask RL

In this section, we present our proposed algorithm for upstream reward-free multitask RL in low-rank MDPs and characterize its theoretical performance.

Since it is impossible to learn a model in polynomial time if there is no assumption on features $\phi_h$ and $\mu_h$, we first adopt the following conventional assumption (Agarwal et al., 2020).

**Assumption 1.** (Realizability). A learning agent can access to a model class $\{(\Phi, \Psi)\}$ that contains the true model, namely, for any $h \in [H], t \in [T]$, the embeddings $\phi_h^* \in \Phi, \mu_h^{(*,t)} \in \Psi$.

While we assume the cardinality of function classes to be finite for simplicity, extensions to infinite classes with bounded statistical complexity are not difficult (Sun et al., 2019; Agarwal et al., 2020).

### 4.1 Algorithm design

We first describe our proposed algorithm REFUEL depicted in Algorithm 1.

In each iteration $n$, for each task $t$ and time step $h$, the agent executes the exploration policy $\pi_t^{n-1}$ followed by two uniformly chosen actions to collect trajectories for $H$ episodes. Note that $\pi_t^{n-1}$ is designed at the previous iteration. Then, to utilize the common feature of data collected from different source tasks, the agent processes the data jointly as following steps.

**Joint MLE-based Modeling Learning:** The agent passes all previously collected data to estimate the low-rank components $\widehat{\phi}_h^{(n)}, \widehat{\mu}_h^{(n,1)}, \ldots, \widehat{\mu}_h^{(n,t)}$ simultaneously via the MLE oracle $MLE\left( \bigcup_{t \in [T]} \mathcal{D}_h^{(n,t)} \right)$ (line 8) on the joint distribution defined as follows:

$$\left( \widehat{\phi}_h^{(n)}, \widehat{\mu}_h^{(n,1)}, \ldots, \widehat{\mu}_h^{(n,T)} \right) = \arg\max_{\phi_h \in \Phi, \mu_h^1, \ldots, \mu_h^T \in \Psi} \sum_{n=1}^N \log\left( \prod_{t=1}^T \langle \phi_h(s_h^{(n,t,h)}, a_h^{(n,t,h)}), \mu_h^t(s_{h+1}^{(n,t,h)}) \rangle \right). \quad (1)$$

---
**Algorithm 1 REFUEL** (**RE**ward **F**ree m**U**ltitask r**E**presentation **L**earning)
---
1: **Input:** Regularizer $\lambda_n$, parameter $\widetilde{\alpha}_n, \zeta_n, B$, $\delta, \epsilon_u$, Models $\{(\mu, \phi) : \mu \in \Psi, \phi \in \Phi\}$.
2: Initialize $\pi_0(\cdot|s)$ to be uniform, set $\mathcal{D}_h^{(0,t)} = \emptyset$ for $h \in [H], t \in [T]$.
3: **for** $n = 1, \ldots, N_u$ **do**
4:    **for** $h = 1, \ldots, H$ **do**
5:       **for** $t = 1, \ldots, T$ **do**
6:          Under MDP $\mathcal{M}^t$, use $\pi_t^{n-1}$ to roll into $s_{h-1}^{(n,t,h)}$, uniformly choose $a_{h-1}^{(n,t,h)}, a_h^{(n,t,h)}$ and enter into $s_h^{(n,t,h)}, s_{h+1}^{(n,t,h)}$ and collect data $s_1^{(n,t,h)}, a_1^{(n,t,h)}, \ldots, s_h^{(n,t,h)}, a_h^{(n,t,h)}, s_{h+1}^{(n,t,h)}$.
7:          Add the triple $(s_h^{(n,t,h)}, a_h^{(n,t,h)}, s_{h+1}^{(n,t,h)})$ to the dataset $\mathcal{D}_h^{(n,t)} = \mathcal{D}_h^{(n-1,t)} \cup \{(s_h^{(n,t,h)}, a_h^{(n,t,h)}, s_{h+1}^{(n,t,h)})\}$.
8:       Learn $\left(\widehat{\phi}_h^{(n)}, \widehat{\mu}_h^{(n,1)}, \ldots, \widehat{\mu}_h^{(n,T)}\right)$ via $MLE\left(\bigcup_{t \in [T]} \mathcal{D}_h^{(n,t)}\right)$ as Equation (1).
9:       **for** $t = 1, \ldots, T$ **do**
10:          Update transition kernels $\widehat{P}_h^{(n,t)}(\cdot|\cdot, \cdot) = \langle \widehat{\phi}_h^{(n)}(\cdot, \cdot), \widehat{\mu}_h^{(n,t)}(\cdot) \rangle$, empirical covariance matrix $\widehat{U}_h^{(n,t)}$ as Equation (2) and exploration-driven reward $\widehat{b}^{(n,t)}(\cdot, \cdot)$ as Equation (3).
11:    Set **Pseudo Cumulative Value Function** $PCV\left(\widehat{P}^{(n,t)}, \widehat{b}_h^{(n,t)}, \pi_t; T\right)$ as Equation (4).
12:    Get exploration policy $\pi_1^n, \ldots, \pi_T^n = \arg\max_{\pi_1, \ldots, \pi_T} PCV\left(\widehat{P}^{(n,t)}, \widehat{b}_h^{(n,t)}, \pi_t; T\right)$.
13:    **if** $2PCV\left(\widehat{P}^{(n,t)}, \widehat{b}_h^{(n,t)}, \pi_t; T\right) + 2\sqrt{KT\zeta_n} \leq T\epsilon_u$ **then**
14:       **Terminate** and set $n_u = n$ and **output** $\widehat{\phi} = \widehat{\phi}^{(n_u)}, \widehat{P}^{(1)} = \widehat{P}^{(n_u,1)}, \ldots, \widehat{P}^{(T)} = \widehat{P}^{(n_u,T)}$.
15: **Output:** $\widehat{\phi} = \widehat{\phi}^{(n_u)}, \widehat{P}^{(1)} = \widehat{P}^{(n_u,1)}, \ldots, \widehat{P}^{(T)} = \widehat{P}^{(n_u,T)}$.
---

The MLE oracle above generalizes the MLE oracle commonly adopted for single-task RL in the literature (Agarwal et al., 2020; Uehara et al., 2022). In practice, MLE oracle can be efficiently implemented if the model classes $\Phi, \Psi$ are properly parameterized such as by neural networks.

Next, for each task $t$, the agent uses the learned embeddings $\widehat{\phi}_h^{(n)}, \widehat{\mu}_h^{(n,t)}$ to update the estimated transition kernel $\widehat{P}^{(n,t)}$ at each step $h$ as $\widehat{P}_h^{(n,t)}(s'|s, a) = \langle \widehat{\phi}_h^{(n)}(s, a), \widehat{\mu}_h^{(n,t)}(s') \rangle$.

**Pseudo Cumulative Value Function Construction for Exploration:** The agent first uses the representation estimator $\widehat{\phi}_h^{(n)}$ to update the empirical covariance matrix $\widehat{U}_h^{(n,t)}$ as

$$\widehat{U}_h^{(n,t)} = \sum_{\tau=1}^n \widehat{\phi}_h^{(n)}(s_h^{(\tau,t,h+1)}, a_h^{(\tau,t,h+1)})(\widehat{\phi}_h^{(n)}(s_h^{(\tau,t,h+1)}, a_h^{(\tau,t,h+1)}))^\top + \lambda_n I, \quad (2)$$

where $\{s_h^{(\tau,t,h+1)}, a_h^{(\tau,t,h+1)}, s_{h+1}^{(\tau,t,h+1)}\}$ is the tuple collected at the $\tau$-th iteration, $(h + 1)$-th episode, and step $h$. Then, the agent uses both $\widehat{\phi}_h^{(n)}$ and $\widehat{U}_h^{(n,t)}$ to provide an exploration-driven reward as

$$\widehat{b}_h^{(n,t)}(s_h, a_h) = \min\left\{\widetilde{\alpha}_n \left\|\widehat{\phi}_h^{(n)}(s_h, a_h)\right\|_{(\widehat{U}_h^{(n,t)})^{-1}}, B\right\}, \quad (3)$$

where $B$ and $\widetilde{\alpha}_n$ are pre-determined parameters. To integrate information of learned models and exploration-driven rewards for $T$ tasks, we define a **Pseudo Cumulative Value Function** (PCV) as

$$PCV\left(\widehat{P}^{(n,t)}, \widehat{b}_h^{(n,t)}, \pi_t; T\right) = \sum_{h=1}^{H-1} \sqrt{\sum_{t=1}^T \left\{\mathbb{E}_{\substack{s_h \sim (\widehat{P}^{(n,t)}, \pi_t) \\ a_h \sim \pi_t}} \left[\widehat{b}_h^{(n,t)}(s_h, a_h)\right]\right\}^2}. \quad (4)$$

**Joint Policy Learning for Exploration:** The agent then learns $T$ policies via optimization over the PCV, and uses these policies as the exploration policy for each task in the next iteration. Since PCV measures the cumulative model estimation error over all tasks, these policies will explore the state-action space where the model estimation has large uncertainty on average of $T$ tasks. Here we adopt an optimization oracle for line 12 in Algorithm 1.

**Termination:** REFUEL is equipped with a termination condition and outputs the current estimated model and representation during iterations if the PCV plus certain minor term is below a threshold, which suffices to guarantee that for each task $t$, the returned model $\widehat{P}^{(t)}$ from REFUEL reaches an average proximity to the true model $P^{(*,t)}$.

The joint learning and decision making among all tasks involved in the four main components differentiate REFUEL from single-task RL algorithms, and enable it to exploit the shared representation across the source tasks to improve the sample efficiency.

## 4.2 Theoretical results on sample complexity

In this section, we first establish the sample complexity upper bound for REFUEL and then compare our multitask result with that of single-task RL.

**Theorem 1.** *For any fixed $\delta \in (0, |\Psi|^{\min\{T, -\frac{K}{d^2}\}})$, set $\lambda_n = O(d\log(|\Phi|nTH/\delta))$, $\zeta_n = O(\log(|\Phi||\Psi|^T nH/\delta)/n)$, $\widetilde{\alpha}_n = O(\sqrt{K\log(|\Phi||\Psi|^T nH/\delta) + \lambda_n dT})$ and $B = O(\sqrt{T + K/d^2})$. Let $\widehat{\phi}, \widehat{P}^{(1)}, \ldots, \widehat{P}^{(T)}$ be the output of Algorithm 1. Then, under Assumption 1, with probability at least $1 - \delta$, for any policy $\pi_t$ of task $t \in [T]$ and any step $h \in [H]$, we have*

$$\frac{1}{T}\sum_{t=1}^{T}\mathbb{E}_{s_h\sim(P^{(*,t)},\pi_t),a_h\sim\pi_t}\left[\left\|\widehat{P}_h^{(t)}(\cdot|s_h,a_h) - P_h^{(*,t)}(\cdot|s_h,a_h)\right\|_{TV}\right] \le \epsilon_u. \tag{5}$$

*Further, for any task $t$, given any reward $r^t$, let $\widehat{\pi}_t = \arg\max_\pi V_{\widehat{P}^{(t)},r^t}^\pi$. Then with probability at least $1 - \delta$, $\frac{1}{T}\sum_{t=1}^{T}[V_{P^{(*,t)},r^t}^* - V_{P^{(*,t)},r^t}^{\widehat{\pi}_t}] \le \epsilon_u$.*

*Meanwhile, the number of trajectories collected by each task is at most*

$$\widetilde{O}\left(\frac{H^3d^2K^2}{T\epsilon_u^2} + \frac{\left(H^3d^4K + H^5dK^2 + H^5d^3K\right)}{\epsilon_u^2} + \frac{H^5K^3}{dT\epsilon_u^2}\right). \tag{6}$$

Equation (5) in Theorem 1 indicates that the estimated transition kernels of the $T$ MDPs meet the required accuracy on average. Theorem 1 further indicates that with the data collected during such reward-free exploration, if every task is further given any reward function, near-optimal policies can be found for all tasks on average as well.

For the sample complexity bound in Equation (6), the first term is inversely proportional to $T$, which captures sample-benefit for each task due to learning the $T$ models jointly. The second term is related to the number of samples to guarantee the concentration of empirical covariance matrix $\widehat{U}_h^{(n,t)}$ in order to identify desirable exploration policies. Such concentration needs to be satisfied for each task $t$ independently and can not be implemented jointly, which makes the second term independent with $T$. The third term originally arises from the model estimation error shift among $T$ tasks when implementing joint MLE-based model estimation. In the algorithm, this error shift is then transferred to $B$ defined in Equation (3), which provides an explicit upper bound for the exploration-driven reward $\widehat{b}_h^{(n,t)}$. Note that $\widehat{b}_h^{(n,t)}$ consists the PCV to guide exploration, which finally contributes to convergence of REFUEL and affects its sample complexity.

Compared with the state-of-the-art sample complexity that scales in $\widetilde{O}\left(\frac{H^5d^4K^2}{\epsilon_u^2}\right)$ [1] when performing $T$ single-task RL independently (Uehara et al., 2022), REFUEL achieves lower sample complexity when $T = \Omega\left(\frac{K}{d^5}\right)$, and it reduces the single-task sample complexity in (Uehara et al., 2022) by an order of $O\left(\min\{H^2d^2T, H^2K, d^3, dK, \frac{d^5T}{K}\}\right)$.

Technically, the joint learning feature of REFUEL requires new analytical techniques. We outline the main steps of the proof of Theorem 1 here and defer the detailed proof to Appendix A. **Step 1:** We develop a new upper bound on model estimation error for each task, which captures the advantage of joint MLE model estimation over single-task learning, as shown in Proposition 1. **Step 2:** We establish the PCV as an uncertainty measure that captures the difference between the estimated and ground truth models. This justifies using such a PCV as a guidance for further exploration, as shown in Proposition 2. **Step 3:** We show that the summation of the PCVs over all iterations is sublinear with respect to the total number $N_u$ of iterations, as shown in Proposition 3, which further implies polynomial efficiency of REFUEL in learning the models.

## 5 Downstream offline and online RL

In downstream RL, the agent is given a new target task $T + 1$ under MDP $\mathcal{M}^{T+1}$ and aims to find a near-optimal policy for it. The downstream task is assumed to share the same representation

---

[1]We convert their results in discounted setting to our episodic MDP setting by replacing $1/(1 - \gamma) = \Theta(H)$. And similarly in following sections, we convert results in other settings to the same setting with us for a fair comparason .

as upstream tasks, and hence the agent can adopt the representation learned from the upstream to expedite its learning. Since the agent does not know the task-specific embedding function $\mu^{T+1}$, the agent is allowed to interact with MDP $\mathcal{M}^{T+1}$ in an online RL setting or have access to an offline dataset $\mathcal{D}_{\text{down}} = \{(s_h^\tau, a_h^\tau, r_h^\tau, s_{h+1}^\tau)\}_{\tau, h=1}^{N_{\text{off}}, H}$ in an offline RL setting, which is rolled out from some behavior policy $\rho$.

In this section, we first introduce the connections between upstream and downstream MDPs in Section 5.1. We then provide algorithms for the downstream task and characterize their suboptimality gap for **offline** RL in Section 5.2, and for **online** RL in Section 5.3, respectively.

## 5.1 Connections between upstream and downstream MDPs

In downstream RL tasks, the agent will adopt the feature $\widehat{\phi}$ learned from upstream Algorithm 1. To ensure that the feature learned in upstream can still work well in downstream tasks, upstream and downstream MDPs should have certain connections. We next elaborate some reasonable assumptions on transition kernels to build such connections.

The following reachability assumption is common in relevant literature (Modi et al., 2021).

**Assumption 2** (Reachability). For each source task $t$ in upstream, whose transition kernel is $P^{(*,t)}$, there exists a policy $\pi_t^0$ such that $\min_{s \in \mathcal{S}} \mathbb{P}_h^{(\pi_t^0, t)}(s) \geq \kappa_u$, where $\mathbb{P}_h^{(\pi_t^0, t)}(\cdot) : \mathcal{S} \to \mathbb{R}$ is the density function over $\mathcal{S}$ using policy $\pi_t^0$ to roll into state $s$ at timestep $h$.

**Assumption 3.** The state space $\mathcal{S}$ is compact.

Combining Assumption 3 with Assumption 2, it can be shown that there exists a uniform distribution on $\mathcal{S}$ with the density function $f(s) = \upsilon$, where $1/\upsilon$ is the measure of $\mathcal{S}$. Then, we denote $\mathcal{U}(\mathcal{S})$ as the uniform distribution over $\mathcal{S}$ and $\mathcal{U}(\mathcal{S}, \mathcal{A})$ as the uniform distribution over $\mathcal{S} \times \mathcal{A}$.

**Assumption 4.** For any two different models in the model class $\{(\Phi, \Psi)\}$, say $P^1(s'|s, a) = \langle \phi^1(s, a), \mu^1(s') \rangle$ and $P^2(s'|s, a) = \langle \phi^2(s, a), \mu^2(s') \rangle$, there exists a constant $C_R$ such that for all $(s, a) \in \mathcal{S} \times \mathcal{A}$ and $h \in [H]$,

$$\|P_h^1(\cdot|s, a) - P_h^2(\cdot|s, a)\|_{TV} \leq C_R \mathbb{E}_{(s_h, a_h) \sim \mathcal{U}(\mathcal{S}, \mathcal{A})} \|P_h^1(\cdot|s_h, a_h) - P_h^2(\cdot|s_h, a_h)\|_{TV}. \quad (7)$$

For normalization, we assume that for any $\phi \in \Phi$, $\|\phi(s, a)\|_2 \leq 1$ and for any $\mu \in \Psi$ and any function $g : \mathcal{S} \to [0, 1]$, $\left\| \int \mu_h(s) g(s) ds \right\|_2 \leq \sqrt{d}$.

Assumption 4 ensures that for each source task $t$ any $(s, a) \in \mathcal{S} \times \mathcal{A}$, the total variation distance between the learned $\widehat{P}^{(t)}(\cdot|s, a)$ and true transition kernels $\widehat{P}^{(*,t)}(\cdot|s, a)$ will not be too large when the expectation of the total variation distance over the entire state action space is small.

**Assumption 5.** For the underlying MDP of task $T + 1$, we assume that the reward function $r^{T+1}$ is chosen from a function class $\mathcal{R}$ with $\varepsilon$-covering number $\mathcal{N}_{\mathcal{R}}(\varepsilon)$ for any given $\varepsilon$, and $\mathcal{N}_{\mathcal{R}}(\varepsilon) = \widetilde{O}\left(\frac{1}{\varepsilon^p}\right)$. The true transition kernel $P^{(*,T+1)}$ can be $\xi$-approximated by a linear combination of $T$ source tasks, i.e. there exist $T$ (unknown) coefficients $c_1, \ldots, c_T \in [0, C_L]$ such that

$$\forall (s, a) \in \mathcal{S} \times \mathcal{A}, h \in [H], \quad \left\| P^{(*,T+1)}(\cdot|s, a) - \sum_{t=1}^T c_t P^{(*,t)}(\cdot|s, a) \right\|_{TV} \leq \xi. \quad (8)$$

Assumption 5 establishes the underlying connection between upstream source tasks and the downstream target task. The assumption on $\mathcal{N}_\varepsilon(\mathcal{R})$ is more general than the common assumption on reward under linear MDP and can be easily achieved in practice. For example, if the reward is linear with respect to an unknown feature $\widetilde{\phi} : \mathcal{S} \times \mathcal{A} \to \mathbb{R}^p$, i.e., there exists $\theta \in \mathbb{R}^p$ and $\|\theta\| \leq R$, $r(s, a) = \langle \widetilde{\phi}(s, a), \theta \rangle$, then $\mathcal{N}_\varepsilon(\mathcal{R}) = \widetilde{O}\left(\left(1 + \frac{2R}{\varepsilon}\right)^p\right)$ (see Lemma D.5 in Jin et al. (2020)), which satisfies the assumption.

Since the downstream task applies the estimated feature from the upstream, the downstream performance is highly affected by the upstream estimation accuracy of the feature. Hence, we first introduce the definition of $\epsilon$-approximate linear MDP, and provide a guarantee for the estimated feature.

**Definition 2** ($\epsilon$-Approximate linear MDP). For any $\epsilon > 0$, MDP $\mathcal{M} = (\mathcal{S}, \mathcal{A}, H, P, r)$ is an $\epsilon$-approximate linear MDP with a time-dependent feature map $\phi : \mathcal{S} \times \mathcal{A} \to \mathbb{R}^d$ if for any $h \in [H]$, there exist a time-dependent unknown (signed) measures $\mu$ over $\mathcal{S}$ such that for any $(s, a) \in \mathcal{S} \times \mathcal{A}$,

$$\|P_h(\cdot|s, a) - \langle \phi_h(s, a), \mu_h(\cdot) \rangle\|_{TV} \leq \epsilon. \quad (9)$$

Any $\phi$ satisfying Equation (9) is called an $\epsilon$-approximate feature map for $\mathcal{M}$.

Let $\xi_{\text{down}} = \xi + \frac{C_L C_R T \upsilon \epsilon_u}{\kappa_u}$. The following lemma shows that the feature $\widehat{\phi}$ learned in upstream is a $\xi_{\text{down}}$-approximate feature map and can approximate the true feature in the new task.

**Lemma 1.** *Under Assumptions 2 to 5, the output of Algorithm 1 $\widehat{\phi}$ is a $\xi_{\text{down}}$-approximate feature for MDP $\mathcal{M}^{T+1} = (\mathcal{S}, \mathcal{A}, H, P^{(*,T+1)}, r, s_1)$, i.e. there exist a time-dependent unknown (signed) measures $\widehat{\mu}^*$ over $\mathcal{S}$ such that for any $(s,a) \in \mathcal{S} \times \mathcal{A}$, we have*

$$\left\| P_h^{(*,T+1)}(\cdot|s,a) - \left\langle \widehat{\phi}_h(s,a), \widehat{\mu}_h^*(\cdot) \right\rangle \right\|_{TV} \leq \xi_{\text{down}}. \tag{10}$$

*Furthermore, for any $g : \mathcal{S} \to [0,1]$, $\left\| \int \widehat{\mu}_h^*(s)g(s)ds \right\|_2 \leq C_L \sqrt{d}$.*

The proof of Lemma 1 is presented in Appendix B.

## 5.2  Downstream Offline RL

We present our pessimistic value iteration algorithm for downstream offline RL called DOFRL and defer detailed Algorithm 2 to Appendix C. Although our algorithm shares similar design principles as traditional algorithms for linear MDPs (Jin et al., 2021), it differs from them significantly as described in the following, due to the misspecification of representation from upstream and the general rather than linear reward function adopted. For ease of exposition, we define Bellman operator $\mathbb{B}_h$ as $(\mathbb{B}_h f)(s,a) = r_h(s,a) + (P_h^{(*,T+1)} f)(s,a)$ for any $f : \mathcal{S} \times \mathcal{A} \mapsto \mathbb{R}$. The main body of the algorithm consists of a backward iteration over steps. In each iteration $h$, the agent executes the following main steps.

**Approximate Bellman update with general reward function.** We construct an estimated Bellman update $\widehat{\mathbb{B}}_h \widehat{V}_{h+1}$ based on the dataset $\mathcal{D}_{\text{down}}$ to approximate $\mathbb{B}_h \widehat{V}_{h+1}$. Here $\widehat{V}_{h+1}$ is an estimated value function constructed in the previous iteration based on $\mathcal{D}_{\text{down}}$. Since our reward function is more general and not necessarily linear in the feature, we choose to parameterized $P_h^{(*,T+1)} V_{h+1}^*$ instead of $Q_h^{*,T+1}(s,a) = r_h(s,a) + (P_h^{(*,T+1)} V_{h+1}^*)(s,a)$ due to the linear structure of $P_h^{(*,T+1)}$. Although the ground truth representation is unknown, the agent is able to use the estimated representation learned from the upstream learning to parameterize the linear term $(P_h^{(*,T+1)} V_{h+1}^*)(s,a)$. Then, the coefficient $w_h^*$ can be estimated by solving the following regularized least squares problem:

$$\widehat{w}_h = \operatorname{argmin}_{w \in \mathbb{R}^d} \lambda \|w\|_2^2 + \sum_{\tau=1}^{N_{\text{off}}} \left[ w^\top \widehat{\phi}_h(s_h^\tau, a_h^\tau) - \widehat{V}_{h+1}(s_{h+1}^\tau) \right]^2, \tag{11}$$

which has the closed-from solution $\widehat{w}_h = \Lambda_h^{-1} \sum_{\tau=1}^{N_{\text{off}}} \widehat{\phi}_h(s_h^\tau, a_h^\tau) \widehat{V}_{h+1}(s_{h+1}^\tau)$ with $\Lambda_h = \lambda I + \sum_{\tau=1}^{N_{\text{off}}} \widehat{\phi}_h(s_h^\tau, a_h^\tau) \widehat{\phi}_h(s_h^\tau, a_h^\tau)^\top$. We obtain an estimate $(\widehat{\mathbb{B}}_h \widehat{V}_{h+1})(\cdot, \cdot) = r_h(\cdot, \cdot) + \widehat{w}_h^\top \widehat{\phi}_h(\cdot, \cdot)$.

**Design pessimism for incorporating upstream misspecification of representation.** Pessimism is realized by subtracting an uncertainty metric $\Gamma_h$ from $\widehat{Q}_h$ (line 5). The uncertainty metric $\Gamma_h$ quantifies the Bellman update error $|(\widehat{\mathbb{B}}_h \widehat{V}_{h+1} - \mathbb{B}_h \widehat{V}_{h+1})(s,a)|$ at step $h$. Our selection of $\Gamma_h$ is in the form of $\xi_{\text{down}} + \beta(\widehat{\phi}_h \Lambda_h^{-1} \widehat{\phi}_h)^{1/2}$, which captures both the **approximation error** (first term) and the **estimation error** in the Bellman update (second term). Both of the above errors are affected by the misspecification of representation from upstream learning. Intuitively, $m := (\widehat{\phi}_h \Lambda_h^{-1} \widehat{\phi}_h)^{-1}$ represents the effective number of samples the agent has explored along the $\widehat{\phi}$ direction, and the penalty term $\beta/\sqrt{m}$ represents the uncertainty along the $\widehat{\phi}$ direction. By choosing a proper value for $\beta$, we can prove that with high probability, the uncertainty metric $\Gamma_h$ always upper bounds the Bellman update error.

**Select greedy policy.** Line 6 executes a greedy policy $\widehat{\pi}_h$ to maximize $\widehat{Q}_h$, which will be the output of the algorithm after the backward iteration over steps completes.

Then we make the following coverage assumption, which can be easily checked in practice.

**Assumption 6** (Feature coverage). There exists absolute constant $\kappa_\rho$ such that

$$\forall h \in [H], \quad \lambda_{\min}(\Sigma_h) \geq \kappa_\rho, \text{ where } \Sigma_h = \mathbb{E}_\rho[\widehat{\phi}_h(s_h, a_h)\widehat{\phi}_h(s_h, a_h)^\top | s_1 = s].$$

We are ready to provide our main result in the following theorem and defer the proof to Appendix C.

**Theorem 2.** *Under Assumptions 2 to 5, for any $\delta \in (0,1)$, if we set $\lambda = 1$ and $\beta = O(d\sqrt{\iota} + \sqrt{dN_{\text{off}}}\xi_{\text{down}} + \sqrt{p\log N_{\text{off}}})$, where $\iota = O(\log(pdHN_{\text{off}}\max\{\xi_{\text{down}},1\}/\delta))$, then with probability at least $1 - \delta$, the suboptimality gap of Algorithm 2 is at most*

$$V^*_{P(*,T+1),r} - V^{\widehat{\pi}}_{P(*,T+1),r} \leq 2H\xi_{\text{down}} + 2\beta \sum_{h=1}^{H} \mathbb{E}_{\pi^*}\left[\left\|\widehat{\phi}_h(s_h, a_h)\right\|_{\Lambda_h^{-1}} \Big| s_1 = s\right]. \tag{12}$$

*If Assumption 6 holds, and $N_{\text{off}} \geq 40\kappa_\rho \cdot \log(4dH/\delta)$, then the suboptimality gap is at most*

$$O\left(\kappa_\rho^{-1/2}Hd^{1/2}\xi_{\text{down}} + \kappa_\rho^{-1/2}Hd\sqrt{\frac{\log(pdHN_{\text{off}}\max\{\xi_{\text{down}},1\}/\delta)}{N_{\text{off}}}} + \kappa_\rho^{-1/2}H\sqrt{\frac{p\log N_{\text{off}}}{N_{\text{off}}}}\right). \tag{13}$$

*Further, if the linear combination misspecification error in Assumption 5 satisfies $\xi = \tilde{O}(\sqrt{d}/\sqrt{N_{\text{off}}})$, then with at most $\widetilde{O}\left(H^3dK^2TN_{\text{off}} + H^5d^3K^2T^2N_{\text{off}} + \frac{H^5K^3TN_{\text{off}}}{d^2}\right)$ trajectories collected in upstream, we have*

$$V^*_{P(*,T+1),r} - V^{\widehat{\pi}}_{P(*,T+1),r} \leq \widetilde{O}\left(\kappa_\rho^{-1/2}N_{\text{off}}^{-1/2}H\max\{d,\sqrt{p}\}\right). \tag{14}$$

To the best of our knowledge, Theorem 2 provides the first suboptimality gap for offline RL with approximation error in representation. The first result in Theorem 2 characterizes the suboptimality gap if the offline data is rolled out from a behavior policy[2]. If the offline data further satisfies the feature coverage assumption, we are able to upper bound the suboptimality gap by three terms in Equation (13). Such a result can be further simplified as $\widetilde{O}\left(Hd^{1/2}\xi_{\text{down}} + N_{\text{off}}^{-1/2}H\max\{d,\sqrt{p}\}\right)$, where the first term is a constant error that arises from the **approximation error** in Bellman update. The second term diminishes as $N_{\text{off}}$ grows to infinity, and it arises from the **estimation error** of $\widehat{w}_h$ in regularized linear regression. The dependence on $d$ and $\sqrt{p}$ arises from the model complexity of the linear function approximation class when we seek concentration of the estimated value function.

Compared to standard linear MDP with the knowledge of ground truth representation Jin et al. (2021), our suboptimality gap contains an additional upstream misspecification error (i.e., $\xi_{\text{down}}$). The remaining term yields an order of $\widetilde{O}(HdN_{\text{off}}^{-1/2})$ which matches that in Jin et al. (2021) if we specialize our reward to be linear with $p = O(d)$.

Theorem 2 also demonstrates that our downstream offline RL benefits from the learned representation in upstream. As shown in Equation (14), if the linear combination misspecification error $\xi$ is small enough, and the number of trajectories collected in upstream learning is sufficient large, then the suboptimality gap is dominated by the term $\widetilde{O}\left(N_{\text{off}}^{-1/2}H\max\{d,\sqrt{p}\}\right)$, which improves $\widetilde{O}(N_{\text{off}}^{-1/2}H^2d^2)$ of REP-LCB (Uehara et al., 2022) under low-rank MDP with unknown representation, with better dependence on $H$ and $d$ due to the benefit of upstream representation learning.

### 5.3 Downstream Online RL

We present our downstream online RL algorithm called DONRL and defer the detailed Algorithm 3 in Appendix D, where the agent is allowed to interact with the new task environment for policy optimization, and utilizes the learned feature from upstream to implement linear approximation for state-action value function with traditional UCB type of algorithms.

The algorithm consists of three iterations, one outer iteration and two inner backward iterations over steps. **Construct optimistic action value function**: In each inner backward iteration, the agent first constructs the empirical feature covariance matrices and estimates the weights $\widehat{w}$ via the trajectories collected in previous iterations. Then, the empirical covariance matrices and the weights $\widehat{w}$ are used to approximate Bellman update and construct an optimistic action value function (line 7). **Learn optimistic policy**: Next, in the forward inner iteration, the agent learns a greedy optimistic policy and uses it to explore and collect a new trajectory. After the outer iteration completes, the greedy policies are output by Algorithm 3.

Algorithm 3 differs from traditional UCB type of algorithms under linear MDP (i.e., LSVI-UCB in Jin et al. (2020)) as follows. First, Jin et al. (2020) assumes the true representation $\phi^*$ is known with a

---

[2]The result remains valid if offline data is compliant with MDP $\mathcal{M}^{T+1}$ (Assumption 2.2 in Jin et al. (2021)).

small model misspecification error, whereas our representation is obtained from upstream, and hence the upstream learning error affects the learning accuracy here. Second, Algorithm 3 is applicable to reward functions from a general function class $\mathcal{R}$, which do not need to be linear with respect to the representation. Both of the above differences will require additional techniques in our analysis of the performance guarantee.

To analyze our downstream online Algorithm 3, we need to characterize how misspecification error of the learned representation from upstream affects downstream suboptimality gap. We further need to handle the general reward rather than the linear reward. Both make our analysis different from that in Jin et al. (2020) which directly assumes an approximation of the representation and works with linear reward functions. We next provide our main result and defer the proof to Appendix D.

**Theorem 3.** *Under Assumptions 2 to 5, fix $\delta \in (0,1)$. If we set $\lambda = 1$ and $\beta_n = O\left(d\sqrt{\iota_n} + \sqrt{nd}\xi_{\text{down}} + \sqrt{p\log n}\right)$ where $\iota_n = \log\left(2pdnH\max\{\xi_{\text{down}}, 1\}/\delta\right)$. Let $\tilde{\pi}$ be the uniform mixture of $\pi^1, \ldots, \pi^n$. Then with probability at least $1 - \delta$, the suboptimality gap of Algorithm 3 satisfies*

$$V_{P^{(*,T+1)},r}^* - V_{P^{(*,T+1)},r}^{\tilde{\pi}} = \widetilde{O}\left(Hd\xi_{\text{down}} + Hd^{1/2}N_{\text{on}}^{-1/2}\max\{d, \sqrt{p}\}\right). \tag{15}$$

*Furthermore, if the linear combination misspecification error $\xi$ in Assumption 5 satisfies $\tilde{O}(\sqrt{d}/\sqrt{N_{\text{on}}})$, then with at most $\widetilde{O}\left(H^3dK^2TN_{\text{on}} + H^5d^3K^2T^2N_{\text{on}} + \frac{H^5K^3TN_{\text{on}}}{d^2}\right)$ trajectories collected in upstream, $\xi_{\text{down}}$ reduces to $\tilde{O}(\sqrt{d}/\sqrt{N_{\text{on}}})$ and the second term in Equation (15) dominates so that the the suboptimality gap is bounded as*

$$V_{P^{(*,T+1)},r}^* - V_{P^{(*,T+1)},r}^{\tilde{\pi}} = \widetilde{O}\left(Hd^{1/2}N_{\text{on}}^{-1/2}\max\{d, \sqrt{p}\}\right). \tag{16}$$

We first explain the terms in suboptimality gap in Equation (15). The first term captures how the **approximation error** of using the estimated representation from upstream affects suboptimality gap, and becomes small if such the approximation error is well controlled. The second term arises from the **estimation error** of $\widehat{w}_h$ in constructing the optimistic action value function and vanishes as the number $N_{\text{on}}$ of samples becomes large. The dependence on $d$ and $\sqrt{p}$ arises from model complexity of the linear function approximation class when we seek a concentration of the estimated value function. Thus, if the learned representation in upstream approximates the ground truth well and $\xi_{\text{down}}$ reduces to $\tilde{O}(\sqrt{d}/\sqrt{N_{\text{on}}})$, then suboptimality gap is dominated by the second term $\widetilde{O}\left(Hd^{1/2}N_{\text{on}}^{-1/2}\max\{d, \sqrt{p}\}\right)$ as in Equation (16). If we specialize rewards to be linear and $p = O(d)$, this matches suboptimality gap $\widetilde{O}\left(\sqrt{H^2d^3/N_{\text{on}}}\right)$ in linear MDP (Jin et al., 2020).

Theorem 3 also demonstrates that our downstream online RL benefits from the learned representation from upstream. Compared to the state-of-the-art work under the low-rank MDP with unknown representation (Uehara et al., 2022) which has the suboptimality gap being bounded as $\widetilde{O}(\sqrt{H^5d^4K^2/N_{\text{on}}})$, our downstream online RL has an improved suboptimality gap in terms of the dependence on $H, d$ and $K$ benefited from the upstream representation learning.

## 6 Conclusion

In this paper, we showed that a shared representation of low rank MDPs can be learned more efficiently and in a transferred manner. We provided upper bounds on the sample complexity of upstream reward-free multitask RL, and on the suboptimality gap of both downstream offline and online RL settings with the help of the learned representation from upstream. Our results show that multitask representation learning is provably more sample efficient than learning each task individually and capture the benefit of employing the learned representation from upstream to learn a near-optimal policy of a new task in downstream that shares the same representation. As future work, we note that our analysis of the downstream RL depends on some assumptions that connect the upstream and downstream MDPs. It is interesting to explore whether these assumptions can be further relaxed.

## 7 Acknowledgement

The work of S. Feng and Y. Liang was supported in part by the startup fund of the Ohio State University.

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
