# Supplementary Materials

## A   Proof of Theorem 1

We define $K = |\mathcal{A}|$, i.e., the cardinality of the action space. We next summarize frequently used notations in the following list.

$$K \qquad\qquad |\mathcal{A}|$$

$$\Pi_t^n \qquad\qquad \mathcal{U}(\pi_t^1, ..., \pi_t^{n-1})$$

$$f_h^{(n,t)}(s,a) \qquad\qquad \|\widehat{P}_h^{(n,t)}(\cdot|s,a) - P_h^{(*,t)}(\cdot|s,a)\|_{TV}$$

$$\widehat{b}_h^{(n,t)}(s_h, a_h) \qquad\qquad \min\left\{\widetilde{\alpha}_n \left\|\widehat{\phi}_h^{(n)}(s_h, a_h)\right\|_{(\widehat{U}_h^{(n,t)})^{-1}}, B\right\}$$

$$PCV\left(\widehat{P}^{(n,t)}, \widehat{b}_h^{(n,t)}, \pi_t; T\right) \qquad \sum_{h=1}^{H-1}\sqrt{\sum_{t=1}^{T}\left\{\mathbb{E}_{\substack{s_h\sim(\widehat{P}^{(n,t)},\pi_t)\\a_h\sim\pi_t}}\left[\widehat{b}_h^{(n,t)}(s_h, a_h)\right]\right\}^2}$$

$$\lambda_n \qquad\qquad O(d\log(|\Phi|nTH/\delta))$$

$$U_{h,\phi}^{(n,t)} \qquad\qquad n\mathbb{E}_{s_h\sim(P^{(*,t)},\Pi_t^n), a_h\sim\mathcal{U}(\mathcal{A})}\left[\phi(s_h, a_h)(\phi(s_h, a_h))^\top\right] + \lambda_n I_d$$

$$W_{h,\phi}^{(n,t)} \qquad\qquad n\mathbb{E}_{(s_h,a_h)\sim(P^{(*,t)},\Pi_t^n)}\left[\phi(s_h, a_h)(\phi(s_h, a_h))^\top\right] + \lambda_n I_d$$

$$\zeta_h^{(n,t)} \qquad\qquad \mathbb{E}_{\substack{s_{h-1}\sim(P^{(*,t)},\Pi_t^n)\\a_{h-1},a_h\sim\mathcal{U}(\mathcal{A})\\s_h\sim P^{(*,t)}(\cdot|s_{h-1},a_{h-1})}}\left[f_h^{(n,t)}(s_h, a_h)^2\right], \quad h \geq 2$$

$$\zeta_1^{(n,t)} \qquad\qquad \mathbb{E}_{\substack{s_1\sim(P^{(*,t)},\pi_t)\\a_1\sim\mathcal{U}(\mathcal{A})}}\left[f_1^{(n,t)}(s_1, a_1)^2\right]$$

$$\zeta_n \qquad\qquad \frac{2\log\left(2|\Phi||\Psi|^T nH/\delta\right)}{n}$$

$$\alpha_h^{(n,t)} \qquad\qquad \sqrt{nK\zeta_h^{(n,t)} + nK\zeta_{h-1}^{(n,t)} + \lambda_n d}, \quad h \geq 2$$

$$\alpha_n \qquad\qquad \sqrt{2nK\zeta_n + \lambda_n Td}$$

$$B_n \qquad\qquad \sqrt{T + \frac{2K}{d^2} + \frac{2KT\ln(|\Psi|)}{d^2\ln(n|\Phi|T/\delta)}}$$

$$B \qquad\qquad 2\sqrt{T + K/d^2}$$

$$\beta_1, \beta_2 \qquad\qquad \text{some absolute constants}$$

$$\widetilde{\alpha}_n \qquad\qquad \frac{\alpha_n}{\beta_1}$$

For consistency of notation, in some generic cases, we write $s_h \sim (P, \pi)$ for all $h$ for the entire episode. But for such cases, $s_1$ should still be understood as a fixed state and independent of transition model $P$ and policy $\pi$. For example in $\zeta_1^{(n,t)}$ as defined above, the expectation is taken with respect to $a_1 \sim \mathcal{U}(\mathcal{A})$ and fixed state $s_1$. Furthermore, $\mathcal{U}(\pi_t^1, ..., \pi_t^{n-1})$ denotes uniform mixture of previous $n-1$ exploration policies $\pi_t^1, ..., \pi_t^{n-1}$.

**Proof Overview:** The proof of Theorem 1 consists of three main steps, a final sample complexity step, and several supporting lemmas. **Step 1:** We develop a new upper bound on model estimation error for each task, which captures the advantage of joint MLE model estimation over single-task learning, as shown in Proposition 1. **Step 2:** We establish the PCV as an uncertainty measure that captures the difference between the estimated and ground truth models. This justifies using such a PCV as a guidance for further exploration, as shown in Proposition 2. **Step 3:** We show that the summation of the PCVs over all iterations is sublinear with respect to the total number $N_u$ of iterations, as shown in Proposition 3, which further implies polynomial efficiency of REFUEL in learning the models. **Complexity characterization:** Combining the three steps together with a contradiction argument yields the final sample complexity.

We provide details for the three main steps in Appendix A.1-Appendix A.3, the complexity characterization in Appendix A.4, and the supporting lemmas in Appendix A.5.

### A.1   Step 1: A new upper bound on model estimation error.

We develop a new upper bound on model estimation error for each task, which captures the advantage of joint MLE model estimation over single-task learning, as shown in Proposition 1.

**Proposition 1.** *For any $n \in [N_u]$, task $t$, policy $\pi_t$ and reward $r^t$, for all $h \geq 2$, we have*

$$\mathbb{E}_{\substack{s_h \sim (\widehat{P}^{(n,t)}, \pi_t) \\ a_h \sim \pi_t}} \left[ f_h^{(n,t)}(s_h, a_h) \right]$$

$$\leq \mathbb{E}_{\substack{s_{h-1} \sim (\widehat{P}^{(n,t)}, \pi_t) \\ a_{h-1} \sim \pi_t}} \left[ \min \left\{ \alpha_h^{(n,t)} \left\| \widehat{\phi}_{h-1}^{(n)}(s_{h-1}, a_{h-1}) \right\|_{(U_{h-1,\widehat{\phi}^{(n,t)}}^{(n,t)})^{-1}}, 1 \right\} \right], \quad (17)$$

*and for $h = 1$, we have*

$$\mathbb{E}_{a_1 \sim \pi_t} \left[ f_1^{(n,t)}(s_1, a_1) \right] \leq \sqrt{K \zeta_1^{(n,t)}}. \quad (18)$$

*Proof.* For $h = 1$,

$$\mathbb{E}_{a_1 \sim \pi_t} \left[ f_1^{(n,t)}(s_1, a_1) \right] \overset{(i)}{\leq} \sqrt{\mathbb{E}_{a_1 \sim \pi_t} \left[ f_1^{(n,t)}(s_1, a_1)^2 \right]} \overset{(ii)}{\leq} \sqrt{K \zeta_1^{(n,t)}},$$

where $(i)$ follows from Jensen's inequality, and $(ii)$ follows from importance sampling.

Then for $h \geq 2$, we derive the following bound:

$$\mathbb{E}_{\substack{s_h \sim (\widehat{P}^{(n,t)}, \pi_t) \\ a_h \sim \pi_t}} \left[ f_h^{(n,t)}(s_h, a_h) \right]$$

$$\overset{(i)}{\leq} \mathbb{E}_{\substack{s_{h-1} \sim (\widehat{P}^{(n,t)}, \pi_t) \\ a_{h-1} \sim \pi_t}} \left[ \left\| \widehat{\phi}_{h-1}^{(n)}(s_{h-1}, a_{h-1}) \right\|_{(U_{h-1,\widehat{\phi}^{(n,t)}}^{(n,t)})^{-1}} \times \right.$$

$$\left. \sqrt{nK \, \mathbb{E}_{\substack{s_{h-1} \sim (P^{(*,t)}, \Pi_t^n) \\ a_{h-1}, a_h \sim \mathcal{U}(\mathcal{A}) \\ s_h \sim P^{(*,t)}(\cdot | s_{h-1}, a_{h-1})}} \left[ f_h^{(n,t)}(s_h, a_h)^2 \right] + \lambda_n d + nK \, \mathbb{E}_{\substack{s_{h-2} \sim (P^{(*,t)}, \Pi_t^n) \\ a_{h-2}, a_{h-1} \sim \mathcal{U}(\mathcal{A}) \\ s_{h-1} \sim P^{(*,t)}(\cdot | s_{h-2}, a_{h-2})}} \left[ f_{h-1}^{(n,t)}(s_{h-1}, a_{h-1})^2 \right]} \right]$$

$$\overset{(ii)}{\leq} \mathbb{E}_{\substack{s_{h-1} \sim (\widehat{P}^{(n,t)}, \pi_t) \\ a_{h-1} \sim \pi_t}} \left[ \sqrt{nK \zeta_h^{(n,t)} + nK \zeta_{h-1}^{(n,t)} + \lambda_n d} \left\| \widehat{\phi}_{h-1}^{(n)}(s_{h-1}, a_{h-1}) \right\|_{(U_{h-1,\widehat{\phi}^{(n,t)}}^{(n,t)})^{-1}} \right]$$

$$= \mathbb{E}_{\substack{s_{h-1} \sim (\widehat{P}^{(n,t)}, \pi_t) \\ a_{h-1} \sim \pi_t}} \left[ \alpha_h^{(n,t)} \left\| \widehat{\phi}_{h-1}^{(n)}(s_{h-1}, a_{h-1}) \right\|_{(U_{h-1,\widehat{\phi}^{(n,t)}}^{(n,t)})^{-1}} \right],$$

where $(i)$ follows from Lemma 6 and because $|f_h^{(n,t)}(s_h, a_h)| \leq 1$, the first term inside the square root follows from the definition of $U_{h-1,\widehat{\phi}^{(n,t)}}^{(n,t)}$, the third term inside the square root follows from importance sampling and $(ii)$ follows from Lemma 3.

The proof is completed by noting that $|f_h^{(n,t)}(s_h, a_h)| \leq 1$. $\qquad \square$

The following corollary is a direct application of Proposition 1.

**Corollary 1** (Bounded difference of value functions). *For $n \in [N]$, any task $t$, policy $\pi_t$ and reward $r^t$, we have*

$$\left| V_{P^{(*,t)}, r^t}^{\pi_t} - V_{\widehat{P}^{(n,t)}, r^t}^{\pi_t} \right|$$

$$\leq \sum_{h=1}^{H-1} \mathbb{E}_{\substack{s_h \sim (\widehat{P}^{(n,t)}, \pi_t) \\ a_h \sim \pi_t}} \left[ \min \left\{ \alpha_h^{(n,t)} \left\| \widehat{\phi}_h^{(n)}(s_h, a_h) \right\|_{(U_{h,\widehat{\phi}^{(n,t)}}^{(n,t)})^{-1}}, 1 \right\} \right] + \sqrt{K \zeta_1^{(n,t)}}. \quad (19)$$

*Proof.* For task $t$, we have

$$\left| V_{P^{(*,t)}, r^t}^{\pi_t} - V_{\widehat{P}^{(n,t)}, r^t}^{\pi_t} \right|$$

$$\overset{(i)}{=} \left| \sum_{h=1}^{H} \underset{\substack{s_h \sim (\widehat{P}^{(n,t)}, \pi_t) \\ a_h \sim \pi_t}}{\mathbb{E}} \left[ (P_h^{(*,t)} - \widehat{P}_h^{(n,t)}) V_{h+1, P^{(*,t)}, r^t}^{\pi_t}(s_h, a_h) \right] \right|$$

$$\overset{(ii)}{\leq} \sum_{h=1}^{H} \underset{\substack{s_h \sim (\widehat{P}^{(n,t)}, \pi_t) \\ a_h \sim \pi_t}}{\mathbb{E}} \left[ f_h^{(n,t)}(s_h, a_h) \right]$$

$$\overset{(iii)}{\leq} \sum_{h=2}^{H} \underset{\substack{s_{h-1} \sim (\widehat{P}^{(n,t)}, \pi_t) \\ a_{h-1} \sim \pi_t}}{\mathbb{E}} \left[ \min \left\{ \alpha_h^{(n,t)} \left\| \widehat{\phi}_{h-1}^{(n)}(s_{h-1}, a_{h-1}) \right\|_{(U_{h-1,\widehat{\phi}^{(n,t)}}^{(n,t)})^{-1}}, 1 \right\} \right] + \sqrt{K \zeta_1^{(n,t)}},$$

where $(i)$ follows from Lemma 15, $(ii)$ follows from the fact that $V_{P^{(*,t)}, r^t}^{\pi_t} \leq 1$, and $(iii)$ follows from Proposition 1. $\qquad\square$

## A.2   Step 2: PCV as a new uncertainty metric

In Proposition 1 and Corollary 1, we provide upper bounds on both the total variation distance between the learned model and the true model, and the difference of value functions under any policy with arbitrary rewards. For each single task, such upper bounds measure the uncertainty of model estimation and guide exploration in the next iteration. In multitask RL, although these upper bounds also hold for each individual source task under any policy and reward even with tighter terms $\alpha_h^{(n,t)}$, the RHS of Equation (19) cannot be used to guide exploration for each task $t$ individually as in single-task RL because $\zeta_h^{(n,t)}$ in $\alpha_h^{(n,t)}$ as a joint MLE Guarantee is unknown individually for each task $t$.

This motivate us to jointly consider all the explorations of all source tasks. As we establish below in Proposition 2 that our defined new notion of PCV used in Algorithm 1 is a known upper bound for both the summation of the difference of value function and the summation of total variation distance over $T$ tasks, and can therefore serve as an uncertainty quantifier to guide exploration.

**Proposition 2** (PCV as uncertainty metric)**.** *Given any $\delta \in \left(0, |\Psi|^{-\min\{T, \frac{K}{d^2}\}}\right)$, for any $n$, policy $\pi_t$ and reward function $r^t$ of task $t \in [T]$, set $\lambda_n = O(d \log(|\Phi| nTH/\delta))$. Then with probability $1 - \delta$, we have*

$$\sum_{t=1}^{T} \left| V_{P^{(*,t)}, r^t}^{\pi_t} - V_{\widehat{P}^{(n,t)}, r^t}^{\pi_t} \right| \leq PCV \left( \widehat{P}^{(n,t)}, \widehat{b}_h^{(n,t)}, \pi_t; T \right) + \sqrt{KT\zeta_n},$$

$$\sum_{t=1}^{T} \sum_{h=1}^{H} \underset{\substack{s_h \sim (\widehat{P}^{(n,t)}, \pi_t) \\ a_h \sim \pi_t}}{\mathbb{E}} \left[ f_h^{(n,t)}(s_h, a_h) \right] \leq PCV \left( \widehat{P}^{(n,t)}, \widehat{b}_h^{(n,t)}, \pi_t; T \right) + \sqrt{KT\zeta_n}.$$

*Proof.* Following from the fact that $\sum_{t=1}^{T} \zeta_h^{(n,t)} \leq \zeta_n$, for all $h$ (see Lemma 3), and the definitions of $\alpha_h^{(n,t)}$ and $\alpha_n$ for $h \geq 2$, with probability at least $1 - \delta/2$, we have

$$\sum_{t=1}^{T} (\alpha_h^{(n,t)})^2 \leq nK \sum_{t=1}^{T} \left( \zeta_h^{(n,t)} + \zeta_{h-1}^{(n,t)} \right) + \lambda_n T d \leq 2nK\zeta_n + \lambda_n T d = \alpha_n^2. \qquad (20)$$

Moreover, the ratio between $\alpha_n$ and $\alpha_h^{(n,t)}$ is upper bounded as follows:

$$
\begin{aligned}
\frac{\alpha_n}{\alpha_h^{(n,t)}} &= \sqrt{\frac{2nK\zeta_n + \lambda_n T d}{nK\zeta_h^{(n,t)} + nK\zeta_{h-1}^{(n,t)} + \lambda_n d}} \\
&\leq \sqrt{\frac{2nK\zeta_n + \lambda_n T d}{\lambda_n d}} \\
&= \sqrt{T + \frac{2K \ln(n|\Phi||\Psi|^T/\delta)}{d^2 \ln(n|\Phi|T/\delta)}}
\end{aligned}
$$

$$= \sqrt{T + \frac{2K}{d^2} + \frac{2KT\ln(|\Psi|)}{d^2\ln(n|\Phi|T/\delta)}} = B_n \leq B, \tag{21}$$

where the last inequality follows because $B_n \leq B = 2\sqrt{T + K/d^2}$ if $\delta \in (0, |\Psi|^{-\frac{\min\{T,K\}}{d^2}})$.

Following from Corollary 1, we take the summation over all source tasks and obtain

$$\sum_{t=1}^{T} \left| V_{P^{(*,t)}, r^t}^{\pi_t} - V_{\widehat{P}^{(n,t)}, r^t}^{\pi_t} \right|$$

$$\leq \sum_{h=2}^{H} \sum_{t=1}^{T} \mathbb{E}_{\substack{s_{h-1} \sim (\widehat{P}^{(n,t)}, \pi_t) \\ a_{h-1} \sim \pi_t}} \left[ \min\left\{ \alpha_h^{(n,t)} \left\| \widehat{\phi}_{h-1}^{(n)}(s_{h-1}, a_{h-1}) \right\|_{(U_{h-1,\widehat{\phi}^{(n,t)}}^{(n,t)})^{-1}}, 1 \right\} \right] + \sum_{t=1}^{T} \sqrt{K\zeta_1^{(n,t)}}$$

$$= \sum_{h=2}^{H} \sum_{t=1}^{T} \alpha_h^{(n,t)} \mathbb{E}_{\substack{s_{h-1} \sim (\widehat{P}^{(n,t)}, \pi_t) \\ a_{h-1} \sim \pi_t}} \left[ \min\left\{ \left\| \widehat{\phi}_{h-1}^{(n)}(s_{h-1}, a_{h-1}) \right\|_{(U_{h-1,\widehat{\phi}^{(n,t)}}^{(n,t)})^{-1}}, (\alpha_h^{(n,t)})^{-1} \right\} \right] + \sum_{t=1}^{T} \sqrt{K\zeta_1^{(n,t)}}$$

$$\overset{(i)}{\leq} \sum_{h=2}^{H} \sqrt{\sum_{t=1}^{T} \left[ \mathbb{E}_{\substack{s_{h-1} \sim (\widehat{P}^{(n,t)}, \pi_t) \\ a_{h-1} \sim \pi_t}} \min\left\{ \left\| \widehat{\phi}_{h-1}^{(n)}(s_{h-1}, a_{h-1}) \right\|_{(U_{h-1,\widehat{\phi}^{(n,t)}}^{(n,t)})^{-1}}, (\alpha_h^{(n,t)})^{-1} \right\} \right]^2 \left( \sum_{t=1}^{T} (\alpha_h^{(n,t)})^2 \right)}$$
$$\quad + \sqrt{KT\zeta_n}$$

$$\overset{(ii)}{\leq} \sum_{h=2}^{H} \sqrt{\sum_{t=1}^{T} \left[ \mathbb{E}_{\substack{s_{h-1} \sim (\widehat{P}^{(n,t)}, \pi_t) \\ a_{h-1} \sim \pi_t}} \min\left\{ \alpha_n \left\| \widehat{\phi}_{h-1}^{(n)}(s_{h-1}, a_{h-1}) \right\|_{(U_{h-1,\widehat{\phi}^{(n,t)}}^{(n,t)})^{-1}}, \frac{\alpha_n}{\alpha_h^{(n,t)}} \right\} \right]^2} + \sqrt{KT\zeta_n}$$

$$\overset{(iii)}{\leq} \sum_{h=2}^{H} \sqrt{\sum_{t=1}^{T} \left[ \mathbb{E}_{\substack{s_{h-1} \sim (\widehat{P}^{(n,t)}, \pi_t) \\ a_{h-1} \sim \pi_t}} \min\left\{ \alpha_n \left\| \widehat{\phi}_{h-1}^{(n)}(s_{h-1}, a_{h-1}) \right\|_{(U_{h-1,\widehat{\phi}^{(n,t)}}^{(n,t)})^{-1}}, B \right\} \right]^2} + \sqrt{KT\zeta_n}$$

$$\overset{(iv)}{\leq} \sum_{h=2}^{H} \sqrt{\sum_{t=1}^{T} \left\{ \mathbb{E}_{\substack{s_{h-1} \sim (\widehat{P}^{(n,t)}, \pi_t) \\ a_{h-1} \sim \pi_t}} \left[ \widehat{b}_{h-1}^{(n,t)} \right] \right\}^2} + \sqrt{KT\zeta_n}$$

$$= PCV\left( \widehat{P}^{(n,t)}, \widehat{b}_h^{(n,t)}, \pi_t; T \right) + \sqrt{KT\zeta_n}.$$

where in $(i)$, we apply Cauchy-Schwarz inequality for both terms and use the fact that $\sum_{t=1}^{T} \zeta_h^{(n,t)} \leq \zeta_n$ (see Lemma 3), $(ii)$ follows from Equation (20), $(iii)$ follows from Equation (21), and $(iv)$ follows from Corollary 2 in Appendix A.5.

Similarly, following from Proposition 1, we take the summation over all source tasks and obtain:

$$\sum_{t=1}^{T} \sum_{h=1}^{H} \mathbb{E}_{\substack{s_h \sim (\widehat{P}^{(n,t)}, \pi_t) \\ a_h \sim \pi_t}} \left[ f_h^{(n,t)}(s_h, a_h) \right] \leq PCV\left( \widehat{P}^{(n,t)}, \widehat{b}_h^{(n,t)}, \pi_t; T \right) + \sqrt{KT\zeta_n}.$$

$\square$

## A.3 Step 3: Sublinear accumulation of PCV

Recall that the exploration policy is derived by an oracle:

$$\pi_1^n, \ldots, \pi_T^n = \arg\max_{\pi_1, \ldots, \pi_T} PCV\left( \widehat{P}^{(n,t)}, \widehat{b}_h^{(n,t)}, \pi_t; T \right) \tag{22}$$

In this step, we show that the summation of exploration-driven reward function PCV over $n, t, h$ is sublinear with respect to the total number $N_u$ of iterations, as given in Proposition 3, which further implies polynomial efficiency of REFUEL in learning the models.

**Proposition 3.** *Given any* $\delta \in \left( 0, |\Psi|^{-\min\{T, \frac{K}{d^2}\}} \right)$, *set* $\lambda_n = O(d\log(|\Phi|nTH/\delta))$. *Then with probability* $1 - \delta$, *under exploration policy* $\{\pi_t^n\}_{n \in [N_u]}$ *for each task* $t$, *the summation of PCVs over*

*n* is sublinear with respect to the number $N_u$ of iteration rounds:

$$\sum_{n=1}^{N_u} \left\{ PCV\left(\widehat{P}^{(n,t)}, \widehat{b}_h^{(n,t)}, \pi_t^n; T\right) + \sqrt{KT\zeta_n} \right\}$$

$$\leq \frac{4\beta_2}{\beta_1} H\sqrt{N_u T d}\left(\sqrt{K^2 d \log^2(|\Phi||\Psi|^T N_u H/\delta)} + \sqrt{Kd^3 T \log^2(2N_u T H|\Phi|/\delta)} + \sqrt{d^2 B^2 \log^2(N_u T H|\Phi|/\delta)}\right)$$

$$+ 6BH^2\sqrt{N_u TKd}\left(\sqrt{K\log^2\left(|\Phi||\Psi|^T N_u H/\delta\right)} + \sqrt{d^2 \log^2(N_u T H|\Phi|/\delta)}\right). \tag{23}$$

*Proof.* We proceed the bound as follows:

$$\sum_{n=1}^{N_u} \left\{ PCV\left(\widehat{P}^{(n,t)}, \widehat{b}_h^{(n,t)}, \pi_t^n; T\right) + \sqrt{KT\zeta_n} \right\}$$

$$\leq N_u\sqrt{KT\zeta_n} + \sum_{n=1}^{N_u}\sum_{h=1}^{H-1}\sqrt{\sum_{t=1}^{T}\left\{ \mathop{\mathbb{E}}_{\substack{s_h \sim (\widehat{P}^{(n,t)},\pi_t^n) \\ a_h \sim \pi_t^n}} \left[\widehat{b}_h^{(n,t)}(s_h, a_h)\right]\right\}^2}$$

$$\overset{(i)}{\leq} N_u\sqrt{KT\zeta_n} + \underbrace{\sum_{n=1}^{N_u}\sum_{h=1}^{H-1}\sqrt{\sum_{t=1}^{T}\left\{ \mathop{\mathbb{E}}_{\substack{s_h \sim (P^{(*,t)},\pi_t^n) \\ a_h \sim \pi_t^n}} \left[\widehat{b}_h^{(n,t)}(s_h, a_h)\right]\right\}^2}}_{(a)}$$

$$+ \underbrace{\sum_{n=1}^{N_u}\sum_{h=1}^{H-1}\sqrt{\sum_{t=1}^{T}\left\{ \mathop{\mathbb{E}}_{\substack{s_h \sim (P^{(*,t)},\pi_t^n) \\ a_h \sim \pi_t^n}} \left[\widehat{b}_h^{(n,t)}(s_h, a_h)\right] - \mathop{\mathbb{E}}_{\substack{s_h \sim (\widehat{P}^{(n,t)},\pi_t^n) \\ a_h \sim \pi_t^n}} \left[\widehat{b}_h^{(n,t)}(s_h, a_h)\right]\right\}^2}}_{(b)}, \tag{24}$$

where $(i)$ follows from the fact that for any vector $x, y \in \mathbb{R}^T$, $\|x + y\|_2 \leq \|x\|_2 + \|y\|_2$. We remark here that term $(a)$ is in fact $\sum_{n=1}^{N_u} PCV\left(P^{(*,t)}, \widehat{b}_h^{(n,t)}, \pi_t^n; T\right)$, in this proposition, we develop an inequality to bound the difference of PCVs under different transitions.

In the sequel, we first upper-bound the terms $(a)$ and $(b)$, and then combine the upper bounds with Equation (24) as our final step to obtain the desired result.

**I) Bound term $(a)$.**

Denote the trace operator as $\mathrm{tr}(\cdot)$. We first obtain

$$nK\mathop{\mathbb{E}}_{\substack{s_h \sim (P^*,\Pi_t^n) \\ a_h \sim \mathcal{U}(\mathcal{A})}}\left[\widehat{b}_h^{(n,t)}(s_h, a_h)\right]$$

$$\overset{(i)}{\leq} \frac{nK\beta_2^2}{\beta_1^2}\mathop{\mathbb{E}}_{\substack{s_h \sim (P^*,\Pi_t^n) \\ a_h \sim \mathcal{U}(\mathcal{A})}}\left[\alpha_n^2\left\|\widehat{\phi}_h^{(n)}(s_h, a_h)\right\|_{(U_{h,\widehat{\phi}^{(n,t)}}^{(n,t)})^{-1}}^2\right]$$

$$\overset{(ii)}{\leq} \frac{K\beta_2^2\alpha_n^2}{\beta_1^2}\mathrm{tr}(I_d) = K\beta_2^2(2nK\zeta_n + T\lambda_n d)d = \frac{Kd\beta_2^2\alpha_n^2}{\beta_1^2}, \tag{25}$$

where $(i)$ follows from Corollary 2 and $(ii)$ follows from the following derivation:

$$n\mathop{\mathbb{E}}_{\substack{s_h \sim (P^*,\Pi_t^n) \\ a_h \sim \mathcal{U}(\mathcal{A})}}\left[\left\|\widehat{\phi}_h^{(n)}(s_h, a_h)\right\|_{(U_{h,\widehat{\phi}^{(n,t)}}^{(n,t)})^{-1}}^2\right]$$

$$= n\mathop{\mathbb{E}}_{\substack{s_h \sim (P^*,\Pi_t^n) \\ a_h \sim \mathcal{U}(\mathcal{A})}}\left[\mathrm{tr}\left(\widehat{\phi}_h^{(n,t)}(s_h, a_h)\widehat{\phi}_h^{(n,t)}(s_h, a_h)^\top(U_{h,\widehat{\phi}^{(n,t)}}^{(n,t)})^{-1}\right)\right]$$

$$=\mathrm{tr}\left(\mathbb{E}_{\substack{s_h\sim(P^*,\Pi_t^n)\\a_h\sim\mathcal{U}(\mathcal{A})}}\left[n\widehat{\phi}_h^{(n,t)}(s_h,a_h)\widehat{\phi}_h^{(n,t)}(s_h,a_h)^\top\right]\left(n\mathbb{E}_{s_h\sim(P^*,\Pi_t^n),a_h\sim\mathcal{U}(\mathcal{A})}\left[\phi(s_h,a_h)(\phi(s_h,a_h))^\top\right]+\lambda_n I_d\right)^{-1}\right)$$

$$\leq\mathrm{tr}(I_d).$$

Following from Lemma 6, for $h\geq 2$, we have

$$\mathbb{E}_{\substack{s_h\sim(P^{(*,t)},\pi_t^n)\\a_h\sim\pi_t^n}}\left[\widehat{b}_h^{(n,t)}(s_h,a_h)\Big|s_{h-1},a_{h-1}\right]$$

$$\leq\left\|\phi_{h-1}^*(s_{h-1},a_{h-1})\right\|_{(W_{h-1,\phi^*}^{(n,t)})^{-1}}\sqrt{nK\mathbb{E}_{\substack{s_h\sim(P^*,\Pi_t^n)\\a_h\sim\mathcal{U}(\mathcal{A})}}\left[\widehat{b}_h^{(n,t)}(s_h,a_h)\right]+\lambda_n dB^2}$$

$$\leq\left\|\phi_{h-1}^*(s_{h-1},a_{h-1})\right\|_{(W_{h-1,\phi^*}^{(n,t)})^{-1}}\sqrt{\frac{Kd\beta_2^2\alpha_n^2}{\beta_1^2}+\lambda_n dB^2}. \tag{26}$$

where the last inequality follows from Equation (25).

Furthermore, for $h=1$, we have

$$\mathbb{E}_{\substack{s_1\sim(P^{(*,t)},\pi_t^n)\\a_1\sim\pi_t^n}}\left[\widehat{b}_1^{(n,t)}(s_1,a_1)\right]\stackrel{(i)}{=}\mathbb{E}_{a_1\sim\pi_t^n}\left[\widehat{b}_1^{(n,t)}(s_1,a_1)\right]\stackrel{(ii)}{\leq}\sqrt{\mathbb{E}_{a_1\sim\pi_t^n}\left[\widehat{b}_1^{(n,t)}(s_1,a_1)^2\right]}$$

$$\stackrel{(iii)}{\leq}\sqrt{K\mathbb{E}_{a_1\sim\mathcal{U}(\mathcal{A})}\left[\left\|\widehat{\phi}_1^{(n)}(s_1,a_1)\right\|_{(\widehat{U}_{1,\widehat{\phi}^{(n,t)}}^{(n,t)})^{-1}}^2\alpha_n^2\right]}$$

$$\stackrel{(iv)}{\leq}\sqrt{\frac{K\beta_2^2\alpha_n^2 d}{n\beta_1^2}}, \tag{27}$$

where $(i)$ follows from the fact that the initial state $s_1$ is fixed, $(ii)$ follows from Cauchy Schwarz inequality and Jensen's inequality, $(iii)$ follows from importance sampling, and $(iv)$ follows from a step similar to Equation (25).

Substituting Equation (26) and Equation (27) into the term $(a)$, we obtain

$$\sum_{n=1}^{N_u}\sum_{h=1}^{H-1}\sqrt{\sum_{t=1}^T\left\{\mathbb{E}_{\substack{s_h\sim(P^{(*,t)},\pi_t^n)\\a_h\sim\pi_t^n}}\left[\widehat{b}_h^{(n,t)}(s_h,a_h)\right]\right\}^2}$$

$$\leq\sum_{n=1}^{N_u}\left\{\sum_{h=2}^{H-1}\sqrt{\sum_{t=1}^T\left\{\mathbb{E}_{\substack{s_h\sim(P^{(*,t)},\pi_t^n)\\a_h\sim\pi_t^n}}\left[\left\|\phi_{h-1}^*(s_{h-1},a_{h-1})\right\|_{(W_{h-1,\phi^*}^{(n,t)})^{-1}}\sqrt{\frac{Kd\beta_2^2\alpha_n^2}{\beta_1^2}+\lambda_n dB^2}\right]\right\}^2}\right.$$

$$\left.+\sqrt{\frac{TK\beta_2^2\alpha_n^2 d}{n\beta_1^2}}\right\}$$

$$\stackrel{(i)}{\leq}\frac{\beta_2}{\beta_1}\left\{\sqrt{K\alpha_{N_u}^2 d+\lambda_{N_u}dB^2}\sum_{n=1}^{N_u}\sum_{h=2}^{H-1}\sqrt{\sum_{t=1}^T\mathbb{E}_{\substack{s_h\sim(P^{(*,t)},\pi_t^n)\\a_h\sim\pi_t^n}}\left[\left\|\phi_{h-1}^*(s_{h-1},a_{h-1})\right\|_{(W_{h-1,\phi^*}^{(n,t)})^{-1}}^2\right]}\right.$$

$$\left.+2\sqrt{N_u TK\alpha_{N_u}^2 d}\right\}$$

$$\stackrel{(ii)}{\leq}\frac{\beta_2}{\beta_1}\left\{\sqrt{K\alpha_{N_u}^2 d+\lambda_{N_u}dB^2}\sum_{h=2}^{H-1}\sqrt{N_u\sum_{t=1}^T\sum_{n=1}^{N_u}\mathbb{E}_{\substack{s_h\sim(P^{(*,t)},\pi_t^n)\\a_h\sim\pi_t^n}}\left[\left\|\phi_{h-1}^*(s_{h-1},a_{h-1})\right\|_{(W_{h-1,\phi^*}^{(n,t)})^{-1}}^2\right]}\right.$$

$$\left.+2\sqrt{N_u TK\alpha_{N_u}^2 d}\right\}$$

$$\stackrel{(iii)}{\leq}\frac{\beta_2}{\beta_1}\left\{\sqrt{K\alpha_{N_u}^2 d+\lambda_{N_u}dB^2}\sum_{h=2}^{H-1}\sqrt{N_u\sum_{t=1}^T d\log\left(1+\frac{N_u}{d\lambda_1}\right)}+2\sqrt{N_u TK\alpha_{N_u}^2 d}\right\}$$

$$\leq \frac{2\beta_2}{\beta_1}\sqrt{K\alpha_{N_u}^2 d + \lambda_{N_u} dB^2} H\sqrt{N_u T d \log\left(1 + \frac{N_u}{d\lambda_1}\right)}, \tag{28}$$

where $(i)$ follows from the fact that $\frac{\beta_2}{\beta_1} \geq 1$ (see Corollary 2), $\alpha_{N_u} \geq \alpha_{N_u-1} \geq \ldots \geq \alpha_1$ and $\sum_{n=1}^{N_u} 1/\sqrt{n} \leq 1 + \int_1^{N_u} 1/\sqrt{x}\,\mathrm{d}x \leq 2\sqrt{N_u}$, $(ii)$ follows from Cauchy-Schwarz inequality, and $(iii)$ follows from Lemma 17.

**II) Bound term $(b)$.**

We proceed the derivation as follows:

$$\sqrt{\sum_{t=1}^{T}\left\{\mathop{\mathbb{E}}_{\substack{s_h \sim (P^{(*,t)},\pi_t^n)\\ a_h \sim \pi_t^n}}\left[\widehat{b}_h^{(n,t)}(s_h,a_h)\right] - \mathop{\mathbb{E}}_{\substack{s_h \sim (\widehat{P}^{(n,t)},\pi_t^n)\\ a_h \sim \pi_t^n}}\left[\widehat{b}_h^{(n,t)}(s_h,a_h)\right]\right\}^2}$$

$$\overset{(i)}{\leq}\sqrt{\sum_{t=1}^{T}\left\{\sum_{h'=1}^{h} B\mathop{\mathbb{E}}_{\substack{s_{h'} \sim (P^{(*,t)},\pi_t^n)\\ a_{h'} \sim \pi_t^n}}\left[f_{h'}^{(n,t)}(s_{h'},a_{h'})\right]\right\}^2}$$

$$\overset{(ii)}{\leq}\sqrt{B^2 \sum_{t=1}^{T} h \sum_{h'=1}^{h}\left\{\mathop{\mathbb{E}}_{\substack{s_{h'} \sim (P^{(*,t)},\pi_t^n)\\ a_{h'} \sim \pi_t^n}}\left[f_{h'}^{(n,t)}(s_{h'},a_{h'})\right]\right\}^2}$$

$$\leq B\sqrt{\sum_{t=1}^{T} H\left\{\sum_{h'=2}^{H}\left\{\mathop{\mathbb{E}}_{\substack{s_{h'} \sim (P^{(*,t)},\pi_t^n)\\ a_{h'} \sim \pi_t^n}}\left[f_{h'}^{(n,t)}(s_{h'},a_{h'})\right]\right\}^2 + \left[\mathop{\mathbb{E}}_{a_1 \sim \pi_t^n} f_1^{(n,t)}(s_1,a_1)\right]^2\right\}}$$

$$\overset{(iii)}{\leq} B\sqrt{\sum_{t=1}^{T} H \sum_{h'=2}^{H}\left\{\mathop{\mathbb{E}}_{\substack{s_{h'-1} \sim (P^{(*,t)},\pi_t^n)\\ a_{h'-1} \sim \pi_t^n}}\left[\left\|\phi_{h'-1}^*(s_{h'-1},a_{h'-1})\right\|_{(U_{h'-1,\phi^*}^{(n,t)})^{-1}}\sqrt{nK\zeta_h^{(n,t)}+\lambda_n d}\right]\right\}^2 + KHT\zeta_n}$$

$$\leq B\sqrt{H\sum_{h=1}^{H-1}\sum_{t=1}^{T}\left\{\mathop{\mathbb{E}}_{\substack{s_h \sim (P^{(*,t)},\pi_t^n)\\ a_h \sim \pi_t^n}}\left[\left\|\phi_h^*(s_h,a_h)\right\|_{(U_{h,\phi^*}^{(n,t)})^{-1}}\right]\right\}^2 (nK\zeta_h^{(n,t)} + \lambda_n d) + KHT\zeta_n}, \tag{29}$$

where $(i)$ follows from Lemma 15 with a sparse reward $r_{h'}(\cdot,\cdot) = \widehat{b}_{h'}^{(n,t)}(\cdot,\cdot)\mathbf{1}\{h' = h\}$, where $\mathbf{1}\{\cdot\}$ is the indicator function, $(ii)$ follows because $(\sum_{h'=1}^{h} x_{h'})^2 \leq h(\sum_{h'=1}^{h} x_{h'}^2)$, and $(iii)$ follows from Lemma 6, importance sampling and because $\zeta_1^{(n,t)} \leq \zeta_n$ (see Lemma 3).

We further substitute Equation (29) into the term $(b)$ and obtain

$$\sum_{n=1}^{N_u}\sum_{h=1}^{H-1} B\sqrt{H\sum_{h=1}^{H-1}\sum_{t=1}^{T}\left\{\mathop{\mathbb{E}}_{\substack{s_h \sim (P^{(*,t)},\pi_t^n)\\ a_h \sim \pi_t^n}}\left[\left\|\phi_h^*(s_h,a_h)\right\|_{(U_{h,\phi^*}^{(n,t)})^{-1}}\right]\right\}^2 (nK\zeta_h^{(n,t)}+\lambda_n d) + KHT\zeta_n}$$

$$\overset{(i)}{\leq}\sum_{h=1}^{H-1} B\sqrt{N_u H(N_u K\zeta_{N_u}+\lambda_{N_u} d)\sum_{n=1}^{N_u}\sum_{h=1}^{H-1}\sum_{t=1}^{T}\left\{\mathop{\mathbb{E}}_{\substack{s_h \sim (P^{(*,t)},\pi_t^n)\\ a_h \sim \pi_t^n}}\left[\left\|\phi_h^*(s_h,a_h)\right\|_{(U_{h,\phi^*}^{(n,t)})^{-1}}\right]\right\}^2}$$

$$+ \sum_{h=1}^{H-1} B\sqrt{KHTN_u\sum_{n=1}^{N_u}\zeta_n}$$

$$\overset{(ii)}{\leq}\sum_{h=1}^{H-1} B\sqrt{N_u H(N_u K\zeta_{N_u}+\lambda_{N_u} d)K\sum_{h=1}^{H-1}\sum_{t=1}^{T}\sum_{n=1}^{N_u}\mathop{\mathbb{E}}_{\substack{s_h \sim (P^{(*,t)},\pi_t^n)\\ a_h \sim \mathcal{U}(\mathcal{A})}}\left[\left\|\phi_h^*(s_h,a_h)\right\|_{(U_{h,\phi^*}^{(n,t)})^{-1}}^2\right]}$$

$$+ \sqrt{N_u KHTHBN_u\zeta_{N_u}}$$

$$\overset{(iii)}{\leq} HB\sqrt{N_u H^2 T(N_u K\zeta_{N_u} + \lambda_{N_u}d)Kd\log\left(1 + \frac{N_u}{d\lambda_1}\right)} + \sqrt{N_u KHT}HBN_u\zeta_{N_u}$$

, 

$$\tag{30}$$

where $(i)$ follows from Cauchy-Schwarz inequality, and because $\sqrt{x+y} \leq \sqrt{x} + \sqrt{y}$ for $x, y \geq 0$, and $nK\zeta_h^{(n,t)} + \lambda_n d \leq nK\zeta_n + \lambda_n d$, where the latter bound is increasing in $n$, $(ii)$ follows because $\sum_{n=1}^{N_u} \zeta_n \leq \log^2\left(2|\Phi||\Psi|^T N_u H/\delta\right) \leq N_u^2\zeta_{N_u}^2$, and $(iii)$ follows from Lemma 17 and importance sampling.

### III) Final step.

We substitute Equation (28) and Equation (30) into Equation (24), and have

$$\sum_{n=1}^{N_u}\left\{PCV\left(\widehat{P}^{(n,t)}, \widehat{b}_h^{(n,t)}, \pi_t^n; T\right) + \sqrt{KT\zeta_n}\right\}$$

$$\leq \sqrt{N_u KHT}HBN_u\zeta_{N_u} + \frac{2\beta_2}{\beta_1}\sqrt{K\alpha_{N_u}^2 d + \lambda_{N_u}dB^2}H\sqrt{N_u Td\log\left(1 + \frac{N_u}{d\lambda_1}\right)}$$

$$+ B\sqrt{N_u H^4 T(N_u K\zeta_{N_u} + \lambda_{N_u}d)Kd\log\left(1 + \frac{N_u}{d\lambda_1}\right)}. \tag{31}$$

Then, we substitute the definitions of $\zeta_{N_u}$, $\alpha_{N_u}$ and $\lambda_{N_u}$ into Equation (31) and simplify the expression by taking only dominating terms as follows:

$$\sum_{n=1}^{N_u}\left\{PCV\left(\widehat{P}^{(n,t)}, \widehat{b}_h^{(n,t)}, \pi_t^n; T\right) + \sqrt{KT\zeta_n}\right\}$$

$$\leq 2\sqrt{N_u KHT}HB\log\left(2|\Phi||\Psi|^T N_u H/\delta\right)$$

$$+ \frac{2\beta_2}{\beta_1}\sqrt{Kd(4K\log(2|\Phi||\Psi|^T N_u H/\delta) + d^2 T\log(2N_u TH|\Phi|/\delta)) + d^2\log(2N_u TH|\Phi|/\delta)B^2}H\sqrt{N_u Td\log\left(1 + \frac{N_u}{d\lambda_1}\right)}$$

$$+ B\sqrt{N_u H^4 T(2K\log\left(2|\Phi||\Psi|^T N_u H/\delta\right) + d^2\log(2N_u TH|\Phi|/\delta))Kd\log\left(1 + \frac{N_u}{d\lambda_1}\right)}$$

$$\leq \frac{4\beta_2}{\beta_1}H\sqrt{N_u Td}\left(\sqrt{K^2 d\log^2(|\Phi||\Psi|^T N_u H/\delta)} + \sqrt{Kd^3 T\log^2(2N_u TH|\Phi|/\delta)} + \sqrt{d^2 B^2\log^2(N_u TH|\Phi|/\delta)}\right)$$

$$+ 6BH^2\sqrt{N_u TKd}\left(\sqrt{K\log^2\left(|\Phi||\Psi|^T N_u H/\delta\right)} + \sqrt{d^2\log^2(N_u TH|\Phi|/\delta)}\right).$$

$$\square$$

## A.4 Complexity characterization: Proof of Theorem 1

Next, equipped with Propositions 1 to 3 and Corollary 1, we are able to derive the sample complexity bound of Algorithm 1.

We prove Theorem 1 by contradiction. First for any $n$ and policy $\pi_t$, we have

$$\sum_{t=1}^{T}\mathop{\mathbb{E}}_{\substack{s_h\sim(P^{(*,t)},\pi_t)\\a_h\sim\pi_t}}\left[f_h^{(n,t)}(s_h, a_h)\right]$$

$$= \sum_{t=1}^{T}\left(\mathop{\mathbb{E}}_{\substack{s_h\sim(\widehat{P}^{(n,t)},\pi_t)\\a_h\sim\pi_t}}\left[f_h^{(n,t)}(s_h, a_h)\right] - \mathop{\mathbb{E}}_{\substack{s_h\sim(P^{(*,t)},\pi_t)\\a_h\sim\pi_t}}\left[f_h^{(n,t)}(s_h, a_h)\right]\right)$$

$$+ \sum_{t=1}^{T}\mathop{\mathbb{E}}_{\substack{s_h\sim(\widehat{P}^{(n,t)},\pi_t)\\a_h\sim\pi_t}}\left[f_h^{(n,t)}(s_h, a_h)\right]$$

$$\overset{(i)}{\leq} 2\left\{PCV\left(\widehat{P}^{(n,t)}, \widehat{b}_h^{(n,t)}, \pi_t; T\right) + \sqrt{KT\zeta_n}\right\}$$

$$\overset{(ii)}{\leq} 2\left\{ PCV\left(\widehat{P}^{(n,t)}, \widehat{b}_h^{(n,t)}, \pi_t^n; T\right) + \sqrt{KT\zeta_n}\right\}, \tag{32}$$

where $(i)$ follows from Proposition 2 and $(ii)$ follows from the definition of $\{\pi_t^n\}_{t\in[T]}$ (see Equation (22)).

If for any $n \in [N_u]$, $T\epsilon_u < 2\left\{PCV\left(\widehat{P}^{(n,t)}, \widehat{b}_h^{(n,t)}, \pi_t^n; T\right) + \sqrt{KT\zeta_n}\right\}$, which is exactly the termination criteria in Algorithm 1, then

$N_u T\epsilon_u$

$$< \sum_{n=1}^{N_u} 2\left\{PCV\left(\widehat{P}^{(n,t)}, \widehat{b}_h^{(n,t)}, \pi_t^n; T\right) + \sqrt{KT\zeta_n}\right\}$$

$$\leq \tfrac{4\beta_2}{\beta_1} H\sqrt{N_u T d}\left(\sqrt{K^2 d\log^2(|\Phi||\Psi|^T N_u H/\delta)} + \sqrt{Kd^3 T\log^2(2N_u TH|\Phi|/\delta)} + \sqrt{d^2 B^2 \log^2(N_u TH|\Phi|/\delta)}\right)$$

$$+ 6BH^2\sqrt{N_u TKd}\left(\sqrt{K\log^2(|\Phi||\Psi|^T N_u H/\delta)} + \sqrt{d^2\log^2(N_u TH|\Phi|/\delta)}\right), \tag{33}$$

where the last inequality follows from Proposition 3 and Equation (32).

Note that we assume $\delta$ is small enough satisfying $\delta \leq |\Psi|^{-\frac{\min\{T,K\}}{d^2}}$. If

$$N_u > \frac{400\beta_2^2 H^2 T d^2 K^2 \log^2\left(400\beta_2^2 H^2 d^2 K^2 |\Phi| H/(\beta_1^2\delta^2\epsilon_u^2)\right)}{T\beta_1^2\epsilon_u^2},$$

then by the fact that $\forall c \geq e^2, n \geq 1, \alpha \in \mathbb{R}^+, n \geq 4c\log^2(\alpha c) \Rightarrow n \geq c\log^2(\alpha n)$, we have

$$\frac{4\beta_2}{\beta_1} H\sqrt{N_u T d}\sqrt{K^2 d\log^2(|\Phi||\Psi|^T N_u H/\delta)} \leq \frac{\epsilon_u TN_u}{5},$$

which is exactly the first term in Equation (33). Similarly, we are able to upper bound each of the other four terms by $\frac{\epsilon_u N_u}{5}$ in Equation (33) with the iteration number $N_u$ being at most:

$$\widetilde{O}\left(\frac{H^2 d^2 K^2}{T\epsilon_u^2} + \frac{(H^2 d^4 K + H^4 dK^2 + H^4 d^3 K)}{\epsilon_u^2} + \frac{H^4 K^3}{dT\epsilon_u^2}\right).$$

Combining the above bound with Equation (33), we have

$$N_u T\epsilon_u < 5 \times \frac{N_u T\epsilon_u}{5} = N_u T\epsilon_u,$$

which leads to a contradiction and shows that Algorithm 1 is able to terminate at a certain iteration $n_u$ and output desired models with the number $HN_u$ of trajectories being at most:

$$\widetilde{O}\left(\frac{H^3 d^2 K^2}{T\epsilon_u^2} + \frac{(H^3 d^4 K + H^5 dK^2 + H^5 d^3 K)}{\epsilon_u^2} + \frac{H^5 K^3}{dT\epsilon_u^2}\right).$$

Furthermore, let $\pi_t^*$ be the optimal policy under $\mathcal{M}^t$ given reward the $r^t$. And form Algorithm 1, the algorithm terminates at iteration $n_u$ and outputs $\widehat{P}^{(t)}$ for $t \in [T]$. Then we have

$$\sum_{t=1}^T V_{P^{(*,t)}, r^t}^* - V_{P^{(*,t)}, r^t}^{\widehat{\pi}_t}$$

$$= \sum_{t=1}^T V_{P^{(*,t)}, r^t}^* - V_{\widehat{P}^{(t)}, r^t}^{\pi_t^*} + V_{\widehat{P}^{(t)}, r^t}^{\pi_t^*} - V_{\widehat{P}^{(t)}, r^t}^{\widehat{\pi}_t} + V_{\widehat{P}^{(t)}, r^t}^{\widehat{\pi}_t} - V_{P^{(*,t)}, r^t}^{\widehat{\pi}_t}$$

$$\overset{(i)}{\leq} \sum_{h=1}^{H-1}\sqrt{\sum_{t=1}^T \underset{\substack{s_h\sim(\widehat{P}^{(t)},\pi_t^*)\\a_h\sim\pi_t^*}}{\mathbb{E}}\left[\widehat{b}_h^{(n_u,t)}(s_h,a_h)\right]^2} + \sqrt{KT\zeta_{n_u}}$$

$$+ \sum_{h=1}^{H-1}\sqrt{\sum_{t=1}^T \underset{\substack{s_h\sim(\widehat{P}^{(t)},\widehat{\pi}_t)\\a_h\sim\widehat{\pi}_t}}{\mathbb{E}}\left[\widehat{b}_h^{(n_u,t)}(s_h,a_h)\right]^2} + \sqrt{KT\zeta_{n_u}}$$

$$\leq 2\left\{PCV\left(\widehat{P}^{(t)}, \widehat{b}_h^{(n_u,t)}, \pi_t^{n_u}; T\right) + \sqrt{KT\zeta_{n_u}}\right\}$$

$$\overset{(ii)}{\leq} T\epsilon_u,$$

where $(i)$ follows from the definition of $\{\pi_t^*\}_{t\in[T]}$ and Proposition 2, and $(ii)$ follows from the termination criteria of Algorithm 1.

## A.5 Supporting Lemmas

Recall $U_{h,\phi}^{(n,t)} = n\mathbb{E}_{s_h\sim(P^*,\Pi_t^n),a_h\sim\mathcal{U}(\mathcal{A})}\left[\phi(s_h,a_h)(\phi(s_h,a_h))^\top\right] + \lambda_n I$. Then $U_{h,\widehat{\phi}^{(n)}}^{(n,t)}$ is the counterpart of $\widehat{U}_h^{(n,t)}$ in expectation. The following lemma provides the concentration of the bonus term. See Lemma 39 in Zanette et al. (2020) for the version of fixed $\phi$ and Lemma 11 in Uehara et al. (2022).

**Lemma 2.** *(Concentration of the bonus term). Fix $\delta \in (0,1)$, and set $\lambda_n = \Theta(d\log(2nTH|\Phi|/\delta))$ for any $n$. With probability at least $1 - \delta/2$, we have that $\forall n \in \mathbb{N}^+, h \in [H], t \in [T], \widehat{\phi} \in \Phi$,*

$$\beta_1\left\|\widehat{\phi}_h^{(n)}(s,a)\right\|_{(U_{h,\widehat{\phi}}^{(n,t)})^{-1}} \leq \left\|\widehat{\phi}_h^{(n)}(s,a)\right\|_{(\widehat{U}_h^{(n,t)})^{-1}} \leq \beta_2\left\|\widehat{\phi}_h^{(n)}(s,a)\right\|_{(U_{h,\widehat{\phi}}^{(n,t)})^{-1}}.$$

Since $\widehat{b}_h^{(n,t)}(s_h,a_h) = \min\left\{\widetilde{\alpha}_n\left\|\widehat{\phi}_h^{(n)}(s,a)\right\|_{(\widehat{U}_h^{(n,t)})^{-1}}, B\right\}$. Setting $\widetilde{\alpha}_n = \frac{\alpha_n}{\beta_1}$ and applying Lemma 2, we can immediately obtain the following corollary.

**Corollary 2.** *Fix $\delta \in (0,1)$, under the same setting of Lemma 2, with probability at least $1 - \delta/2$, we have that $\forall n \in \mathbb{N}^+, h \in [H], \phi \in \Phi$,*

$$\min\left\{\alpha_n\left\|\widehat{\phi}_h^{(n)}(s_h,a_h)\right\|_{(U_{h,\widehat{\phi}}^{(n,t)})^{-1}}, B\right\} \leq \widehat{b}_h^{(n,t)}(s_h,a_h) \leq \frac{\beta_2}{\beta_1}\alpha_n\left\|\widehat{\phi}_h^{(n)}(s_h,a_h)\right\|_{(U_{h,\widehat{\phi}}^{(n,t)})^{-1}}.$$

Recall that $f_h^{(n,t)}(s,a) = \|\widehat{P}_h^{(n,t)}(\cdot|s,a) - P_h^{(*,t)}(\cdot|s,a)\|_{TV}$ represents the estimation error of task $t$ in terms of the total variation distance in the $n$-th iteration at step $h$, given state $s$ and action $a$ in Algorithm 1. Inspired by the proof of Theorem 21 in Agarwal et al. (2020), We show that if we uniformly choose the exploration policies for each task, the summation of the estimation error can be bounded with high probability.

**Lemma 3** (Multitask MLE guarantee). *Given $\delta \in (0,1)$, consider the transition kernels learned from line 8 and 10 in Algorithm 1, we have the following inequality holds for any $n, h \geq 2$ with probability at least $1 - \delta/2$:*

$$\sum_{t=1}^T \underset{\substack{s_{h-1}\sim(P^{(*,t)},\Pi_t^n)\\a_{h-1},a_h\sim\mathcal{U}(\mathcal{A})\\s_h\sim P^{(*,t)}(\cdot|s_{h-1},a_{h-1})}}{\mathbb{E}}\left[f_h^{(n,t)}(s_h,a_h)^2\right] \leq \zeta_n, \quad where \ \zeta_n := \frac{2\log\left(2|\Phi||\Psi|^T nH/\delta\right)}{n}. \quad (34)$$

*In addition, for $h = 1$,*

$$\sum_{t=1}^T \underset{a_1\sim\mathcal{U}(\mathcal{A})}{\mathbb{E}}\left[f_1^{(n,t)}(s_1,a_1)^2\right] \leq \zeta_n.$$

*Furthermore, define*

$$\zeta_h^{(n,t)} = \underset{\substack{s_{h-1}\sim(P^{(*,t)},\Pi_t^n)\\a_{h-1},a_h\sim\mathcal{U}(\mathcal{A})\\s_h\sim P^{(*,t)}(\cdot|s_{h-1},a_{h-1})}}{\mathbb{E}}\left[f_h^{(n,t)}(s_h,a_h)^2\right], h \geq 2, \quad (35)$$

$$\zeta_1^{(n,t)} = \underset{\substack{s_1\sim(P^{(*,t)},\pi_t)\\a_1\sim\mathcal{U}(\mathcal{A})}}{\mathbb{E}}\left[f_1^{(n,t)}(s_1,a_1)^2\right]. \quad (36)$$

*We have*

$$\zeta_h^{(n,t)} \leq \sum_{t=1}^T \zeta_h^{(n,t)} \leq \zeta_n = \frac{2\log\left(2|\Phi||\Psi|^T nH/\delta\right)}{n}. \quad (37)$$

*Proof of Lemma 3.* Consider a sequential conditional probability estimation setting with an instance space $\mathcal{X}$ and a target space $\mathcal{Y}$ where the conditional density is given by $p(y|x) = f^*(x, y)$. We are given a dataset $D := \{(x_i, y_i)\}_{i=1}^n$, where $x_i \sim \mathcal{D}_i = \mathcal{D}_i(x_{1:i-1}, y_{1:i-1})$ and $y_i \sim p(\cdot|x_i)$. Let $D'$ denote a tangent sequence $\{(x_i', y_i')\}_{i=1}^n$ where $x_i' \sim \mathcal{D}_i(x_{1:i-1}, y_{1:i-1})$ and $y_i' \sim p(\cdot|x_i')$. Further, we consider a function class $\mathcal{F} : (\mathcal{X} \times \mathcal{Y}) \to R$ and assume that the reachability condition $f^* \in \mathcal{F}$ holds.

We first introduce two useful lemmas from Agarwal et al. (2020).

**Lemma 4** (Lemma 25 of Agarwal et al. (2020)). *For any two conditional probability densities $f_1, f_2$ and any distribution $\mathcal{D} \in \triangle(\mathcal{X})$, we have*

$$\mathbb{E}_{x \sim D}\|f_1(x, \cdot) - f_2(x, \cdot)\|_{TV}^2 \leq -2\log \mathbb{E}_{x \sim \mathcal{D}, y \sim f_2(\cdot|x)}\left[\exp\left(-\frac{1}{2}\log(f_2(x, y)/f_1(x, y))\right)\right]$$

**Lemma 5** (Lemma 24 of Agarwal et al. (2020)). *Let $D$ ba a dataset of $n$ samples and $D'$ be corresponding tangent sequence. Let $L(f, D) = \sum_{i=1}^n l(f, (x_i, y_i))$ be any function that decomposes additively across examples where $l$ is any function, and let $\widehat{f}(D)$ be any estimator taking as input random variable $D$ and with range $\mathcal{F}$. Then*

$$\mathbb{E}_D\left[\exp\left(L(\widehat{f}(D), D) - \log\mathbb{E}_{D'}\left[\exp(L(\widehat{f}(D), D'))\right] - \log|\mathcal{F}|\right)\right] \leq 1.$$

Suppose $\widehat{f}(D)$ is learned from the following maximum likelihood problem:

$$\widehat{f}(D) := \arg\max_{f \in \mathcal{F}} \sum_{(x_i, y_i) \in D} \log f(x_i, y_i). \tag{38}$$

Combining Chernoff method and Lemma 5, we obtain an exponential tail bound, i.e., with probability at least $1 - \delta$,

$$-\log\mathbb{E}_{D'}\left[\exp(L(\widehat{f}(D), D'))\right] \leq -L(\widehat{f}(D), D) + \log|\mathcal{F}| + \log(1/\delta). \tag{39}$$

To proceed, we let $L(f, D) = \sum_{i=1}^n -\frac{1}{2}\log(f^*(x_i, y_i)/f(x_i, y_i))$ where $D$ is a dataset $\{(x_i, y_i)\}_{i=1}^n$ (and $D' = \{(x_i', y_i')\}_{i=1}^n$ is tangent sequence). In multitask RL setting, let $x = \{(s^t, a^t)\}_{t=1}^T, y = \{(s')^t\}_{t=1}^T$ and $f(x, y) = \prod_{t=1}^T P^t[(s')^t|s^t, a^t]$. Then, dataset $D$ can be decomposed into $D = \bigcup_{t=1}^T D^t$ where $D^t = \{s_i^t, a_i^t, (s')_i^t\}_{i=1}^n$. Similarly $D' = \bigcup_{t=1}^T (D')^t$, and $\mathcal{D}_i^t := \mathcal{D}_i^t(s_{1:i-1}^t, a_{1:i-1}^t, (s')_{1:i-1}^t)$. Hence, the cardinality $|\mathcal{F}| = |\Phi||\Psi|^T$ in the multitask setting.

Then, the RHS of Equation (39) can be bounded as

$$\text{RHS of Equation (39)} = \sum_{i=1}^n \frac{1}{2}\log(f^*(x_i, y_i)/\widehat{f}(x_i, y_i)) + \log|\mathcal{F}| + \log(1/\delta)$$
$$\leq \log|\mathcal{F}| + \log(1/\delta) = \log\left(|\Phi||\Psi|^T/\delta\right), \tag{40}$$

where the inequality follows because $\widehat{f}$ is MLE and from the assumption of reachability, and the last equality follows because $|\mathcal{F}| = |\Phi||\Psi|^T$.

Next, the LHS of Equation (39) can be bounded as

$$\text{LHS of Equation (39)} \stackrel{(i)}{=} -\log\mathbb{E}_{D'}\left[\exp\left(\sum_{i=1}^n -\frac{1}{2}\log\left(\frac{f^*(x_i', y_i')}{\widehat{f}(x_i', y_i')}\right)\right)\Big|D\right]$$

$$\stackrel{(ii)}{=} -\log\mathbb{E}_{D'}\left[\exp\left(\sum_{i=1}^n -\frac{1}{2}\log\left(\prod_{t=1}^T \frac{P^{(*,t)}[(s')_i^t|s_i^t, a_i^t]}{\widehat{P}^{(n,t)}[(s')_i^t|s_i^t, a_i^t]}\right)\right)\Big|D\right]$$

$$\stackrel{(iii)}{=} -\sum_{t=1}^T \log\mathbb{E}_{(D')^t}\left[\exp\left(\sum_{i=1}^n -\frac{1}{2}\log\left(\frac{P^{(*,t)}[(s')_i^t|s_i^t, a_i^t]}{\widehat{P}^{(n,t)}[(s')_i^t|s_i^t, a_i^t]}\right)\right)\Big|D\right]$$

$$\stackrel{(iv)}{=} -\sum_{t=1}^T \sum_{i=1}^n \log\mathbb{E}_{D_i^t}\left[\exp\left(-\frac{1}{2}\log\left(\frac{P^{(*,t)}[(s')_i^t|s_i^t, a_i^t]}{\widehat{P}^{(n,t)}[(s')_i^t|s_i^t, a_i^t]}\right)\right)\right]$$

$$\overset{(v)}{\geq} \sum_{t=1}^{T} \frac{1}{2} \sum_{i=1}^{n} \mathbb{E}_{(s,a)\sim\mathcal{D}_i^t} \left\| \widehat{P}^{(n,t)}(\cdot|s,a) - P^{(*,t)}(\cdot|s,a) \right\|_{TV}^2$$

$$\overset{(vi)}{=} \frac{n}{2} \sum_{t=1}^{T} \mathbb{E}_{\substack{s_{h-1}\sim(P^{(*,t)},\Pi_t^n) \\ a_{h-1},a_h\sim\mathcal{U}(\mathcal{A}) \\ s_h\sim P^{(*,t)}(\cdot|s_{h-1},a_{h-1})}} \left[ f_h^{(n,t)}(s_h,a_h)^2 \right], \tag{41}$$

where $(i)$ follows from the above definition of $L(f,D)$, $(ii)$ follows from the above definition of $f(x,y)$, $(iii)$ follows because the data of $T$ tasks are independent conditional on $D$, $(iv)$ follows because $\widehat{P}^{(n,t)}$ is independent of the dataset $(D')^t$ and from the definition of $D'$, $(v)$ follows from Lemma 4, and $(vi)$ follows because the data collected in $i$-th iteration uses policy $\pi_{i-1}^t$ followed by two steps of uniform random actions and from the definition of $\Pi_n^t$.

Combining Equations (39) to (41), we have

$$\frac{n}{2} \sum_{t=1}^{T} \mathbb{E}_{\substack{s_{h-1}\sim(P^{(*,t)},\Pi_t^n) \\ a_{h-1},a_h\sim\mathcal{U}(\mathcal{A}) \\ s_h\sim P^{(*,t)}(\cdot|s_{h-1},a_{h-1})}} \left[ f_h^{(n,t)}(s_h,a_h)^2 \right] \leq \log\left(|\Phi||\Psi|^T/\delta\right). \tag{42}$$

We substitute $\delta$ with $\delta/2nH$ to ensure Equation (42) holds for any $h \in [H]$ and $n$ with probability at least $1 - \delta/2$, which finishes the proof. $\qquad\square$

We next introduce a one-step back lemma, which extends the one-step back inequality for infinite-horizon stationary MDP in Uehara et al. (2022); Agarwal et al. (2020) to non-stationary transition kernels with finite horizon. The lemma shows that for any function $g \in \mathcal{S} \times \mathcal{A} \to \mathbb{R}$, policy $\pi$ and transition kernel $P$, we can upper bound the expectation $\mathbb{E}_{\substack{s_h\sim(P,\pi) \\ a_h\sim\pi}}[g(s_h,a_h)]$ by the product of two terms. The first term represents the convergence guarantee of $g(s_h,a_h)$ following other policies, which is $\mathbb{E}_{\substack{s_h\sim(P^*,\Pi) \\ a_h\sim\mathcal{U}(\mathcal{A})}}[g^2(s_h,a_h)]$. The second term can be described as the distribution shift coefficient $\mathbb{E}_{\substack{s_{h-1}\sim(P,\pi) \\ a_{h-1}\sim\pi}} \left[ \|\phi_{h-1}(s_{h-1},a_{h-1})\|_{(U_{h-1,\phi})^{-1}} \right]$, which measures the difference caused by distribution shift from $\pi$ and other policies.

**Lemma 6** (One-step back inequality for non-stationary finite-horizon MDP). *For each task $t$, let $P \in \{\widehat{P}^{(n,t)}, P^{(*,t)}\}$ with embeddings $\phi$ and $\mu$ be a generic MDP model, and $U_{h,\phi}^t = \lambda I + n\mathbb{E}_{s_h,a_h\sim(P^{(*,t)},\Pi)}[\phi\phi^\top] \in \{U_{h,\phi}^{(n,t)}, W_{h,\phi}^{(n,t)}\}$ be the covariance matrix following a generic policy $\Pi$ under the true environment $P^{(*,t)}$. Note that $\phi \in \{\widehat{\phi}^{(n)}, \phi^*\}$ corresponds to $P$. Further, let $f^t(s_h,a_h)$ be the total variation between $P^{(*,t)}$ and $P$ at time step $h$. Take any $g \in \mathcal{S} \times \mathcal{A} \to \mathbb{R}$ such that $\|g\|_\infty \leq B_g$, i.e., $\sup_{s,a} |g(s,a)| \leq B_g$. Then, $\forall h \geq 2, \forall$ policy $\pi$,*

$$\mathbb{E}_{\substack{s_h\sim(P,\pi) \\ a_h\sim\pi}}[g(s_h,a_h)] \leq \mathbb{E}_{\substack{s_{h-1}\sim(P,\pi) \\ a_{h-1}\sim\pi}} \left[ \|\phi_{h-1}(s_{h-1},a_{h-1})\|_{(U_{h-1,\phi}^t)^{-1}} \times \right.$$
$$\left. \sqrt{nK\mathbb{E}_{\substack{s_h\sim(P^{(*,t)},\Pi) \\ a_h\sim\mathcal{U}(\mathcal{A})}}[g^2(s_h,a_h)]+\lambda dB_g^2+nB_g^2\mathbb{E}_{\substack{s_{h-1}\sim(P^{(*,t)},\Pi) \\ a_{h-1}\sim\Pi}}[f^t(s_{h-1},a_{h-1})^2]} \right].$$

*Proof.* First, we have

$$\mathbb{E}_{\substack{s_h\sim(P,\pi) \\ a_h\sim\pi}}[g(s_h,a_h)]$$

$$= \mathbb{E}_{\substack{s_{h-1}\sim(P,\pi) \\ a_{h-1}\sim\pi}} \left[ \int_{s_h} \sum_{a_h} g(s_h,a_h)\pi(a_h|s_h)\langle\phi_{h-1}(s_{h-1},a_{h-1}),\mu_{h-1}(s_h)\rangle ds_h \right]$$

$$\leq \mathbb{E}_{\substack{s_{h-1}\sim(P,\pi) \\ a_{h-1}\sim\pi}} \left[ \|\phi_{h-1}(s_{h-1},a_{h-1})\|_{(U_{h-1,\phi}^t)^{-1}} \left\| \int \sum_{a_h} g(s_h,a_h)\pi(a_h|s_h)\mu_{h-1}(s_h)ds_h \right\|_{U_{h-1,\phi}^t} \right],$$

where the inequality follows from Cauchy's inequality. We further develop the following bound:

$$\left\|\int \sum_{a_h} g(s_h,a_h)\pi(a_h|s_h)\mu_{h-1}(s_h)ds_h\right\|_{U^t_{h-1,\phi}}^2$$

$$\overset{(i)}{\leq} n \underset{\substack{s_{h-1}\sim(P^{(*,t)},\Pi)\\ a_{h-1}\sim\Pi}}{\mathbb{E}}\left[\left(\int_{s_h}\sum_{a_h}g(s_h,a_h)\pi(a_h|s_h)\mu(s_h)^\top\phi(s_{h-1},a_{h-1})ds_h\right)^2\right]+\lambda dB_g^2$$

$$\leq n \underset{\substack{s_{h-1}\sim(P^{(*,t)},\Pi)\\ a_{h-1}\sim\Pi}}{\mathbb{E}}\left[\underset{\substack{s_h\sim P(\cdot|s_{h-1},a_{h-1})\\ a_h\sim\pi}}{\mathbb{E}}\left[g(s_h,a_h)^2\right]\right]+\lambda dB_g^2$$

$$\overset{(ii)}{\leq} n \underset{\substack{s_{h-1}\sim(P^{(*,t)},\Pi)\\ a_{h-1}\sim\Pi}}{\mathbb{E}}\left[\underset{\substack{s_h\sim P^{(*,t)}\\ a_h\sim\pi}}{\mathbb{E}}\left[g(s_h,a_h)^2\right]\right]+\lambda dB_g^2+nB_g^2\underset{\substack{s_{h-1}\sim(P^{(*,t)},\Pi)\\ a_{h-1}\sim\Pi}}{\mathbb{E}}\left[f^t(s_{h-1},a_{h-1})^2\right]$$

$$\overset{(iii)}{\leq} nK \underset{\substack{s_h\sim(P^{(*,t)},\Pi)\\ a_h\sim\mathcal{U}(\mathcal{A})}}{\mathbb{E}}\left[g(s_h,a_h)^2\right]+\lambda dB_g^2+nB_g^2\underset{\substack{s_{h-1}\sim(P^{(*,t)},\Pi)\\ a_{h-1}\sim\Pi}}{\mathbb{E}}\left[f^t(s_{h-1},a_{h-1})^2\right],$$

where $(i)$ follows from the assumption $\|g\|_\infty\leq B_g$, $(ii)$ follows because $f(s_h,a_h)$ is the total variation between $P^*$ and $P$ at time step $h$, and $(iii)$ follows from importance sampling. This finishes the proof. $\qquad\square$

# B Proof of Lemma 1

Lemma 1 serves a central role for bridging the upstream and downstream learning, which shows that the feature $\widehat{\phi}$ learned in upstream is a $\xi_{down}$-approximate feature map and can approximate the true feature in the new task.

*Proof of Lemma 1.* Under Assumptions 2 to 4, for any $t\in[T]$, we have

$$\max_{s\in\mathcal{S},a\in\mathcal{A}}\|P_h^1(\cdot|s,a)-P_h^2(\cdot|s,a)\|_{TV}$$

$$\overset{(i)}{\leq} C_R \underset{(s_h,a_h)\sim\mathcal{U}(\mathcal{S},\mathcal{A})}{\mathbb{E}}\|P_h^1(\cdot|s_h,a_h)-P_h^2(\cdot|s_2,a_2)\|_{TV}$$

$$\overset{(ii)}{\leq} \frac{C_R\upsilon}{\kappa_u}\underset{\substack{s_h\sim(P^{(*,t)},\pi_t^0)\\ a_h\sim\mathcal{U}(\mathcal{A})}}{\mathbb{E}}\left[\left\|P_h^1(\cdot|s,a)-P_h^2(\cdot|s,a)\right\|_{TV}\right], \tag{43}$$

where $(i)$ follows from Assumption 4 and $(ii)$ follows Assumption 2 and Assumption 3.

Then, $\forall(s,a)\in\mathcal{S}\times\mathcal{A}, h\in[H]$, we have

$$\sum_{t=1}^T\|\widehat{P}_h^{(t)}(\cdot|s,a)-P_h^{(*,t)}(\cdot|s,a)\|_{TV}\leq\sum_{t=1}^T\max_{s\in\mathcal{S},a\in\mathcal{A}}\|\widehat{P}_h^{(t)}(\cdot|s,a)-P_h^{(*,t)}(\cdot|s,a)\|_{TV}$$

$$\overset{(i)}{\leq}\frac{C_R\upsilon}{\kappa_u}\sum_{t=1}^T\underset{\substack{s_h\sim(P^{(*,t)},\pi_t^0)\\ a_h\sim\mathcal{U}(\mathcal{A})}}{\mathbb{E}}\left[\left\|\widehat{P}_h^{(t)}(\cdot|s,a)-P_h^{(*,t)}(\cdot|s,a)\right\|_{TV}\right]$$

$$\overset{(ii)}{\leq}\frac{C_RT\upsilon\epsilon_u}{\kappa_u}, \tag{44}$$

where $(i)$ follows from Equation (43), and $(ii)$ follows from Theorem 1.

Define $\widehat{\mu}^*(\cdot)=\sum_{t=1}^T c_t\widehat{\mu}^{(t)}(\cdot)$, then we have

$$\left\|P_h^{(*,T+1)}(\cdot|s,a)-\left\langle\widehat{\phi}_h(s,a),\widehat{\mu}_h^*(\cdot)\right\rangle\right\|_{TV}$$

$$= \left\| P_h^{(*,T+1)}(\cdot|s,a) - \left\langle \widehat{\phi}_h(s,a), \sum_{t=1}^{T} c_t \widehat{\mu}_h^{(t)}(\cdot) \right\rangle \right\|_{TV}$$

$$\leq \left\| P_h^{(*,T+1)}(\cdot|s,a) - \sum_{t=1}^{T} c_t \widehat{P}_h^{(t)}(\cdot|s,a) \right\|_{TV}$$

$$\leq \left\| P_h^{(*,T+1)}(\cdot|s,a) - \sum_{t=1}^{T} c_t P_h^{(*,t)}(\cdot|s,a) \right\|_{TV} + \sum_{t=1}^{T} c_t \left\| P_h^{(*,t)}(\cdot|s,a) - \widehat{P}_h^{(t)}(\cdot|s,a) \right\|_{TV}$$

$$\overset{(i)}{\leq} \xi + \frac{C_L C_R T v \epsilon_u}{\kappa_u},$$

where $(i)$ follows from Assumption 5, Equation (44) and the fact that $c_t \in [0, C_L]$.

Furthermore, by normalization for any $g : \mathcal{S} \to [0,1]$, we obtain

$$\left\| \int \widehat{\mu}_h^*(s) g(s) ds \right\|_2 \leq \sum_{t=1}^{T} c_t \left\| \int \widehat{\mu}^{(t)}(s) g(s) ds \right\|_2 \leq C_L \sqrt{d}.$$

$\square$

# C   Algorithm 2 and Proof of Theorem 2

---

**Algorithm 2 DOFRL (Downstream OFfline RL)**

---

1: **Input:** Feature $\widehat{\phi}$, dataset $\mathcal{D}_{\text{down}} = \{(s_h^\tau, a_h^\tau, r_h^\tau, s_{h+1}^\tau)\}_{\tau,h=1}^{N_{\text{off}},H}$, parameters $\lambda$, $\beta$, $\xi_{\text{down}}$.

2: **Initialization:** $\widehat{V}_{H+1} = 0$.

3: **for** $h = H, H-1, \ldots, 1$ **do**

4:    $\widehat{w}_h = \Lambda_h^{-1} \sum_{\tau=1}^{N_{\text{off}}} \widehat{\phi}_h(s_h^\tau, a_h^\tau) \widehat{V}_{h+1}(s_{h+1}^\tau)$ where $\Lambda_h = \sum_{\tau=1}^{N_{\text{off}}} \widehat{\phi}_h(s_h^\tau, a_h^\tau) \widehat{\phi}_h(s_h^\tau, a_h^\tau)^\top + \lambda I_d$.

5:    $\widehat{Q}_h(\cdot,\cdot) = \min\{r_h(\cdot,\cdot) + \widehat{\phi}_h(\cdot,\cdot)^\top \widehat{w} - \Gamma_h(\cdot,\cdot), 1\}^+$, where $\Gamma_h(\cdot,\cdot) = \xi_{\text{down}} + \beta[\widehat{\phi}_h(\cdot,\cdot)^\top \Lambda_h^{-1} \widehat{\phi}_h(\cdot,\cdot)]^{1/2}$.

6:    $\widehat{V}_h(\cdot) = \widehat{Q}_h(\cdot, \widehat{\pi}_h(\cdot))$, where $\widehat{\pi}_h(\cdot) = \arg\max_{\pi_h} \widehat{Q}_h(\cdot, \widehat{\pi}_h(\cdot))$.

7: **Output:** $\{\widehat{\pi}_h\}_{h=1}^{H}$.

---

Recall $\xi_{\text{down}} = \xi + \frac{C_L C_R T v \epsilon_u}{\kappa_u}$ and for any $h \in [H]$, we define

$$P_h^{(*,T+1)}(\cdot|s,a) = \langle \phi_h^*(s,a), \mu_h^{(*,T+1)}(\cdot) \rangle,$$

$$\overline{P}_h(\cdot|s,a) = \langle \widehat{\phi}_h(s,a), \widehat{\mu}_h^*(\cdot) \rangle.$$

Given a reward function $r$, for any function $f : \mathcal{S} \mapsto \mathbb{R}$ and $h \in [H]$, we define the transition operators and their corresponding Bellman operators as

$$(P_h^{(*,T+1)} f)(s,a) = \int_{s'} \langle \phi_h^*(s,a), \mu_h^{(*,T+1)}(s') \rangle f(s') ds',$$

$$(\mathbb{B}_h f)(s,a) = r_h(s,a) + (P_h^{(*,T+1)} f)(s,a),$$

$$(\overline{P}_h f)(s,a) = \int_{s'} \widehat{\phi}_h(s,a) \widehat{\mu}_h^*(s') f(s') ds',$$

$$(\overline{\mathbb{B}}_h f)(s,a) = r_h(s,a) + (\overline{P}_h f)(s,a).$$

We further denote $(\widehat{\mathbb{B}}_h \widehat{V}_{h+1})(s,a) = r_h(s,a) + \widehat{\phi}_h(s,a)^\top \widehat{w}_h$, $h \in [H]$.

We remark here throughout this section, the expectation is taken with respect to the transition kernel of the target task, i.e., $P^{(*,T+1)}$.

**Proof Overview:** The proof of Theorem 2 consists of two main steps and a final suboptimality gap characterization. **Step 1:** We decompose the suboptimality gap into the summation of the uncertainty metric of each step in Lemma 8. We note that the reward function $r_h$ here is from a

general class, not necessarily a linear function. **Step 2:** We provide an upper bound on the Bellman update error as shown in Lemma 10, where our main technical contribution lies in capturing the impact of the misspecification of the representation taken from upstream estimation on such an error. **Suboptimality gap characterization:** Based on the first two steps, we select uncertainty metric $\Gamma_h$ and obtain an instance-dependent suboptimality gap, which we further bound under the feature coverage assumption.

We provide details for the two main steps and the suboptimality gap characterization in Appendix C.1-Appendix C.3.

## C.1 Suboptimality Decomposition

In this step, we decompose the suboptimality gap into the summation of the uncertainty metric of each step in Lemma 8. We note that the reward function $r$ here is from a general class, not necessarily a linear function. To this end, we first provide the following lemma.

**Lemma 7.** *If* $|(\mathbb{B}_h\widehat{V}_{h+1} - \widehat{\mathbb{B}}_h\widehat{V}_{h+1})(s,a)| \leq \Gamma_h(s,a)$ *for all* $(h,s,a) \in [H] \times \mathcal{S} \times \mathcal{A}$, *then it holds that* $(\mathbb{B}_h\widehat{V}_{h+1})(s,a) \leq 1$, $\forall(h,s,a) \in [H] \times \mathcal{S} \times \mathcal{A}$.

*Proof.* It suffices to show

$$(\mathbb{B}_h\widehat{V}_{h+1})(s,a) \leq \max_{a_{h+1},\dots,a_H} \mathbb{E}\left[\sum_{h'=h}^{H} r_{h'}(s_{h'}, a_{h'})\Big| s_h = s, a_h = a\right].$$

We prove it by induction. For $h' = H$, since $\widehat{V}_{H+1} = 0$, we have

$$(\mathbb{B}_H\widehat{V}_{H+1})(s,a) = r_H(s,a) + (P_H^{(*,T+1)}\widehat{V}_{H+1})(s,a) = r_H(s,a).$$

Suppose for $h' = h+1$, $h \in [H-1]$, we have

$$(\mathbb{B}_{h+1}\widehat{V}_{h+2})(s,a) \leq \max_{a_{h+2},\dots,a_H} \mathbb{E}\left[\sum_{h'=h+1}^{H} r_{h'}(s_{h'}, a_{h'})\Big| s_{h+1} = s, a_{h+1} = a\right],$$

which is bounded in $[0,1]$ since $r_h \geq 0$, $\forall h$, and for any trajectory it holds that $\sum_{h=1}^{H} r_h \leq 1$.

Further note that

$$\widehat{Q}_{h+1}(s,a) = \min\{r_{h+1}(s,a) + \widehat{w}_{h+1}^\top\widehat{\phi}_{h+1}(s,a) - \Gamma_{h+1}(s,a), 1\}^+$$

$$\overset{(i)}{\leq} \min\{(\mathbb{B}_{h+1}\widehat{V}_{h+2})(s,a), 1\}^+$$

$$\overset{(ii)}{\leq} \max\{0, (\mathbb{B}_{h+1}\widehat{V}_{h+2})(s,a)\}$$

$$\overset{(iii)}{\leq} \max_{a_{h+2},\dots,a_H} \mathbb{E}\left[\sum_{h'=h+1}^{H} r_{h'}(s_{h'}, a_{h'})\Big| s_{h+1} = s, a_{h+1} = a\right],$$

where $(i)$ follows from the assumption $|(\mathbb{B}_h\widehat{V}_{h+1} - \widehat{\mathbb{B}}_h\widehat{V}_{h+1})(s,a)| \leq \Gamma_h(s,a)$, $(ii)$ follows because $(\mathbb{B}_{h+1}\widehat{V}_{h+2})(s,a) \leq 1$ by the induction hypothesis, and $(iii)$ follows from the fact that $r_h \geq 0$, $\forall h$.

Therefore, for $h' = h$, we have

$$(\mathbb{B}_h\widehat{V}_{h+1})(s,a) = r_h(s,a) + (P_h^{(*,T+1)}\widehat{V}_{h+1})(s,a)$$

$$= r_h(s,a) + \int_{s'} P_h^{(*,T+1)}(s'|s,a)\widehat{V}_{h+1}(s')ds'$$

$$\leq r_h(s,a) + \int_{s'} P_h^{(*,T+1)}(s'|s,a)\max_{a'}\widehat{Q}_{h+1}(s',a')ds'$$

$$\leq r_h(s,a) + \int_{s'} ds' P_h^{(*,T+1)}(s'|s,a)\max_{a',a_{h+2},\dots,a_H} \mathbb{E}\left[\sum_{h'=h+1}^{H} r_{h'}(s_{h'}, a_{h'})\Big| s_{h+1} = s', a_{h+1} = a'\right]$$

$$\leq r_h(s,a) + \int_{s'} ds' P_h^{(*,T+1)}(s'|s,a) \max_{a_{h+1},a_{h+2},\ldots,a_H} \mathbb{E}\left[\sum_{h'=h+1}^H r_{h'}(s_{h'},a_{h'})\middle| s_{h+1}=s'\right]$$

$$\leq \max_{a_{h+1},\ldots,a_H} \mathbb{E}\left[\sum_{h'=h}^H r_{h'}(s_{h'},a_{h'})\middle| s_h=s, a_h=a\right].$$

By backward induction from $H$ to 1, the proof is complete. $\qquad\square$

We denote the Bellman update error as $\zeta_h(s,a) = (\mathbb{B}_h\widehat{V}_{h+1})(s,a) - \widehat{Q}_h(s,a)$. The following lemma shows that it is sufficient to bound the pessimistic penalty.

**Lemma 8.** *Suppose with probability at least $1-\delta$, for all $(h,s,a) \in [H] \times \mathcal{S} \times \mathcal{A}$, it holds that $|(\mathbb{B}_h\widehat{V}_{h+1} - \widehat{\mathbb{B}}_h\widehat{V}_{h+1})(s,a)| \leq \Gamma_h(s,a)$. $\{\widehat{\pi}_h\}_{h=1}^H$ is the output of Algorithm 2. Then with probability at least $1-\delta$, for any $(h,s,a) \in [H] \times \mathcal{S} \times \mathcal{A}$, we have $0 \leq \zeta_h(s,a) \leq 2\Gamma_h(s,a)$. Moreover, it holds that for any policy $\pi$, with probability at least $1-\delta$,*

$$V_{P^{(*,T+1)},r}^\pi(s) - V_{P^{(*,T+1)},r}^{\widehat{\pi}}(s) \leq 2\sum_{h=1}^H \mathbb{E}_\pi[\Gamma_h(s_h,a_h)|s_1=s].$$

*Proof.* First, we show that $\zeta_h(s,a) \geq 0$. Recall

$$\widehat{Q}_h(\cdot,\cdot) = \min\{r_h(\cdot,\cdot) + \widehat{\phi}_h(\cdot,\cdot)\widehat{w}_h - \Gamma_h(\cdot,\cdot), 1\}^+.$$

If $r_h(s,a) + \widehat{\phi}_h(s,a)^\top \widehat{w}_h - \Gamma_h(s,a) \leq 0$, then $\widehat{Q}_h(s,a) = 0$, which implies that $\zeta_h(s,a) = (\mathbb{B}_h\widehat{V}_{h+1})(s,a) - \widehat{Q}_h(s,a) = (\mathbb{B}_h\widehat{V}_{h+1})(s,a) \geq 0$.

If $r_h(s,a) + \widehat{\phi}_h(s,a)^\top \widehat{w}_h - \Gamma_h(s,a) > 0$, then $\widehat{Q}_h \leq r_h(s,a) + \widehat{\phi}_h(s,a)^\top \widehat{w}_h - \Gamma_h(s,a) = (\widehat{\mathbb{B}}_h\widehat{V}_{h+1})(s,a) - \Gamma_h(s,a)$, which implies that

$$\zeta_h(s,a) = (\mathbb{B}_h\widehat{V}_{h+1})(s,a) - \widehat{Q}_h(s,a) \geq (\mathbb{B}_h\widehat{V}_{h+1})(s,a) - (\widehat{\mathbb{B}}_h\widehat{V}_{h+1})(s,a) + \Gamma_h(s,a) \geq 0.$$

We next show that $\zeta_h(s,a) \leq 2\Gamma_h(s,a)$. Note that

$$r_h(s,a) + \widehat{\phi}_h(s,a)^\top \widehat{w}_h - \Gamma_h(s,a) \overset{(i)}{=} (\widehat{\mathbb{B}}_h\widehat{V}_{h+1})(s,a) - \Gamma_h(s,a) \overset{(ii)}{\leq} (\mathbb{B}_h\widehat{V}_{h+1})(s,a) \overset{(iii)}{\leq} 1,$$

where $(i)$ follows from the definition of $(\widehat{\mathbb{B}}_h\widehat{V}_{h+1})(s,a)$, $(ii)$ follows because $|(\mathbb{B}_h\widehat{V}_{h+1} - \widehat{\mathbb{B}}_h\widehat{V}_{h+1})(s,a)| \leq \Gamma_h(s,a)$, and $(iii)$ follows from Lemma 7. Therefore,

$$\begin{aligned}
\widehat{Q}_h(s,a) &= \min\{r_h(s,a) + \widehat{\phi}_h(s,a)^\top \widehat{w}_h - \Gamma_h(s,a), 1\}^+ \\
&= \max\{r_h(s,a) + \widehat{\phi}_h(s,a)^\top \widehat{w}_h - \Gamma_h(s,a), 0\} \\
&\geq r_h(s,a) + \widehat{\phi}_h(s,a)^\top \widehat{w}_h - \Gamma_h(s,a) \\
&= (\widehat{\mathbb{B}}_h\widehat{V}_{h+1})(s,a) - \Gamma_h(s,a).
\end{aligned}$$

By the definition of $\zeta_h$, we have

$$\begin{aligned}
\zeta_h(s,a) &= (\mathbb{B}_h\widehat{V}_{h+1})(s,a) - \widehat{Q}_h(s,a) \\
&\leq (\mathbb{B}_h\widehat{V}_{h+1})(s,a) - (\widehat{\mathbb{B}}_h\widehat{V}_{h+1})(s,a) + \Gamma_h(s,a) \\
&\leq 2\Gamma_h(s,a).
\end{aligned}$$

Then we obtain

$$\begin{aligned}
&V_{P^{(*,T+1)},r}^\pi(s) - V_{P^{(*,T+1)},r}^{\widehat{\pi}}(s) \\
&\overset{(i)}{\leq} \sum_{h=1}^H \mathbb{E}_\pi[\zeta_h(s_h,a_h)|s_1=s] - \sum_{h=1}^H \mathbb{E}_{\widehat{\pi}}[\zeta_h(s_h,a_h)|s_1=s] \qquad (45) \\
&\overset{(ii)}{\leq} 2\sum_{h=1}^H \mathbb{E}_\pi[\Gamma_h(s_h,a_h)|s_1=s],
\end{aligned}$$

where $(i)$ follows from Lemma 16 and definition of $\widehat{\pi}$, and $(ii)$ follows because with probability at least $1-\delta$, for all $(h,s,a) \times [H] \times \mathcal{S} \times \mathcal{A}$, $0 \leq \zeta_h(s,a) \leq 2\Gamma_h(s,a)$ holds. $\qquad\square$

## C.2 Bounding Bellman update error $|(\mathbb{B}_h \widehat{V}_{h+1} - \widehat{\mathbb{B}}_h \widehat{V}_{h+1})(s,a)|$

In this step, we provide an upper bound on the Bellman update error as shown in Lemma 10, where the main effort lies in analyzing the impact of the misspecification of the representation taken from upstream estimation. To this end, we first introduce a concentration lemma that upper-bounds the stochastic noise in regression.

**Lemma 9.** *Under the setting of Theorem 2, if we choose* $\lambda = 1$, $\beta(\delta) = c_\beta \left( d\sqrt{\iota(\delta)} + \sqrt{dN_{\text{off}}}\xi_{\text{down}} + \sqrt{p \log N_{\text{off}}} \right)$ *where* $\iota(\delta) = \log(2pdHN_{\text{off}}\xi_{\text{down}}/\delta)$, *there exists an absolute constant* $\widetilde{C}$ *such that with probability at least* $1 - \delta$, *it holds that for all* $h \in [H]$.

$$\left\| \sum_{\tau=1}^{N_{\text{off}}} \widehat{\phi}_h(s_h^\tau, a_h^\tau) \left[ (P_h^{(*,T+1)}\widehat{V}_{h+1})(s_h^\tau, a_h^\tau) - \widehat{V}_{h+1}(s_{h+1}^\tau) \right] \right\|_{\Lambda_h^{-1}} \le \widetilde{C}\left[d\sqrt{\iota} + \sqrt{p \log N_{\text{off}}}\right].$$

*Proof.* Note that our reward functions here are selected from a general function class $\mathcal{R}$, not necessarily linear with respect to the feature function $\widehat{\phi}$. The value function $\widehat{V}_{h+1}$ has the form of

$$V(\cdot) := \min\left\{ \max_{a\in\mathcal{A}} w^T \phi(\cdot, a) + r(\cdot, a) + \beta\sqrt{\phi(\cdot, a)^\top \Lambda^{-1}\phi(\cdot, a)}, 1 \right\} \tag{46}$$

for some $w \in \mathbb{R}^d, r \in \mathcal{R}$ and positive definite matrix $\Lambda \succeq \lambda I_d$. Let $\mathcal{V}$ be the function class of $V(\cdot)$ and $\mathcal{N}_\varepsilon$ be the $\varepsilon$-covering number of $\mathcal{V}$ with respect to the distance $\text{dist}(V, V') = \sup_s |V(s) - V'(s)|$.

Note that for any $h \in [H], v \in \mathbb{R}^d$, we have

$$\left| v^\top \widehat{w}_h \right| = \left| v^\top \Lambda_h^{-1} \sum_{\tau=1}^{N_{\text{off}}} \widehat{\phi}_h(s_h^\tau, a_h^\tau)\widehat{V}_{h+1}(s_{h+1}^\tau) \right|$$

$$\le \sum_{\tau=1}^{N_{\text{off}}} \left| v^\top \Lambda_h^{-1} \widehat{\phi}_h(s_h^\tau, a_h^\tau) \right|$$

$$\le \sqrt{\left[\sum_{\tau=1}^{N_{\text{off}}} \|v\|_{\Lambda_h^{-1}}^2\right] \left[\sum_{\tau=1}^{N_{\text{off}}} \left\|\widehat{\phi}_h(s_h^\tau, a_h^\tau)\right\|_{\Lambda_h^{-1}}^2\right]}$$

$$\le \|v\|_2 \sqrt{dN_{\text{off}}/\lambda},$$

where the second inequality follows from Cauchy-Schwarz inequality and the last inequality follows from the fact that $\|v\|_{\Lambda_h^{-1}} = \left\|\Lambda_h^{-1}\right\|_{\text{op}}^{1/2} \cdot \|v\|_2 \le \sqrt{1/\lambda}\|v\|_2$, where $\|\cdot\|_{\text{op}}$ is the matrix operator norm and

$$\sum_{\tau=1}^{N_{\text{off}}} \left\|\widehat{\phi}_h(s_h^\tau, a_h^\tau)\right\|_{\Lambda_h^{-1}}^2 = \text{tr}\left(\Lambda_h^{-1}\sum_{\tau=1}^{N_{\text{off}}}\left(\widehat{\phi}_h(s_h^\tau, a_h^\tau)\widehat{\phi}_h(s_h^\tau, a_h^\tau)^\top\right)\right) \le \text{tr}(I_d) = d. \tag{47}$$

Thus $\|\widehat{w}_h\|_2 = \max_{v:\|v\|_2=1}\left|v^\top w_h^n\right| \le \sqrt{dN_{\text{off}}/\lambda}$.

Then using Lemma D.3, Lemma D.4 in Jin et al. (2020) and Lemma 18, we have for any fixed $\varepsilon > 0$ that with probability at least $1 - \delta$, for all $h \in [H]$:

$$\left\| \sum_{\tau=1}^{N_{\text{off}}} \widehat{\phi}_h(s_h^\tau, a_h^\tau)\left[\widehat{V}_{h+1}(s_{h+1}^\tau) - (P_h^{(*,T+1)}\widehat{V}_{h+1})(s_h^\tau, a_h^\tau)\right]\right\|_{\Lambda_h^{-1}}^2$$

$$\le 4\left[\frac{d}{2}\log\left(\frac{N_{\text{off}}+\lambda}{\lambda}\right) + \log\frac{H\mathcal{N}_\varepsilon}{\delta}\right] + \frac{8N_{\text{off}}^2\varepsilon^2}{\lambda}$$

$$\le 4\left[\frac{d}{2}\log\left(\frac{N_{\text{off}}+\lambda}{\lambda}\right) + d\log\left(1 + \frac{6\sqrt{dN_{\text{off}}}}{\varepsilon\sqrt{\lambda}}\right) + d^2\log\left(1 + 18\frac{d^{1/2}\beta^2}{\varepsilon^2\lambda}\right)\right.$$

$$\left. + \log\mathcal{N}_\mathcal{R}(\frac{\varepsilon}{3}) + \log\frac{H}{\delta}\right] + \frac{8N_{\text{off}}^2\varepsilon^2}{\lambda}$$

$$\overset{(i)}{\leq} 4\left[\frac{d}{2}\log\left(\frac{N_{\text{off}}+\lambda}{\lambda}\right) + d\log\left(1+\frac{6\sqrt{dN_{\text{off}}}}{\varepsilon\sqrt{\lambda}}\right) + d^2\log\left(1+18\frac{d^{1/2}\beta^2}{\varepsilon^2\lambda}\right)\right.$$
$$\left. +p\log\left(\frac{3}{\varepsilon}\right) + \log\frac{H}{\delta}\right] + \frac{8N_{\text{off}}^2\varepsilon^2}{\lambda}, \tag{48}$$

where $(i)$ follows from Assumption 5.

We select the $\varepsilon$-covering number parameters as $R = \sqrt{dN_{\text{off}}/\lambda}$, $B = \beta$ (see Lemma 18). Furthermore, we choose $\lambda = 1$, $\beta(\delta) = c_\beta\left(d\sqrt{\iota(\delta)} + \sqrt{dN_{\text{off}}}\xi_{\text{down}} + \sqrt{p\log N_{\text{off}}}\right)$, $\varepsilon = d/N_{\text{off}}$ where $\iota(\delta) = \log\left(2pdN_{\text{off}}H\max\{\xi_{\text{down}}, 1\}/\delta\right)$. Then Equation (48) can be bounded by

$$d\log\left(1+N_{\text{off}}\right) + d\log\left(1+d^{-1/2}N_{\text{off}}^{3/2}\right) + p\log(\frac{3N_{\text{off}}}{d}) + \log\frac{H}{\delta}$$
$$+ d^2\log(1+d^{-3/2}N_{\text{off}}^2[\beta(\delta)]^2)$$
$$\lesssim d\log\left(N_{\text{off}}\right) + d\log\left(d^{-1/2}N_{\text{off}}^{3/2}\right) + p\log(\frac{3N_{\text{off}}}{d}) + \log\frac{H}{\delta}$$
$$+ d^2\log\left(d^{1/2}\iota N_{\text{off}}^3\xi_{\text{down}}^2 p^2\right)$$
$$\lesssim d^2\iota + p\log N_{\text{off}},$$

where the notation $f(x) \lesssim g(x)$ denotes that there exists a universal positive constant c (independent of $x$) such that $f(x) \leq cg(x)$.

Therefore,

$$\left\|\sum_{\tau=1}^{N_{\text{off}}} \widehat{\phi}_h(s_h^\tau, a_h^\tau)\left[\widehat{V}_{h+1}(s_{h+1}^\tau) - (P_h^{(*,T+1)}\widehat{V}_{h+1})(s_h^\tau, a_h^\tau)\right]\right\|_{\Lambda_h^{-1}} \lesssim d\sqrt{\iota} + \sqrt{p\log N_{\text{off}}},$$

where we use $\sqrt{x+y} \leq \sqrt{x} + \sqrt{y}$ for all $x, y \geq 0$. $\qquad\square$

The following lemma provides our main result which upper-bounds the Bellman update error $|(\mathbb{B}_h\widehat{V}_{h+1} - \widehat{\mathbb{B}}_h\widehat{V}_{h+1})(s,a)|$.

**Lemma 10.** *Under the setting of Theorem 2, fix $\delta \in (0,1)$. If we choose $\lambda = 1$, $\beta(\delta) = c_\beta\left(d\sqrt{\iota(\delta)} + \sqrt{dN_{\text{off}}}\xi_{\text{down}} + \sqrt{p\log N_{\text{off}}}\right)$, where $\iota(\delta) = \log\left(2pdHN_{\text{off}}\xi_{\text{down}}/\delta\right)$, then with probability at least $1 - \delta$, the following bound holds:*

$$\left|(\mathbb{B}_h\widehat{V}_{h+1} - \widehat{\mathbb{B}}_h\widehat{V}_{h+1})(s,a)\right| \leq \beta(\delta)\left\|\widehat{\phi}_h(s,a)\right\|_{\Lambda_h^{-1}} + \xi_{\text{down}}. \tag{49}$$

*Proof of Lemma 10.* For $h \in [H]$, define $\widehat{w}_h^* = \int_{s'}\widehat{\mu}^*(s')\widehat{V}_{h+1}(s')ds'$. It is easy to verify that $\widehat{\phi}_h(s,a)^\top\widehat{w}_h^* = (\overline{P}_h\widehat{V}_{h+1})(s,a)$ and $(\overline{B}_h\widehat{V}_{h+1})(s,a) = r_h(s,a) + \widehat{\phi}_h(s,a)^\top\widehat{w}_h^*$. Then we have

$$\left|(\mathbb{B}_h\widehat{V}_{h+1} - \widehat{\mathbb{B}}_h\widehat{V}_{h+1})(s,a)\right|$$
$$= \left|(\mathbb{B}_h\widehat{V}_{h+1} - \overline{\mathbb{B}}_h\widehat{V}_{h+1} + \overline{\mathbb{B}}_h\widehat{V}_{h+1} - \widehat{\mathbb{B}}_h\widehat{V}_{h+1})(s,a)\right|$$
$$\leq \left|(P_h^{(*,T+1)}\widehat{V}_{h+1})(s,a) - (\overline{P}_h\widehat{V}_{h+1})(s,a)\right| + \left|\widehat{\phi}_h(s,a)^\top(\widehat{w}_h^* - \widehat{w}_h)\right|$$
$$\leq \xi_{\text{down}} + \left|\widehat{\phi}_h(s,a)^\top(\widehat{w}_h^* - \widehat{w}_h)\right|, \tag{50}$$

where the last inequality follows because $\left|\widehat{V}_{h+1}(s)\right| \leq 1$ for all $s \in \mathcal{S}$ and from Lemma 1.

Recall that $\widehat{w}_h = \Lambda_h^{-1}\left(\sum_{\tau=1}^{N_{\text{off}}}\widehat{\phi}_h(s_h^\tau, a_h^\tau)\widehat{V}_{h+1}(s_{h+1}^\tau)\right)$, where $\Lambda_h = \sum_{\tau=1}^{N_{\text{off}}}\widehat{\phi}(s_h, a_h)\widehat{\phi}_h(s_h, a_h)^\top + \lambda I_d$. Then the second term in Equation (50) can be further decomposed as

$$\widehat{\phi}_h(s,a)^\top(\widehat{w}_h^* - \widehat{w}_h)$$

$$= \widehat{\phi}_h(s,a)^\top \Lambda_h^{-1} \left\{ \left( \sum_{\tau=1}^{N_{\text{off}}} \widehat{\phi}(s_h, a_h) \widehat{\phi}_h(s_h, a_h)^\top + \lambda I_d \right) \widehat{w}_h^* - \left( \sum_{\tau=1}^{N_{\text{off}}} \widehat{\phi}_h(s_h^\tau, a_h^\tau) \widehat{V}_{h+1}(s_{h+1}^\tau) \right) \right\}$$

$$= \underbrace{\lambda \widehat{\phi}_h(s,a)^\top \Lambda_h^{-1} \widehat{w}_h^*}_{(\text{I})} + \underbrace{\widehat{\phi}_h(s,a)^\top \Lambda_h^{-1} \left\{ \sum_{\tau=1}^{N_{\text{off}}} \widehat{\phi}_h(s_h^\tau, a_h^\tau) \left[ (P_h^{(*,T+1)} \widehat{V}_{h+1})(s_h^\tau, a_h^\tau) - \widehat{V}_{h+1}(s_{h+1}^\tau) \right] \right\}}_{(\text{II})}$$

$$+ \underbrace{\widehat{\phi}_h(s,a)^\top \Lambda_h^{-1} \left\{ \sum_{\tau=1}^{N_{\text{off}}} \widehat{\phi}_h(s_h^\tau, a_h^\tau) \left[ (\overline{P}_h \widehat{V}_{h+1} - P_h^{(*,T+1)} \widehat{V}_{h+1})(s_h^\tau, a_h^\tau) \right] \right\}}_{(\text{III})}. \tag{51}$$

We next bound the three terms in the above equation individually.

Term (I) is upper-bounded as

$$|(\text{I})| \leq \lambda \|w_h\|_{\Lambda_h^{-1}} \cdot \left\| \widehat{\phi}_h(s,a) \right\|_{\Lambda_h^{-1}} \leq \sqrt{d\lambda} \left\| \widehat{\phi}_h(s,a) \right\|_{\Lambda_h^{-1}}, \tag{52}$$

where the first inequality follows from Cauchy-Schwarz inequality and the second inequality follows from the fact that $\|w_h\|_{\Lambda_h^{-1}} = \left\| \Lambda_h^{-1} \right\|_{\text{op}}^{1/2} \cdot \|w_h\|_2 \leq \sqrt{d/\lambda}$.

Term (II) is upper-bounded as

$$|((\text{II}))| \leq \left\| \widehat{\phi}_h(s,a) \right\|_{\Lambda_h^{-1}} \left\| \sum_{\tau=1}^{\tau} \widehat{\phi}_h(s_h^\tau, a_h^\tau) \left[ (P_h^{(*,T+1)} \widehat{V}_{h+1})(s_h^\tau, a_h^\tau) - \widehat{V}_{h+1}(s_{h+1}^\tau) \right] \right\|_{\Lambda_h^{-1}}$$

$$\leq \widetilde{C} \left[ d\sqrt{\iota} + \sqrt{p \log N_{\text{off}}} \right] \left\| \widehat{\phi}_h(s,a) \right\|_{\Lambda_h^{-1}}, \tag{53}$$

where the first inequality follows from Cauchy-Schwarz inequality and the second inequality follows from Lemma 9.

Term (III) is upper-bounded as

$$|(\text{III})| \leq \left| \widehat{\phi}_h(s,a)^\top \Lambda_h^{-1} \left( \sum_{\tau=1}^{N_{\text{off}}} \widehat{\phi}_h(s_h^\tau, a_h^\tau) \right) \right| \cdot \xi_{\text{down}}$$

$$\overset{(i)}{\leq} \sum_{\tau=1}^{N_{\text{off}}} \left| \widehat{\phi}_h(s,a)^\top \Lambda_h^{-1} \widehat{\phi}_h(s_h^\tau, a_h^\tau) \right| \cdot \xi_{\text{down}}$$

$$\overset{(ii)}{\leq} \sqrt{ \left( \sum_{\tau=1}^{N_{\text{off}}} \left\| \widehat{\phi}_h(s,a) \right\|_{\Lambda_h^{-1}}^2 \right) } \sqrt{ \left( \sum_{\tau=1}^{N_{\text{off}}} \left\| \widehat{\phi}_h(s_h^\tau, a_h^\tau) \right\|_{\Lambda_h^{-1}}^2 \right) } \cdot \xi_{\text{down}}$$

$$\overset{(iii)}{\leq} \xi_{\text{down}} \cdot \sqrt{dN_{\text{off}}} \left\| \widehat{\phi}_h(s,a) \right\|_{\Lambda_h^{-1}}, \tag{54}$$

where $(i)$ follows because $\left| \widehat{V}_{h+1}(s) \right| \leq 1$ for all $s \in \mathcal{S}$ and from Lemma 1, $(ii)$ follows from Cauchy-Schwarz inequality, and $(iii)$ follows from Equation (47).

Choosing $\lambda = 1$, $\beta(\delta) = c_\beta \left( d\sqrt{\iota(\delta)} + \sqrt{dN_{\text{off}}} \xi_{\text{down}} + \sqrt{p \log N_{\text{off}}} \right)$, where $\iota(\delta) = \log \left( 2pdHN_{\text{off}} \max\{\xi_{\text{down}}, 1\}/\delta \right)$, and combining Equations (50) to (54), we conclude that with probability at least $1 - \delta$, for any $(s,a,h) \in \mathcal{S} \times \mathcal{A} \times [H]$, the following bound holds:

$$\left| (\mathbb{B}_h \widehat{V}_{h+1} - \widehat{\mathbb{B}}_h \widehat{V}_{h+1})(s,a) \right| \leq \beta(\delta) \left\| \widehat{\phi}_h(s,a) \right\|_{\Lambda_h^{-1}} + \xi_{\text{down}}. \tag{55}$$

$\square$

## C.3 Suboptimality gap characterization: proof of Theorem 2

Based on the previous lemmas, we establish the suboptimality gap.

In Lemma 8, let $\Gamma_h = \beta \left\| \widehat{\phi}_h(s,a) \right\|_{\Lambda_h^{-1}} + \xi_{\text{down}}$. Then Lemma 8 implies that with probability at least $1 - \delta$,

$$V^*_{P^{(*,T+1)},r} - V^{\widehat{\pi}}_{P^{(*,T+1)},r}$$

$$\leq 2 \sum_{h=1}^{H} \mathbb{E}_{\pi^*} [\Gamma_h(s_h, a_h)|s_1 = \widetilde{s}_1]$$

$$\leq 2H\xi_{\text{down}} + 2\beta \sum_{h=1}^{H} \mathbb{E}_{\pi^*} \left[ \left\| \widehat{\phi}_h(s_h, a_h) \right\|_{\Lambda_h^{-1}} \Big| s_1 = s \right]. \tag{56}$$

We next show the second part of Theorem 2, which is the suboptimality bound under the feature coverage assumption (see Assumption 6). We first note that Appendix B.4 in Jin et al. (2021) shows that if $N_{\text{off}} \geq 40/\kappa_\rho \cdot \log(4dH/\delta)$, then with probability at least $1 - \delta/2$, for all $(s, a, h) \in \mathcal{S} \times \mathcal{A} \times [H]$,

$$\left\| \widehat{\phi}_h(s, a) \right\|_{\Lambda_h^{-1}} \leq \sqrt{\frac{2}{\kappa_\rho}} \cdot \frac{1}{\sqrt{N_{\text{off}}}}.$$

By selecting $\beta(\delta/2) = c_\beta \left( d\sqrt{\iota(\delta/2)} + \sqrt{dN_{\text{off}}}\xi_{\text{down}} + \sqrt{p \log N_{\text{off}}} \right)$ (see Lemma 9), with probability at least $1 - \delta/2$, we have

$$V^*_{P^{(*,T+1)},r} - V^{\widehat{\pi}}_{P^{(*,T+1)},r} \leq 2H\xi_{\text{down}} + 2\beta \sum_{h=1}^{H} \mathbb{E}_{\pi^*} \left[ \left\| \widehat{\phi}_h(s_h, a_h) \right\|_{\Lambda_h^{-1}} \Big| s_1 = s \right].$$

By a union bound, we have with probability at least $1 - \delta$, the following bound holds:

$$V^*_{P^{(*,T+1)},r} - V^{\widehat{\pi}}_{P^{(*,T+1)},r}$$

$$\leq 2H \left( \xi_{\text{down}} + \beta(\delta/2) \cdot \sqrt{\frac{2}{\kappa_\rho}} \cdot \frac{1}{\sqrt{N_{\text{off}}}} \right)$$

$$= O \left( \kappa_\rho^{-1/2} H d^{1/2} \xi_{\text{down}} + \kappa_\rho^{-1/2} H d \sqrt{\frac{\log\left(pdHN_{\text{off}}\xi_{\text{down}}/\delta\right)}{N_{\text{off}}}} + \kappa_\rho^{-1/2} H \sqrt{\frac{p \log N_{\text{off}}}{N_{\text{off}}}} \right).$$

## D Algorithm 3 and Proof of Theorem 3

Recall $\xi_{\text{down}} = \xi + \frac{C_L C_R T v \epsilon_u}{\kappa_u}$ and for any $h \in [H]$, we define $P_h^{(*,T+1)}(\cdot|s,a) = \langle \phi_h^*(s,a), \mu_h^{(*,T+1)}(\cdot) \rangle, \overline{P}_h(\cdot|s,a) = \langle \widehat{\phi}_h(s,a), \widehat{\mu}_h^*(\cdot) \rangle, \widehat{P}_h(\cdot|s,a) = \langle \widehat{\phi}_h(s,a), \widehat{\mu}_h(\cdot) \rangle$. For any function $f : \mathcal{S} \mapsto \mathbb{R}$ and $h \in [H]$, define

$$P_h^{(*,T+1)} f(s,a) = \int_{s'} \langle \phi_h^*(s,a), \mu_h^{(*,T+1)}(s') \rangle f(s')ds',$$

$$(\overline{P}_h f)(s,a) = \int_{s'} \widehat{\phi}(s,a)\widehat{\mu}^*(s')f(s')ds',$$

$$(\widehat{P}_h f)(s,a) = \int_{s'} \widehat{\phi}(s,a)\widehat{\mu}(s')f(s')ds'.$$

Throughout this section, denote $\pi^n$ as the greedy policy induced by $\{Q_h^n\}_{h=1}^{H}$, and note that $\lambda_h^n, w_h^n, Q_h^n, V_h^n$ are defined in Algorithm 3. We further remark that the expectation (for example: $V_h^\pi(s)$) is taken with respect to the transition kernel of the target task, i.e., $P^{(*,T+1)}$.

---

**Algorithm 3** DONRL (**D**ownstream **ON**line **RL**)

---

1: **Input:** Feature $\widehat{\phi}$, parameters $\lambda$, $\beta_n$.
2: **for** $n = 1, \ldots, N$ **do**
3:    Receive the initial state $s_1^n = s_1$.
4:    **for** $h = H, \ldots, 1$ **do**
5:       $\Lambda_h^n = \sum_{\tau=1}^{n-1} \widehat{\phi}_h(s_h^\tau, a_h^\tau) \widehat{\phi}_h(s_h^\tau, a_h^\tau)^\top + \lambda I_d$.
6:       $w_h^n = (\Lambda_h^n)^{-1} \sum_{\tau=1}^{n-1} \widehat{\phi}_h(s_h^\tau, a_h^\tau) V_{h+1}^n(s_{h+1}^\tau)$.
7:       $Q_h^n(\cdot, \cdot) = \min \left\{ r_h(\cdot, \cdot) + \widehat{\phi}_h(\cdot, \cdot)^\top w_h^n + \beta_n \left\| \widehat{\phi}_h(\cdot, \cdot) \right\|_{(\Lambda_h^n)^{-1}}, 1 \right\}$,
         $V_h^n(\cdot) = \max_a Q_h^n(\cdot, a)$.
8:    Let $\pi^n$ be the greedy policy induced by $\{Q_h^n\}_{h=1}^H$, i.e., $\pi_h^n(\cdot) = \arg\max_{a \in \mathcal{A}} Q_h^n(\cdot, a)$
9:    **for** $h = 1, \ldots, H$ **do**
10:       Take action $a_h^n = \pi^n(s_h^n)$, and observe $s_{h+1}^n$.
11: **Output:** $\pi^1, \ldots, \pi^n$.

---

**Proof Overview:** The proof of Theorem 3 consists of two main steps and a final suboptimality gap analysis. **Step 1:** We bound the difference between the estimated action value function $r_h(s, a) + \langle \widehat{\phi}_h(s, a), w_h^n \rangle$ in Algorithm 3 and the true action value function $Q_h^\pi(s, a)$ under a certain policy $\pi$ recursively as shown in Lemma 12. **Step 2:** We prove the estimated action value function $Q_h^n$ in Algorithm 3 is near-optimistic with respect to the optimal true action value function over steps as shown in Lemma 13. Our main technical contribution lies in capturing the impact of the misspecification of the representation taken from upstream learning on these two steps. **Suboptimality gap analysis:** Based on the first two steps, we first decompose the value function difference recursively, and then obtain a final suboptimality gap.

### D.1 Bounding the action value function difference

Following the proof similar to that for Lemma 9, we introduce the concentration lemma for online RL that upper-bounds the stochastic noise in regression.

**Lemma 11.** *Fix $\delta \in (0, 1)$. Under the setting of Theorem 3, we choose $\lambda = 1$, $\beta_n = c_\beta \left( d\sqrt{\iota_n} + \sqrt{dn}\xi_{\text{down}} + \sqrt{p \log n} \right)$, where $\iota_n = \log \left( 2pdHn \max\{\xi_{\text{down}}, 1\}/\delta \right)$. Then, there exists an absolute constant $\widetilde{C}$ such that with probability at least $1 - \delta/2$, the following inequality holds for any $n \in [N_{on}], h \in [H]$:*

$$\left\| \sum_{\tau=1}^{n-1} \widehat{\phi}_h(s_h^\tau, a_h^\tau) \left[ (P_h^{(*,T+1)} V_{h+1}^n)(s_h^\tau, a_h^\tau) - V_{h+1}^n(s_{h+1}^\tau) \right] \right\|_{\Lambda_h^{-1}} \leq \widetilde{C} \left[ d\sqrt{\iota_n} + \sqrt{p \log n} \right].$$

**Lemma 12.** *Fix $\delta \in (0, 1)$. There exists a constant $c_\beta$ such that for $\beta_n = c_\beta \left( d\sqrt{\iota_n} + \sqrt{nd}\xi_{\text{down}} + \sqrt{p \log n} \right)$ where $\iota_n = \log \left( 2dnH\xi_{\text{down}}/\delta \right)$, and for any policy $\pi$, with probability at least $1 - \delta/2$, we have for any $s \in \mathcal{S}, a \in \mathcal{A}, h \in [H], n \in [N_{\text{on}}]$ that:*

$$\left( r_h(s, a) + \langle \widehat{\phi}_h(s, a), w_h^n \rangle \right) - Q_h^\pi(s, a) = P_h^{(*,T+1)}(V_{h+1}^n - V_{h+1}^\pi)(s, a) + \Delta_h^k(s, a),$$

*for some $\Delta_h^n(s, a)$ that satisfies $\|\Delta_h^k(s, a)\| \leq \beta_n \left\| \widehat{\phi}_h(s, a) \right\|_{(\Lambda_h^n)^{-1}} + 2\xi_{\text{down}}$.*

*Proof.* For policy $\pi$, define $w_h^\pi = \int V_{h+1}^\pi(s') \widehat{\mu}^*(s') ds'$. Hence, $\langle \widehat{\phi}_h(s, a), w_h^\pi \rangle = \overline{P}_h V_{h+1}^\pi(s, a)$ and $\|w_h^\pi\|_2 \leq C_L \sqrt{d}$ by Lemma 1. These facts further yield that for any $s \in \mathcal{S}, a \in \mathcal{A}, h \in [H]$:

$$\left| Q_h^\pi(s, a) - \left( r_h(s, a) + \langle \widehat{\phi}_h(s, a), w_h^\pi \rangle \right) \right| = \left| P_h^{(*,T+1)} V_{h+1}^\pi(s, a) - \overline{P}_h V_{h+1}^\pi(s, a) \right| \leq \xi_{\text{down}},$$

where the last inequality follows from Lemma 1.

Then, we further derive

$$
\left( r_h(s,a) + \langle \widehat{\phi}_h(s,a), w_h^n \rangle \right) - Q_h^\pi(s,a)
$$

$$
= \left( r_h(s,a) + \langle \widehat{\phi}_h(s,a), w_h^n \rangle \right) - \left( r_h(s,a) + \langle \widehat{\phi}_h(s,a), w_h^\pi \rangle \right) + \left( r_h(s,a) + \langle \widehat{\phi}_h(s,a), w_h^\pi \rangle \right) - Q_h^\pi(s,a)
$$

$$
\leq \langle \widehat{\phi}_h(s,a), w_h^n \rangle - \langle \widehat{\phi}_h(s,a), w_h^\pi \rangle + \left| \left( r_h(s,a) + \langle \widehat{\phi}_h(s,a), w_h^\pi \rangle \right) - Q_h^\pi(s,a) \right|. \tag{57}
$$

The first term can be bounded by

$$
\langle \widehat{\phi}_h(s,a), w_h^n \rangle - \langle \widehat{\phi}_h(s,a), w_h^\pi \rangle
$$

$$
= \widehat{\phi}_h(s,a)^\top (\Lambda_h^n)^{-1} \sum_{\tau=1}^{n-1} \widehat{\phi}(s_h^\tau, a_h^\tau) V_{h+1}^n(s_{h+1}^\tau) - \widehat{\phi}_h(s,a)^\top w_h^\pi
$$

$$
= \widehat{\phi}_h(s,a)^\top (\Lambda_h^n)^{-1} \left\{ \sum_{\tau=1}^{n-1} \widehat{\phi}(s_h^\tau, a_h^\tau) V_{h+1}^n(s_{h+1}^\tau) - \lambda w_h^\pi - \sum_{\tau=1}^{n-1} \widehat{\phi}(s_h^\tau, a_h^\tau) \overline{P}_h V_{h+1}^\pi \right\}
$$

$$
= \underbrace{-\lambda \widehat{\phi}_h(s,a)^\top (\Lambda_h^n)^{-1} w_h^\pi}_{(I)} + \underbrace{\widehat{\phi}_h(s,a)^\top (\Lambda_h^n)^{-1} \left\{ \sum_{\tau=1}^{n-1} \widehat{\phi}(s_h^\tau, a_h^\tau) \left[ V_{h+1}^n(s_{h+1}^\tau) - P_h^{(*,T+1)} V_{h+1}^n(s_h^\tau, a_h^\tau) \right] \right\}}_{(II)}
$$

$$
+ \underbrace{\widehat{\phi}_h(s,a)^\top (\Lambda_h^n)^{-1} \left\{ \sum_{\tau=1}^{n-1} \widehat{\phi}(s_h^\tau, a_h^\tau) \overline{P}_h \left( V_{h+1}^n - V_{h+1}^\pi \right) (s_h^\tau, a_h^\tau) \right\}}_{(III)}
$$

$$
+ \underbrace{\widehat{\phi}_h(s,a)^\top (\Lambda_h^n)^{-1} \left\{ \sum_{\tau=1}^{n-1} \widehat{\phi}(s_h^\tau, a_h^\tau) \left( P_h^{(*,T+1)} - \overline{P}_h \right) V_{h+1}^n(s_h^\tau, a_h^\tau) \right\}}_{(IV)}.
$$

We next bound the above four terms individually.

For $(I)$, we derive the following bound:

$$
|(I)| \leq \left\| \widehat{\phi}_h(s,a) \right\|_{(\Lambda_h^n)^{-1}} \| \lambda w_h^\pi \|_{(\Lambda_h^n)^{-1}} \leq \sqrt{\lambda} \| w_h^\pi \|_2 \left\| \widehat{\phi}_h(s,a) \right\|_{(\Lambda_h^n)^{-1}} = C_L \sqrt{\lambda d} \left\| \widehat{\phi}_h(s,a) \right\|_{(\Lambda_h^n)^{-1}}. \tag{58}
$$

For $(II)$, by Lemma 11, we have

$$
|(II)|
$$

$$
\leq \left\| \widehat{\phi}_h(s,a) \right\|_{(\Lambda_h^n)^{-1}} \left\| \sum_{\tau=1}^{n-1} \widehat{\phi}_h(s_h^\tau, a_h^\tau) \left[ V_{h+1}^n(s_{h+1}^\tau) - P_h^{(*,T+1)} V_{h+1}^n(s_h^\tau, a_h^\tau) \right] \right\|_{(\Lambda_h^n)^{-1}}
$$

$$
\leq \left( \widetilde{C} d \sqrt{\iota_n} + \sqrt{p \log n} \right) \left\| \widehat{\phi}_h(s,a) \right\|_{(\Lambda_h^n)^{-1}}.
$$

For $(III)$, we have

$$
|(III)|
$$

$$
\leq \left| \widehat{\phi}_h(s,a)^\top (\Lambda_h^n)^{-1} \left\{ \sum_{\tau=1}^{n-1} \widehat{\phi}(s_h^\tau, a_h^\tau) \widehat{\phi}(s_h^\tau, a_h^\tau)^\top \int \left( V_{h+1}^n - V_{h+1}^\pi \right)(s') \widehat{\mu}_h^*(s') ds' \right\} \right|
$$

$$
\leq \left| \widehat{\phi}_h(s,a)^\top (\Lambda_h^n)^{-1} (\Lambda_h^n - \lambda I) \int \left( V_{h+1}^n - V_{h+1}^\pi \right)(s') \widehat{\mu}_h^*(s') ds' \right|
$$

$$
= \underbrace{\left| \widehat{\phi}_h(s,a)^\top \int \left( V_{h+1}^n - V_{h+1}^\pi \right)(s') \widehat{\mu}_h^*(s') ds' \right|}_{(a)} + \underbrace{\left| \lambda \widehat{\phi}_h(s,a)^\top (\Lambda_h^n)^{-1} \int \left( V_{h+1}^n - V_{h+1}^\pi \right)(s') \widehat{\mu}_h^*(s') ds' \right|}_{(b)}.
$$

For term $(a)$, we have

$$
\begin{aligned}
(a) &= \widehat{\phi}_h(s,a)^\top \int \left(V_{h+1}^n - V_{h+1}^\pi\right)(s')\widehat{\mu}_h^*(s')ds' \\
&= \overline{P}_h\left(V_{h+1}^n - V_{h+1}^\pi\right)(s,a) \\
&\le P^{(*,T+1)}\left(V_{h+1}^n - V_{h+1}^\pi\right)(s,a) + \xi_{\text{down}},
\end{aligned}
$$

where the last inequality follows from Lemma 1.

For term (b), similarly to Equation (58), we have

$$
(b) \le C_L\sqrt{\lambda d}\left\|\widehat{\phi}_h(s,a)\right\|_{(\Lambda_h^n)^{-1}}.
$$

For $(IV)$, we derive

$$
\begin{aligned}
|(IV)| &\le \left|\widehat{\phi}_h(s,a)^\top(\Lambda_h^n)^{-1}\left\{\sum_{\tau=1}^{n-1}\widehat{\phi}_h(s_h^\tau, a_h^\tau)\right\}\right|\xi_{\text{down}} \\
&\le \sum_{\tau=1}^{n-1}\left|\widehat{\phi}_h(s,a)^\top(\Lambda_h^n)^{-1}\widehat{\phi}_h(s_h^\tau, a_h^\tau)\right|\xi_{\text{down}} \\
&\overset{(i)}{\le} \sqrt{\left[\sum_{\tau=1}^{n-1}\left\|\widehat{\phi}_h(s,a)\right\|_{(\Lambda_h^n)^{-1}}^2\right]\left[\sum_{\tau=1}^{n-1}\left\|\widehat{\phi}_h(s_h^\tau, a_h^\tau)\right\|_{(\Lambda_h^n)^{-1}}^2\right]}\,\xi_{\text{down}} \\
&\overset{(ii)}{\le} \xi_{\text{down}}\sqrt{dn}\left\|\widehat{\phi}_h(s,a)\right\|_{(\Lambda_h^n)^{-1}},
\end{aligned}
$$

where $(i)$ follows from Cauchy-Schwarz inequality and $(ii)$ follows because

$$
\sum_{\tau=1}^{n-1}\left\|\widehat{\phi}_h(s_h^\tau, a_h^\tau)\right\|_{(\Lambda_h^n)^{-1}}^2 = \text{tr}\left((\Lambda_h^n)^{-1}\sum_{\tau=1}^{n-1}\left(\widehat{\phi}_h(s_h^\tau, a_h^\tau)\widehat{\phi}_h(s_h^\tau, a_h^\tau)^\top\right)\right) \le \text{tr}(I_d) = d.
$$

Substituting the bounds on $(I), (II), (III), (IV)$ into Equation (57), we finish the proof. $\square$

### D.2 Proving optimism of value function

**Lemma 13.** *Under the setting of Theorem 3, with probability at least $1 - \delta/2$, for any $s \in \mathcal{S}, a \in \mathcal{A}, h \in [H], n \in [N_{\text{on}}]$, we have*

$$
Q_h^n(s,a) \ge Q_h^*(s,a) - 2(H - h + 1)\xi_{\text{down}}. \tag{59}
$$

*Proof.* We prove this lemma by induction. First, for step $H$, by Lemma 12, we have

$$
\begin{aligned}
&\left|\left(r_H(s,a) + \langle\widehat{\phi}_H(s,a), w_H^n\rangle\right) - Q_H^\pi(s,a)\right| \\
&= \left|P_H^{(*,T+1)}(V_{H+1}^n - V_{H+1}^\pi)(s,a) + \Delta_H^k(s,a)\right| \\
&\le \beta_n\left\|\widehat{\phi}_H(s,a)\right\|_{(\Lambda_H^n)^{-1}} + 2\xi_{\text{down}}.
\end{aligned}
$$

Thus,

$$
\begin{aligned}
Q_H^n(s,a) &= \min\left\{r_H(\cdot,\cdot) + \langle\widehat{\phi}(\cdot,\cdot), w_H^n\rangle + \beta_n\left\|\widehat{\phi}(\cdot,\cdot)\right\|_{(\Lambda_H^n)^{-1}}, 1\right\} \\
&\ge Q_H^*(s,a) - 2\xi_{\text{down}}.
\end{aligned}
$$

Suppose Equation (59) holds for step $h + 1$. Then for step $h$, following from Lemma 12, we have:

$$
\left(r_h(s,a) + \langle\widehat{\phi}_h(s,a), w_h^n\rangle\right) - Q_h^*(s,a)
$$

$$= \Delta_h^n(s, a) + P_h^{(*,T+1)}(V_{h+1}^n - V_{h+1}^*)(s, a)$$

$$\geq -\beta_n \left\| \widehat{\phi}_h(s, a) \right\|_{(\Lambda_h^n)^{-1}} - 2\xi_{\text{down}} - 2(H - h)\xi_{\text{down}}$$

$$\geq -\beta_n \left\| \widehat{\phi}_h(s, a) \right\|_{(\Lambda_h^n)^{-1}} - 2(H - h + 1)\xi_{\text{down}}.$$

Therefore,

$$Q_h^n(s, a) = \min \left\{ r_h(\cdot, \cdot) + \langle \widehat{\phi}(\cdot, \cdot), w_h^n \rangle + \beta_n \left\| \widehat{\phi}(\cdot, \cdot) \right\|_{(\Lambda_h^n)^{-1}}, 1 \right\}$$

$$\geq Q_h^*(s, a) - 2(H - h + 1)\xi_{\text{down}},$$

which finishes the proof. $\qquad\square$

### D.3 Suboptimality gap: proof of Theorem 3

Before proving Theorem 3, we introduce the following lemma to decompose the value function difference recursively.

**Lemma 14.** *Fix $\delta \in (0, 1)$. Let $\delta_h^n = V_h^n(s_h^n) - V_h^{\pi^n}(s_h^n)$ and $\xi_{h+1}^n = \mathbb{E}\left[\delta_{h+1}^n | s_h^n, a_h^n\right] - \delta_{h+1}^n$. Then, with probability at least $1 - \delta/2$, for $h \in [H], n \in [N_{\text{on}}]$:*

$$\delta_h^n \leq \delta_{h+1}^n + \xi_{h+1}^n + 2\beta_n \left\| \widehat{\phi}_h(s_h^n, a_h^n) \right\|_{(\Lambda_h^n)^{-1}} + 2\xi_{\text{down}}.$$

*Proof.* By Lemma 12, with probability at least $1 - \delta/2$, for any $s \in \mathcal{S}, a \in \mathcal{A}, h \in [H], n \in [N_{\text{on}}]$, we have

$$Q_h^n(s, a) - Q_h^{\pi^n}(s, a)$$
$$= \Delta_h^n(s, a) + P_h^{(*,T+1)}(V_{h+1}^n - V_{h+1}^{\pi^n})(s, a)$$
$$\leq \beta_n \left\| \widehat{\phi}_h(s, a) \right\|_{(\Lambda_h^n)^{-1}} + 2\xi_{\text{down}} + P_h^{(*,T+1)}(V_{h+1}^n - V_{h+1}^{\pi^n})(s, a).$$

By the definition of $\pi^n$ in Algorithm 3, we have $\pi^n(s_h^n) = a_h^n = \arg\max_{a \in \mathcal{A}} Q_h^n(s_h, a)$. Then $Q_h^n(s_h^n, a_h^n) - Q_h^{\pi^n}(s_h^n, a_h^n) = V_h^n(s_h^n) - V_h^{\pi^n}(s_h^n) = \delta_h^n$. Thus,

$$\delta_h^n \leq \delta_{h+1}^n + \xi_{h+1}^n + 2\beta_n \left\| \widehat{\phi}_h(s_h^n, a_h^n) \right\|_{(\Lambda_h^n)^{-1}} + 2\xi_{\text{down}}.$$

$\qquad\square$

Finally, we combine Lemmas 11 to 14 to prove Theorem 3.

*Proof of Theorem 3.* The regret can be bounded by

$$\sum_{n=1}^{N_{\text{on}}} \left( V_{P^{(*,T+1)},r}^* - V_{P^{(*,T+1)},r}^{\pi^n} \right)$$

$$\overset{(i)}{\leq} \sum_{n=1}^{N_{\text{on}}} \left\{ \left( V_1^n - V_{P^{(*,T+1)},r}^{\pi^n} \right) + 2H\xi_{\text{down}} \right\}$$

$$\overset{(ii)}{\leq} \sum_{n=1}^{N_{\text{on}}} \left\{ \sum_{h=1}^{H} \left[ \xi_h^n + 2\beta_n \left\| \widehat{\phi}_h(s_h^n, a_h^n) \right\|_{(\Lambda_h^n)^{-1}} + 2\xi_{\text{down}} \right] + 2H\xi_{\text{down}} \right\}$$

$$\leq \underbrace{\sum_{n=1}^{N_{\text{on}}} \sum_{h=1}^{H} \xi_h^n}_{(I)} + \underbrace{2 \sum_{n=1}^{N_{\text{on}}} \sum_{h=1}^{H} \beta_n \left\| \widehat{\phi}_h(s_h^n, a_h^n) \right\|_{(\Lambda_h^n)^{-1}}}_{(II)} + 4HN_{\text{on}}\xi_{\text{down}}, \qquad (60)$$

where $(i)$ follows from Lemma 13 and $(ii)$ follows from Lemma 14.

For term $(I)$, note that $\{\xi_h^n\}_{n=1,h=1}^{N_{\mathrm{on}},H}$ is a martingale difference with $|\xi_h^n| \leq 2$. By Azuma–Hoeffding inequality, with proability at least $1 - \delta/4$, we have

$$\left| \sum_{n=1}^{N_{\mathrm{on}}} \sum_{h=1}^{H} \xi_h^n \right| \leq \sqrt{8 N_{\mathrm{on}} H \log(8/\delta)}. \tag{61}$$

For term $(II)$, we derive

$$
\begin{aligned}
(II) &= 2 \sum_{h=1}^{H} \sum_{n=1}^{N_{\mathrm{on}}} \beta_n \left\| \widehat{\phi}_h(s_h^n, a_h^n) \right\|_{(\Lambda_h^n)^{-1}} \\
&\overset{(i)}{\leq} 2 \sum_{h=1}^{H} \sqrt{\sum_{n=1}^{N_{\mathrm{on}}} \beta_n^2} \sqrt{\sum_{n=1}^{N_{\mathrm{on}}} \left\| \widehat{\phi}_h(s_h^n, a_h^n) \right\|_{(\Lambda_h^n)^{-1}}^2} \\
&\overset{(ii)}{\leq} 2 \sum_{h=1}^{H} \sqrt{2 c_\beta^2 \left( d^2 \iota_n N_{\mathrm{on}} + N_{\mathrm{on}}^2 d \xi_{\mathrm{down}}^2 + p N_{\mathrm{on}} \log N_{\mathrm{on}} \right)} \sqrt{2d \log \left( 1 + \frac{N_{\mathrm{on}}}{d\lambda} \right)} \\
&\leq 2H \sqrt{2 c_\beta^2 \left( d^2 \iota_n N_{\mathrm{on}} + N_{\mathrm{on}}^2 d \xi_{\mathrm{down}}^2 + p N_{\mathrm{on}} \log N_{\mathrm{on}} \right)} \sqrt{4d \log N_{\mathrm{on}}} \\
&\overset{(iii)}{\leq} 4\sqrt{2} c_\beta \left( \sqrt{H^2 d^3 \iota_n N_{\mathrm{on}} \log N_{\mathrm{on}}} + H d N_{\mathrm{on}} \xi_{\mathrm{down}} \sqrt{\log N_{\mathrm{on}}} + H \sqrt{dp N_{\mathrm{on}}} \log N_{\mathrm{on}} \right),
\end{aligned}
\tag{62}
$$

where $(i)$ follows from Cauchy-Schwarz inequality, $(ii)$ follows from Lemma 17, and $(iii)$ follows because $\forall x, y \geq 0, \sqrt{x+y} \leq \sqrt{x} + \sqrt{y}$.

Combining Equation (60), Equation (61) and Equation (62), we obtain

$$
\begin{aligned}
\sum_{n=1}^{N_{\mathrm{on}}} &\left( V_{P^{(*,T+1)},r}^* - V_{P^{(*,T+1)},r}^{\pi^n} \right) \\
&\leq 8\sqrt{2} c_\beta \left( \sqrt{H^2 d^3 \iota_n N_{\mathrm{on}} \log N_{\mathrm{on}}} + H d N_{\mathrm{on}} \xi_{\mathrm{down}} \sqrt{\log N_{\mathrm{on}}} + H \sqrt{dp N_{\mathrm{on}}} \log N_{\mathrm{on}} \right) \\
&= \widetilde{O} \left( H d N_{\mathrm{on}} \xi_{\mathrm{down}} + H \sqrt{d^3 N_{\mathrm{on}}} + H \sqrt{dp N_{\mathrm{on}}} \right) \\
&= \widetilde{O} \left( H d N_{\mathrm{on}} \xi_{\mathrm{down}} + H \sqrt{d N_{\mathrm{on}}} \max\{d, \sqrt{p}\} \right).
\end{aligned}
$$

Dividing both sides by $N_{\mathrm{on}}$, we have

$$V_{P^{(*,T+1)},r}^* - V_{P^{(*,T+1)},r}^{\tilde{\pi}} \leq \widetilde{O} \left( H d \xi_{\mathrm{down}} + H d^{1/2} N_{\mathrm{on}}^{-1/2} \max\{d, \sqrt{p}\} \right). \tag{63}$$

Furthermore, if the linear combination misspecification error $\xi$ (Assumption 5) is $\tilde{O}(\sqrt{d}/\sqrt{N_{\mathrm{on}}})$, and the number of trajectories collected in upstream is as large as

$$\widetilde{O} \left( H^3 d K^2 T N_{\mathrm{on}} + \left( H^3 d^3 K + H^5 K^2 + H^5 d^2 K \right) T^2 N_{\mathrm{on}} + \frac{H^5 K^3 T N_{\mathrm{on}}}{d^2} \right),$$

then $\xi_{\mathrm{down}}$ reduces to $\tilde{O}(\sqrt{d}/\sqrt{N_{\mathrm{on}}})$ by definition and Theorem 1, and hence the second term in Equation (63) dominates. The suboptimality gap thus becomes

$$\widetilde{O} \left( H d^{1/2} N_{\mathrm{on}}^{-1/2} \max\{d, \sqrt{p}\} \right).$$

$\square$

# E  Discussion of assumptions

In this section, We discuss all of our assumptions in Section 5 in greater detail. Below we elaborate intuitively why these assumptions are useful. Formal argument of their necessity or their possible relaxation can be an interesting future topic.

**Assumption 6** falls into the data-coverage type of assumptions typically adopted in the study of offline RL, such as in linear MDPs (Xie et al., 2021; Yin et al., 2022; Wang et al., 2021) and the OPE problem (Min et al., 2021). Such an assumption has been shown to be necessary to guarantee sample efficient offline RL for tabular (Yin and Wang, 2021) and linear MDPs in Wang et al. (2021). Further, Uehara et al. (2022) relaxed this assumption to a weaker version of the same type, but correspondingly has a slightly weaker result in the suboptimality gap (with respect to the optimal policy only) than those in Xie et al. (2021); Yin et al. (2022); Wang et al. (2021) (suboptimality gap with respect to any policy). One interesting topic for future study is to relax our Assumption 6 by this weaker assumption in Uehara et al. (2022) for our downstream learning.

**Assumptions 2 to 5** are useful for establishing the connection between upstream and downstream learning and then transferring the pre-trained representation from upstream to downstream.

**Assumption 2** requires the upstream exploration to be sufficient over all states so that the pre-trained representation in upstream is accurate for those high-frequent states in downstream MDP, even if these states occur not often in upstream. Such a type of reachability assumption has also been used in previous RL studies such as in Agarwal et al. (2020); Modi et al. (2021).

**Assumption 3** can be simplified to only require that the state space $\mathcal{S}$ is compact. Then, combining with Assumption 2, it can be shown that there exists a uniform distribution on $\mathcal{S}$ with the density function $f(s) = \upsilon$, where $1/\upsilon$ is the measure of $\mathcal{S}$. Essentially, we expect that there exist a distribution (e.g., uniform distribution in the context) that has non-zero density on all states so that every state can be explored well in upstream. Assumption 3 holds obviously in Tabular MDP and can hold for many RL settings with continuous state space as long as it is compact.

**Assumption 4** uses the average total variation (TV) distance to provide a bound for point-wise TV distance, i.e., the TV distance of each state-action pair. Without Assumption 4, it can occur for source tasks that $\mathbb{E}_{(s,a)\sim\mathcal{U}(s,a)}[\|P^* - \langle\cdot,\cdot\rangle\|] \leq \xi_{\text{down}}$ (see Equation (10) in Lemma 1), which is insufficient for pre-trained representation to perform well in downstream target task due to the difference of their transition kernels. Consequently, the straightforward exploration won't benefit from multitask learning. We remark that there might be other alternative assumptions that can help to achieve the same goal as stated above.

**Assumption 5** connects transition kernels between upstream source tasks and the downstream target task. Such a type of assumption is somewhat necessary to guarantee the performance transfer, but the exact form of the assumption may be relaxed.

# F   Discussion of connections to successor features (SF)

In this section, we discuss the related work on successor features (SF) and propose some interesting topics which can be further studied.

In the framework of Successor Features (SFs)  (Barreto et al., 2017), **rewards** are decomposed into feature representation (same for all tasks) and linear weights (different across tasks). Barreto et al. (2017) assumes that the transition kernel for all tasks are the same. In such a case, it can be easily shown that the $Q$ functions also admit a decomposition into common SFs for all tasks and varying weights $w$ across tasks.

Hence, the structure of Q-function naturally leads to a value-based approach to update Q-functions via Bellman equations. On the other hand, our problem formulation (along the line of low-rank MDPs) assumes that the **transition kernels** have decomposed structure of common feature for all tasks and different linear weights across all tasks, and the *reward functions may not have any structure*. Such a formulation naturally leads to a model-based approach, where the policy update is via a policy maximization oracle.

To connect the two formulations, one can consider the following setup, where both rewards (and hence Q-function) and transition kernels have decomposed structures. For such a setting, a combined value-based and model-based approach can be a good design option. One possible way is to learn the transition kernels first based on its structure, then update Q-function based on its structure, and finally update policy greedily based on Q-function. Such a framework has already been considered in Lehnert and Littman (2020), where the focus was on learning state representations, not on the

design of exploration. This is certainly a quite open topic that requires formal efforts to investigate in the future.

## G  Auxiliary Lemmas

In this section, we provide several lemmas that are commonly used for the analysis of MDP problems.

The following lemma (Dann et al., 2017) will be useful to measure the difference between two value functions under two MDPs and reward functions. We define $P_h V_{h+1}(s_h, a_h) = \mathbb{E}_{s \sim P_h(\cdot | s_h, a_h)} [V(s)]$ as a shorthand notation.

**Lemma 15.** (Simulation lemma). *Suppose $P_1$ and $P_2$ are two MDPs and $r_1$, $r_2$ are the corresponding reward functions. Given a policy $\pi$, we have,*

$$V_{h,P_1,r_1}^{\pi}(s_h) - V_{h,P_2,r_2}^{\pi}(s_h)$$

$$= \sum_{h'=h}^{H} \mathbb{E}_{\substack{s_{h'} \sim (P_2, \pi) \\ a_{h'} \sim \pi}} \left[ r_1(s_{h'}, a_{h'}) - r_2(s_{h'}, a_{h'}) + (P_{1,h'} - P_{2,h'}) V_{h'+1,P_1,r}^{\pi}(s_{h'}, a_{h'}) | s_h \right]$$

$$= \sum_{h'=h}^{H} \mathbb{E}_{\substack{s_{h'} \sim (P_1, \pi) \\ a_{h'} \sim \pi}} \left[ r_1(s_{h'}, a_{h'}) - r_2(s_{h'}, a_{h'}) + (P_{1,h'} - P_{2,h'}) V_{h'+1,P_2,r}^{\pi}(s_{h'}, a_{h'}) | s_h \right].$$

The following lemma is essential in bounding the suboptimality in downstream offline RL (see Lemma 3.1 in Jin et al. (2021)).

**Lemma 16.** *Let $\{\widehat{\pi}_h\}_{h=1}^{H}$ be the policy such that $\widehat{V}_h(s) = \langle \widehat{Q}_h(s, \cdot), \widehat{\pi}_h(\cdot | s) \rangle_{\mathcal{A}}$ and $\zeta_h(s, a) = (\mathbb{B}_h \widehat{V}_{h+1}(s, a)) - \widehat{Q}_h(s, a)$. Then for any $\widehat{\pi}$ and $s \in \mathcal{S}$, we have*

$$V_1^{\pi}(s) - V_1^{\widehat{\pi}}(s) = \sum_{h=1}^{H} \mathbb{E}_{\pi}[\zeta_h(s_h, a_a) | s_1 = s] - \sum_{h=1}^{H} \mathbb{E}_{\widehat{\pi}}[\zeta_h(s_h, a_h) | s_1 = s]$$

$$+ \sum_{h=1}^{H} \mathbb{E}_{\pi}[\langle \widehat{Q}_h(s_h, \cdot), \pi_h(\cdot | s_h) - \widehat{\pi}(\cdot | s_h) \rangle | s_1 = s],$$

*where the expectation is taken over $s_h, a_h$.*

The following lemma is a standard inequality in the regret analysis for linear MDPs in reinforcement learning (see Lemma G.2 in Agarwal et al. (2020) and Lemma 10 in Uehara et al. (2022)).

**Lemma 17.** (Elliptical potential lemma). *Consider a sequence of $d \times d$ positive semidefinite matrices $X_1, \ldots, X_N$ with $\text{tr}(X_n) \leq 1$ for all $n \in [N]$. Define $M_0 = \lambda_0 I$ and $M_n = M_{n-1} + X_n$. Then*

$$\sum_{n=1}^{N} \text{tr}\left(X_n M_{n-1}^{-1}\right) \leq 2 \log \det(M_N) - 2 \log \det(M_0) \leq 2d \log \left(1 + \frac{N}{d\lambda_0}\right).$$

Next, we introduce some useful inequalities that help convert the finite sample error bound into the sample complexity.

**Lemma 18** ($\varepsilon$-Covering Number). *Let $\mathcal{V}$ denote a class of function mapping from $\mathcal{S}$ to $\mathbb{R}$ with the following parametric form*

$$V(\cdot) = \min \left\{ \max_{a \in \mathcal{A}} r(\cdot, a) + w^{\top} \phi(\cdot, a) + \alpha \sqrt{\phi(\cdot, a)^{\top} \Lambda^{-1} \phi(\cdot, a)}, 1 \right\},$$

*where the parameters $(r, w, \beta, \Lambda)$ satisfy $r \in \mathcal{R}$, $\|w\| \leq L$, $\alpha \in [0, B]$ and $\Sigma \succeq \lambda I$. Assume $\|\phi(s, a)\| \leq 1$ for all $(s, a)$ pairs, and let $\mathcal{N}(\varepsilon; r, R, B, \lambda)$ be the $\varepsilon$-covering number of $\mathcal{V}$ with respect to the distance $\text{dist}(V, V') = \sup_s |V(s) - V'(s)|$. Further let $\mathcal{N}_{\mathcal{R}}(\varepsilon)$ be the $\epsilon$-covering number of function class $\mathcal{R}$. Then*

$$\log |\mathcal{N}(\epsilon; R, B, \lambda)| \leq d \log(1 + 6R/\varepsilon) + d^2 \log(1 + 18d^{1/2} B^2 / (\varepsilon^2 \lambda)) + \log \mathcal{N}_{\mathcal{R}}\left(\frac{\varepsilon}{3}\right)$$

*where $\mathcal{N}_{\mathcal{R}}(\frac{\epsilon}{3})$ is the $\frac{\epsilon}{3}$ covering number with respect to the reward function class $\mathcal{R}$.*

*Proof.* The proof is essentially the same as that in Jin et al. (2020) except that the function $r(s, a)$ is not necessarily linear with respect to the representation $\phi(s, a)$, and the function $\phi$ is selected from a function class $\Phi$.

Reparametrize the function class $\mathcal{V}$ by letting $A = \alpha^2 \Lambda^{-1}$, and we have

$$V(\cdot) = \min \left\{ \max_{a \in \mathcal{A}} r(\cdot, a) + w^\top \phi(\cdot, a) + \sqrt{\phi(\cdot, a)^\top A \phi(\cdot, a)}, 1 \right\},$$

where $r \in \mathcal{R}$, $\|w\| \leq R$, and $\|A\| \leq B^2 \lambda^{-1}$. For any two functions $V_1, V_2 \in \mathcal{V}$, let them take the above form with parameters $(r_1, w_1, A_1)$ and $(r_2, w_2, A_2)$, respectively. Since both $\min\{\cdot, 1\}$ and $\max_a$ are contractions, we have

$$\text{dist}(V_1, V_2)$$

$$\leq \sup_{s,a} \left| \left[ r_1(s, a) + w_1^\top \phi(s, a) + \sqrt{\phi(s, a)^\top A_1 \phi(s, a)} \right] \right.$$

$$\left. - \left[ r_2(s, a) + w_2^\top \phi(s, a) + \sqrt{\phi(s, a)^\top A_2 \phi(s, a)} \right] \right|$$

$$\leq \sup_{s,a} |r_1(s, a) - r_2(s, a)| + \sup_{\phi : \|\phi\| \leq 1} \left| \left[ w_1^\top \phi + \sqrt{\phi^\top A_1 \phi} \right] - \left[ w_2^\top \phi + \sqrt{\phi^\top A_2 \phi} \right] \right|$$

$$\leq \sup_{s,a} |r_1(s, a) - r_2(s, a)| + \sup_{\phi : \|\phi\| \leq 1} |(w_1 - w_2)^\top \phi| + \sup_{\phi : \|\phi\| \leq 1} \sqrt{|\phi^\top (A_1 - A_2) \phi|}$$

$$= \sup_{s,a} |r_1(s, a) - r_2(s, a)| + \|w_1 - w_2\| + \sqrt{\|A_1 - A_2\|}$$

$$\leq \sup_{s,a} |r_1(s, a) - r_2(s, a)| + \|w_1 - w_2\| + \sqrt{\|A_1 - A_2\|_\text{F}}, \tag{64}$$

where the second to last inequality follows from the fact that $|\sqrt{x} - \sqrt{y}| \leq \sqrt{|x - y|}$ for any $x, y \geq 0$. For matrices, $\|\cdot\|$ and $\|\cdot\|_\text{F}$ denote the matrix operator norm and Frobenius norm, respectively.

Let $\mathcal{C}_\mathcal{R}$ be an $\frac{\varepsilon}{3}$-cover of $\mathcal{R}$ such that $|\mathcal{C}_\mathcal{R}| = \mathcal{N}_\mathcal{R}(\frac{\varepsilon}{3})$. Let $\mathcal{C}_w$ be an $\frac{\varepsilon}{3}$-cover of $\{w \in \mathbb{R}^d | \|w\| \leq R\}$ with respect to the $l_2$-norm of a vector, and let $\mathcal{C}_A$ be an $\frac{\varepsilon^2}{9}$-cover of $\{A \in \mathbb{R}^{d \times d} | \|A\|_\text{F} \leq d^{1/2} B^2 \lambda^{-1}\}$ with respect to the Frobenius norm. By Lemma D.5 in Jin et al. (2020), it holds that

$$|\mathcal{C}_w| \leq (1 + 6R/\varepsilon)^d, \qquad |\mathcal{C}_A| \leq [1 + 18d^{1/2} B^2/(\lambda \varepsilon^2)]^{d^2}.$$

By Equation (64), for any $V_1 \in V$, there exists $r_2 \in \mathcal{C}_\mathcal{R}$, $w_2 \in \mathcal{C}_w$ and $A$ such that $V_2$ parametrized by $(r_2, w_2, A_2)$ satisfies $\text{dist}(V_1, V_2) \leq \varepsilon$. Hence, it holds that $\mathcal{N}(\epsilon; R, B, \lambda) \leq |\mathcal{N}_\mathcal{R}(\frac{\varepsilon}{3})| \cdot |\mathcal{C}_w| \cdot |\mathcal{C}_A|$, which yields the desired result. $\square$