# OpenReview forum: "Provable Benefit of Multitask Representation Learning in Reinforcement Learning"
_NeurIPS.cc/2022/Conference — NeurIPS 2022 Accept_

### Official Review · Reviewer_1pug · 2022-07-11

**Rating:** 7
**Confidence:** 3
**Soundness:** 4 excellent
**Presentation:** 4 excellent
**Contribution:** 3 good

**Summary:**

This paper theoretically characterize the benefit of representation learning under the low-rank Markov decision process (MDP) model. The framework under consideration is a reward-free upstream and reward-known downstream task, where all tasks share the same feature function $\phi∗$ but might differ in $\mu$. The downstream offline RL, consists of a new task sharing the same representation as the
upstream tasks given an offline dataset, and the goal is to find a near-optimal policy.
The authors propose a new theoretical algorithm `REFUEL` (REward Free mUltitask rEpresentation Learning) and theoretically show that a shared transferrable representation of low rank MDPs can be learned efficiently. They provide various theoretical results on representation uncertainty measure, sample complexity, and  upper bound on both the sample complexity and the suboptimality gap of downstream offline RL with the help of the learned representation.

**Questions:**


- In terms of the polynomial efficiency of REFUEL, in Proposition 3, it is shown that the summation of the PCVs over all iterations is sublinear with respect to the total number $N_u$ of iterations, assuming
exploration policy is derived by an oracle. How does this results change in practice when this assumption of existance of an oracle is violated?

- Is  there any theoretical connection between this framework to frameworks such as Successor Features (SFs) where the the representation is decomposed to the dynamics $\phi$ and $w$ representing the task-specific features? In general I have not seen such connection to be made in the majority of related work in  low-rank MDP and Block MDP literature. Is there a difference between this setting of low-rank MDP which is stricly more general than linear MDPs which assume representation is known a priori ?

**Limitations:**

- For the analysis to hold, there is dependency on the assumptions that connect the upstream and downstream MDPs. In practice this can be very restrictive in the space of designing multiple tasks.
- The analysis relies on the existance of an oracle for learning the joint distribution via MLE for the representation and during the policy exploration.

**Strengths And Weaknesses:**

The authors proceed with the algorithm design as follows:

### Joint MLE-based Modeling Learning:
The agent passes all previously collected data to estimate the low-rank components $\phi^n, \mu^{n,1} ... \mu^{n,t}$ simultaneously via the MLE oracle on the joint distribution of these random variables. In practice In practice, the representations $\Phi$, $\Psi$ are properly parameterized and learned by neural networks. Next, for each task t, the agent uses the learned embeddings to update the estimated transition kernel $\hat{P}^{(n,t)}$.

### Pseudo Cumulative Value Function Construction for Exploration
The agent first uses the representation estimator $\hat{\phi}^n$ to update the empirical covariance matrix $\hat{U}^{(n,t)}$. The Pseudo Cumulative Value Function (PCV) is then established as an uncertainty measure that captures the difference between the estimated and ground truth model.

In downstream RL, the agent is given a new target task $T + 1$ under MDP $M^{T+1}$ and aims to find a near-optimal policy for $i$. . The downstream task is assumed to share the same representation as
the upstream tasks, and hence the agent can adopt the representation learned from the upstream to
expedite its learning.

Finally, the joint learning and decision making among all tasks involved in the four main components differ from single-task RL algorithms, and enable it to exploit the shared representation across the source tasks to improve the sample efficiency.

To ensure that the feature learned in upstream can still work well in downstream tasks, upstream and
downstream MDPs should have certain connections. `Assumption 2` (Reachability) provides the assumptions made  on transition kernels to build such connections.

## Strengths
- Good coverage of theoretical results and the overall problem of learning joint representation in the context of multi-task RL is important for the community.
- Theoretical results on sample complexity : sample complexity upper bound for REFUEL and then compare multitask result with that of single-task RL.
- The theoretical results throughout the paper seems sound.

## Weaknesses
- This paper only provides theoretical results without empirical results
- There might be missing connection to related work to other MDP frameworks learning decomposed representations
- The assumptions and upper bounds do not hold in the case where oracle representation is not available and the case where these need to be approximated using function approximation

---

> ### Author Response · Authors · 2022-08-02
> **Response to Reviewer 1pug - Part 1**
>
> Many thanks for providing the helpful review. In the revised version, we have made the changes based on the reviewer’s comments.
>
> **Q1:** Is there any theoretical connection between this framework to frameworks such as Successor Features (SFs) where the the representation is decomposed to the dynamics $\phi$ and $w$ representing the task-specific features? In general I have not seen such connection to be made in the majority of related work in low-rank MDP and Block MDP literature?
>
> **A1:** Many thanks for the great question. In the framework of Successor Features (SFs) (see [1] below), **rewards** are decomposed into feature representation (same for all tasks) and linear weights (different across tasks). [1] assumes that the transition kernel for all tasks are the same. In such a case, it can be easily shown as in [1] that the $Q$ functions also admit a decomposition into common SFs for all tasks and varying weights $w$ across tasks.
> Hence, the structure of Q-function naturally leads to a value-based approach to update Q-functions via Bellman equations. On the other hand, our problem formulation (along the line of low-rank MDPs) assumes that the **transition kernels** have decomposed structure of common feature for all tasks and different linear weights across all tasks, and the *reward functions may not have any structure*. Such a formulation naturally leads to a model-based approach, where the policy update is via a policy maximization oracle.
>
> To connect the two formulations, one can consider the following setup, where both rewards (and hence Q-function) and transition kernels have decomposed structures. For such a setting, a combined value-based and model-based approach can be a good design option. One possible way is to learn the transition kernels first based on its structure, then update Q-function based on its structure, and finally update policy greedily based on Q-function. Such a framework has already been considered in [2], where the focus was on learning state representations, not on the design of exploration. This is certainly a quite open topic that requires formal efforts to investigate in the future.
>
> We have added the above discussions in the revision (in Appendix G in Supplementary).
>
> **Q2:** The assumptions and upper bounds do not hold in the case where oracle representation is not available and the case where these need to be approximated using function approximation. In terms of the polynomial efficiency of REFUEL, in Proposition 3, it is shown that the summation of the PCVs over all iterations is sublinear with respect to the total number $N_u$ of iterations, assuming exploration policy is derived by an oracle. How does this results change in practice when this assumption of existence of an oracle is violated?
>
> **A2:** Many thanks for the great question. We believe this is an interesting question worth serious research efforts in the future. Below we briefly discuss some of our initial thoughts.
>
> For the MLE oracle, in practice, it can be implemented by properly parameterizing the model classes $\Phi, \Psi$ such as by neural networks. Suppose such function approximation will cause the estimation error as $\sum_t || \phi^{\star}(\cdot)\mu_t^{\star}(s,a)-\hat{\phi}(\cdot){\hat{\mu}}_t(s,a)|| _{\mathrm{TV}} = O( \epsilon_0)$ where $(\phi^{\star}, \mu_1^{\star}, \ldots, \mu_T^{\star})$ are the models returned by the oracle and $(\hat{\phi}, \hat{\mu}_1, \ldots, \hat{\mu}_T)$ are learned models in practice. Then the final summation of PCVs will have an additional term of order $\tilde{O}(\epsilon_0N_u)$ (where the dependence on $H,K,d,T$ is omitted). Although such an additional term is linear with respect to $N_u$, the final sample complexity bound in Theorem 1 is robust and still holds when $\epsilon_0$ is sufficiently small.
>
> Regarding the policy optimization oracle, in practice, it may be implemented via a policy-gradient type of approach, namely, treating PCV as the objective function, and parameterizing the policy for example by neural networks. For a sufficient number of policy-gradient updates, we anticipate that the suboptimality gap of PCV will diminish up to a polynomial function of the approximation error $\epsilon'_0$ (i.e., in the form of $\mathrm{poly}(\epsilon'_0)$), which can come from multiple error sources such as approximation error of the optimal policy, function approximation errors of certain value functions in policy gradient, etc. Then the final summation of PCVs will have an additional term of $\tilde{O}(\mathrm{poly}(\epsilon'_0)N_u)$ (where the dependence on $H,K,d,T$ is omitted). Although such an additional term is linear with $N_u$, the final sample complexity bound is robust and still holds when the error $\epsilon'_0$ is sufficiently small.

---

> > ### Author Response · Authors · 2022-08-02
> > **Response to Reviewer 1pug - Part 2**
> >
> > **Q3:** Is there a difference between this setting of low-rank MDP which is strictly more general than linear MDPs which assume representation is known a priori?
> >
> > **A3:** Yes, both low-rank and linear MDPs admit a low-rank decomposition, but low-rank MDPs assume an unknown representation a priori and linear MDPs assume known representation.
> >
> >
> > **Q4:** For the analysis to hold, there is dependency on the assumptions that connect the upstream and downstream MDPs. In practice this can be very restrictive in the space of designing multiple tasks.
> >
> > **A4:** We agree that to design multi-task RL and make it beneficial in practice requires a lot more efforts than in the theoretical setting. To transfer the pre-trained representation to downstream with good performance, our assumptions on the connection between source tasks and the target task mainly aim to guarantee that the pre-trained upstream representation will still be useful under the transition kernel shift, which further causes data distribution shift. In practice, many methods have been developed to address such an issue, such as in meta-RL, the upstream learns a good initial representation and then downstream fine tunes it for specific tasks. It is an interesting future topic to analyze the performance of such types of algorithms, which will broaden our analysis here.
> >
> > **References:**
> >
> > [1] Barreto, A., Dabney, W., Munos, R., Hunt, J. J., Schaul, T., van Hasselt, H. P., and Silver, D. (2017). Successor features for transfer in reinforcement learning. Proc. Advances in Neural Information Processing Systems (NeurIPS).
> >
> > [2] Lehnert, L. and Littman, M. L. (2020). Successor features combine elements of model-free and model-based reinforcement learning. J. Mach. Learn. Res., 21:196:1–196:53.

---

### Official Review · Reviewer_hbyz · 2022-07-12

**Rating:** 6
**Confidence:** 1
**Soundness:** 4 excellent
**Presentation:** 4 excellent
**Contribution:** 2 fair

**Summary:**

The paper proposes a representation learning algorithm for multi-task reinforcement learning under low-rank MDPs assumption (transition kernels have a low-rank decomposition in the form of $P^t(s’|s, a) = \langle \phi(s’), \mu^t(s, a)\rangle$ where the feature of the next state $s’$ is shared across tasks $t \in [T]$). The algorithm adopts the step-by-step exploration approach from FLAMBE (Agarwal et al., 2020), uses the reward bonus from REP-UCB (Uehara et al., 2021), and extend their single-task analyses to the multi-task setting. The authors showed that jointly learning the representation for multiple tasks is more sample efficient than learning each individual task independently, as long as the number of tasks is large enough. The authors also show that the sample complexity of learning a new task in an offline setting using the learned representation can be reduced under mild assumptions on the similarity of the new task to the pre-training tasks.

**Questions:**

- It would be also interesting to see if the pre-trained representation would also benefit online learning of the new task $T+1$ (the paper studied offline learning of new task). Are there some challenges that might hinder such analysis?
- It would be helpful to highlight more on how the proof techniques used in this work are different from the prior work at the end of Section 4.2. The authors listed the new technical pieces that were introduced but I did not have a good understanding on how much different they are from the prior methods.

**Limitations:**

The authors have adequately addressed the limitations of their work.

**Strengths And Weaknesses:**

Strengths:
- The proposed representation learning for multi-task reinforcement learning setting is novel and the proposed algorithm is shown to provide provable sample complexity benefits when the number of tasks is high (with the number only dependent on the number of actions and the low-rank dimension).
- Although I did not check the proof carefully, the theoretical results look reasonable.

Weaknesses:
- Algorithmic-wise the proposed algorithm has limited innovation — 1) for the online setting (Algorithm 1), most components are directly from two prior works (FLAMBE and REP-UCB) with relatively straightforward modifications to make them amenable to the multi-task setting; 2) for the offline setting (Algorithm 2), the algorithm is an one-step version of LSVI-UCB.
- Assumption 4 in the downstream offline RL seems quite strong. It states that the point-wise (for a certain pair of $s, a$) TV distance of two transition kernels is bounded by the average (over the entire support of $S \times A$) TV distance of these two kernels multiplied by a constant. If $C_R$ were to depend on $S$ and $A$, it could be as big as $S \times A$, in which case would not give sample efficiency benefits in the end over the baseline REP-LCB.

---

> ### Author Response · Authors · 2022-08-02
> **Response to Reviewer hbyz - Part 1**
>
> Many thanks for providing the helpful review. In the revised version, we have made the changes based on the reviewer’s comments.
>
> **Q1:** Algorithmic-wise the proposed algorithm has limited innovation for the online setting (Algorithm 1), most components are directly from two prior works (FLAMBE and REP-UCB) with relatively straightforward modifications to make them amenable to the multi-task setting; 2) for the offline setting (Algorithm 2), the algorithm is an one-step version of LSVI-UCB.
>
> **A1:** Thanks for the comments. We want to bring to the reviewer's attention about the new components in our algorithm design, which are novel compared with single-task algorithms FLAMBE and REP-UCB.
>
> (a) The design of the pseudo cumulative value function (PCV) for joint multi-task learning is not straightforward. The PCV takes the $l_1/l_2$ form as
>
> $PCV (\hat{P}^{t},\hat{b}_h^{t},\pi_t;T)=\sum_h\sqrt{\sum_t(E[\hat{b}_h^t|\hat{P}^t, \pi_t])^2}$,
>
> in order to introduce coupling among tasks.
> Such coupling in PCV further yields jointly optimal policies, following which the collected trajectories for each task $t$ are implicitly helpful to guide exploration of other tasks. As a comparison, a natural $l_1/l_1$ type of reward form $\sum_t V_{\hat{P}^{t},\hat{b}_{h}^t}^{\pi_t}=\sum_h \sum_t E [\hat{b}_h^t|\hat{P}^t, \pi_t]$ yields decoupling among tasks and independent exploration policies among tasks, and hence does not lead to any benefit for multitask learning.
>
> (b) Our design idea of exploration is different from FLAMBE ([2]) and REP-UCB ([3]). Although both our algorithm and FLAMBE take reward-free exploration, the design ideas are different. Our PCV provides a policy-wise uncertainty level measurement for model estimation (across all source tasks), and selects policies which explores state-action pair with **large uncertainty of the estimated model**. As a comparison, FLAMBE seeks a mixture exploratory policy $\rho$ to ensure **good coverage over all state space** when executed in the model and collects a number of episodes during each iteration. As a result, FLAMBE may collect many more data of well-estimated states repeatedly in each iteration and has worse sample complexity. REP-UCB takes **reward-driven** exploration, and constructs an optimistic estimation of the value function with given reward and additional bonus in order to find optimistic action selection to maximize value functions. As a result, its exploration is also different from our **reward-free** algorithm.
>
> For downstream learning, with known pre-trained representation, we naturally adopt the pessimistic type algorithm for offline RL under linear MDPs ([1]). Our main contribution for downstream learning is to leverage the benefit of the learned representation from upstream. One notable feature of our downstream learning is that we capture the misspecification error of the pre-trained representation in the pessimism term $\Gamma$ and parameter $\beta$, which is crucial and different from standard algorithms. As a result, the misspecification error from upstream learning results in a constant approximation error (first term in Eqn. (14)).
>
> **Q2:** Assumption 4 in the downstream offline RL seems quite strong. It states that the point-wise (for a certain pair of $(s,a)$ TV distance of two transition kernels is bounded by the average (over the entire support of $S\times A$) TV distance of these two kernels multiplied by a constant. If $C_R$ were to depend on $S$ and $A$, it could be as big as $S\times A$, in which case would not give sample efficiency benefits in the end over the baseline REP-LCB.
>
> **A2:** Many thanks for the great point. In general, we agree that $C_R$ may depend on $\mathcal{S}$ and $\mathcal{A}$, and may scale linearly with $SK$ where $K=|\mathcal{A}|$ (or with $dK$ in low-rank MDPs). However, the sample efficiency benefits in downstream can still be achieved as we explain below.
> Note that $C_R$ is contained only in $\xi_{down}=\xi+\frac{C_LC_RT\upsilon\epsilon_u}{\kappa_u}$ in the final sub-optimality gap $\widetilde{O}\left(Hd^{1/2}\xi_{down}+N_{down}^{-1/2}H\max(d,\sqrt{p})\right)$ (simplified from Eq. (14) of Theorem 2). As long as the approximation error $\epsilon_u$ of pre-trained representation from upstream is sufficiently small, the first term $Hd^{1/2}\xi_{down}$ diminishes and the second term $N_{down}^{-1/2}H\max(d,\sqrt{p})$ dominates. Thus the sub-optimality gap can still enjoy sample efficiency benefits. Certainly, there is a cost due to the scaling of $C_R$, which leads to at most an additional factor of $d^2K^2$ in the number of trajectories collected in the upstream in order to make the approximation error $\epsilon_u$ small. This is typically not a big issue for pre-training in practice.

---

> > ### Author Response · Authors · 2022-08-02
> > **Response to Reviewer hbyz - Part 2**
> >
> > **Q3:** It would be also interesting to see if the pre-trained representation would also benefit online learning of the new task (the paper studied offline learning of new task). Are there some challenges that might hinder such analysis?
> >
> > **A3:** Many thanks for the great question. The answer is affirmative. We have also analyzed the online downstream learning of the new task and have shown the benefits of the pre-trained representation. We have added this study in the revision (see Appendix E in the supplementary). Please feel free to check and comment.
> >
> > **Q4:** It would be helpful to highlight more on how the proof techniques used in this work are different from the prior work at the end of Section 4.2. The authors listed the new technical pieces that were introduced but I did not have a good understanding on how much different they are from the prior methods.
> >
> > **A4:** Thanks for the suggestion. We next explain the difference of our proof techniques from prior work in each of our proof steps summarized at the end of Section 4.2.
> >
> >
> >
> > (1) In Step 1:
> > We upper-bound the summation of model estimation errors over all tasks **under training data** by the joint MLE objective of joint probability distribution. One key step is to use the KL-divergence as an intermediate metric to establish the connection, which is inspired by [2]. Such a bound captures the multitask benefit of model learning in Joint MLE Learning Guarantee (Lemma 3 in Appendix A.5). As a comparison, a straightforward extension of the existing single task analysis would consider a joint probability for all source tasks and provide a guarantee for TV distance between estimated joint distribution and true joint distribution, which cannot be compared directly with the summation of model estimation errors over all tasks to quantify the benefits of multitask learning.
> >
> >
> >
> > (2) In Step 2, we further upper bound the summation of model estimation errors over all tasks **under any policy** as an uncertainty measurement. Our new technique lies in coupling these model estimation errors across tasks, in order to yield a tighter upper bound given by PCV. Then PCV can be regarded as an uncertainty measurement that captures the summation of model estimation errors over all tasks under any policy. Such a coupling technique across multiple tasks has not been used before, and renders a tighter bound and shows the benefit of multitask learning. There would not be any benefit if we treat each model estimation error individually.
> >
> >
> > (3) In Step 3,
> > we show that the summation of the PCVs over all iterations is sublinear with respect to the total number of iterations $N_u$ , as shown in Proposition 3, which further implies polynomial efficiency of REFUEL in learning the model. A new analysis here lies in developing a new bound controlling the difference between PCVs under different transition kernels, which contributes to the final summation bound. As a comparison, in single-task learning, we only need to control the difference between true value functions, where the standard simulation lemma can be applied directly.
> >
> >
> > **References:**
> >
> > [1] Jin, Y., Yang, Z., and Wang, Z. (2021). Is pessimism provably efficient for offline rl? Proc. International Conference on Machine Learning, ICML 2021.
> >
> > [2] Agarwal, A., Kakade, S., Krishnamurthy, A., and Sun, W. (2020). Flambe: Structural complexity and representation learning of low rank mdps. Proc. Advances in Neural Information Processing Systems (NeurIPS).
> >
> > [3] Uehara, M., Zhang, X., and Sun, W. (2022). Representation learning for online and offline RL in low-rank MDPs. In Proc. International Conference on Learning Representations (ICLR).

---

### Official Review · Reviewer_XpNw · 2022-07-16

**Rating:** 6
**Confidence:** 4
**Soundness:** 3 good
**Presentation:** 2 fair
**Contribution:** 3 good

**Summary:**

This paper studies the benefit of joint representation learning in the presence of multiple reinforcement learning tasks. The authors introduce an algorithm called REFUEL that in the presence of $T$ tasks with a shared representation it is able to learn an ensemble of almost optimal policies in a number of trajectory samples that would be necessary if training each of these $T$ tasks independently. The benefits are only seen when $T = \Omega(\frac{K}{d^5})$ where $K$ is the number of actions (although the authors do not explicitly mention this).

These results are in line with a flurry of results in the literature that show the benefit of learning multiple tasks simultaneously in the presence of joint structures. This fact is communicated in the submissions' related work section. The authors then ask the question of wether representation learning can be beneficial in the setting where the learner is to encounter a test task. To show results for this setting the authors introduce strong coverage and reachability assumptions, and also require to assume the point wise TV is upper bounded by a constant multiple of the expected TV of all states sampled under the uniform distribution as well as a `linear dependence' relationship between target and source tasks. Moreover, they also require the learned features during training to satisfy a minimum eigenvalue assumption (over an unspecified measure $\rho$). Combined these assumptions allow the authors to derive guarantees for downstream offline RL.




**Questions:**

I would like to hear some clarification from the authors about the weaknesses described above.

**Limitations:**

This work does not pose any negative societal impact.

**Strengths And Weaknesses:**

Strengths. The setting of multitask representation learning is very interesting and relevant to the community. In particular, studying the problem of downstream offline RL and representational transfer is very exciting. The first part of this work provides expected results for the multi task setting and a well executed algorithm and set of results. Although not super surprising it is a welcome addition to the literature.

Weaknesses: the paper has a couple of typos and undefined variables ($K$, $\rho$). To date I don't know what $\rho$ is in Assumption 6. Although the setup of section 5 is interesting, the results presented require a series of strong assumptions that make the reader believe these are there just to make the results work. It would be beneficial if some work had been put in place to stress test these assumptions (in the form of for example lower bound arguments) and make the reader know which of these are necessary or which ones are just a device to make the proofs go through without further effort.

---

> ### Author Response · Authors · 2022-08-02
> **Response to Reviewer XpNw - Part 1**
>
> Many thanks for providing the helpful review. In the revised version, we have made the changes based on the reviewer’s comments.
>
> **Q1:** The paper has a couple of typos and undefined variables $(K,\rho)$.
>
> **A1:** Thanks for pointing this out. $K$ is the cardinality of the action space (see Section 3.1 in the revision), and $\rho$ is the behavior policy of the offline dataset (see Section 5 in the revision). The changes are highlighted by the blue-colored texts in the revision.
>
> **Q2:** Although the setup of Section 5 is interesting, the results presented require a series of strong assumptions that make the reader believe these are there just to make the results work. It would be beneficial if some work had been put in place to stress test these assumptions (in the form of for example lower bound arguments) and make the reader know which of these are necessary or which ones are just a device to make the proofs go through without further effort.
>
> **A2:** Thanks for the valuable suggestion. We have explained all of our assumptions in Section 5 in greater detail and have also added discussions of the assumptions in the revision (in Appendix F in Supplementary).
>
> Assumption 6 falls into the data-coverage type of assumptions typically adopted in the study of offline RL, such as in linear MDPs [2,3,7] and the OPE problem [4]. Such an assumption has been shown to be necessary to guarantee sample efficient offline RL for tabular [8] and linear MDPs in [7]. Further, [1] relaxed this assumption to a weaker version of the same type, but correspondingly has a slightly weaker result in the suboptimality gap (with respect to the optimal policy only) than those in [2,3,7] (suboptimality gap with respect to any policy). One interesting topic for future study is to relax our Assumption 6 by this weaker assumption in [1] for our downstream learning.
>
> Assumptions 2-5 are useful for establishing the connection between upstream and downstream learning and then transferring the pre-trained representation from upstream to downstream. Below we elaborate intuitively why these assumptions are useful. Formal argument of their necessity or their possible relaxation can be an interesting future topic.
>
> Assumption 2 requires the upstream exploration to be sufficient over all states, so that the pre-trained representation in upstream can be accurate for those high-frequent states in downstream MDP, even if these states occur not often in upstream. Such a type of reachability assumption has also been used in previous RL studies such as in [5,6].
>
> Assumption 3 can be simplified to only require that the state space $\mathcal{S}$ is compact. Then, combining with Assumption 2, it can be shown that there exists a uniform distribution on $\mathcal{S}$ with the density function $f(s)=\upsilon$, where $1/\upsilon$ is the measure of $\mathcal{S}$. Essentially, we expect that there exists a distribution (e.g., uniform distribution in the context) that has non-zero density on all states so that every state can be explored well in upstream. Assumption 3 holds obviously in Tabular MDPs and can hold for many RL settings with continuous state space as long as it is compact.
>
> Assumption 4 uses the average total variation (TV) distance to provide a bound for point-wise TV distance, i.e., the TV distance of each state-action pair. Without Assumption 4, it can occur for source tasks that $E_{(s,a)\sim \mathcal{U}(s,a)}[||P^*-\langle \cdot,\cdot\rangle||]\leq \xi_{\mathrm{down}}$ (see Eqn. (12) in Lemma 1), which is insufficient for pre-trained representation to perform well in downstream target task due to the difference of their transition kernels. Consequently, the straightforward exploration in downstream won't benefit from multitask learning. We remark that there might be other alternative assumptions that can help to achieve the same goal as stated above.
>
> Assumption 5 connects transition kernels between upstream source tasks and the downstream target task. Such a type of assumption is somewhat necessary to guarantee the performance transfer, but the exact form of the assumption may be relaxed.

---

> > ### Author Response · Authors · 2022-08-02
> > **Response to Reviewer XpNw - Part 2 (References for Part 1)**
> >
> > References for Part 1.
> >
> > [1] Uehara, M., Zhang, X., and Sun, W. (2022). Representation learning for online and offline RL in low-rank MDPs. In Proc. International Conference on Learning Representations (ICLR).
> >
> > [2] Xie, T., Cheng, C., Jiang, N., Mineiro, P., and Agarwal, A. (2021). Bellman-consistent pessimism for offline reinforcement learning. In
> > Proc. Advances in Neural Information Processing Systems (NeurIPS).
> >
> > [3] Yin, M., Duan, Y., Wang, M., and Wang, Y.-X. (2022). Near-optimal Offline Reinforcement Learning with Linear Representation: Leveraging Variance Information with Pessimism. arXiv e-prints, page arXiv:2203.05804.
> >
> > [4] Min, Y., Wang, T., Zhou, D., and Gu, Q. (2021). Variance-aware off-policy evaluation with linear function approximation. In Proc. Advances in Neural Information Processing Systems (NeurIPS).
> >
> > [5] Agarwal, A., Kakade, S., Krishnamurthy, A., and Sun, W. (2020). Flambe: Structural complexity and representation learning of low rank mdps. In Proc. Advances in Neural Information Processing Systems (NeurIPS).
> >
> > [6] Modi, A., Chen, J., Krishnamurthy, A., Jiang, N., and Agarwal, A. (2021). Model-free Representation Learning and Exploration in
> > Low-rank MDPs. arXiv e-prints, page arXiv:2102.07035.
> >
> > [7] Wang, R., Foster, D. P., and Kakade, S. M. (2021). What are the statistical limits of offline RL with linear function approximation? In 9th International Conference on Learning Representations (ICLR).
> >
> > [8] Yin, M. and Wang, Y.-X. (2021). Towards instance-optimal offline reinforcement learning with pessimism. In Proc. Advances in Neural Information Processing Systems (NeurIPS).

---

> > > ### Comment · Reviewer_XpNw · 2022-08-09
> > > **Thanks for your response**
> > >
> > > I still share reviewer hbyz concern about this work's lack of algorithmic novelty. As it is pointed out, the results of this submission are a straightforward combination of results in FLAMBE and REP-UCB. Nonetheless, the setting is interesting and the execution is correct. I will keep my score as is.

---

> > > > ### Author Response · Authors · 2022-08-09
> > > > **Thank you for the feedback**
> > > >
> > > > We thank the reviewer for the further feedback and for the positive comments about our setting and execution. Regarding the novelty of the algorithm, we want to bring to the reviewer's attention of our response to Reviewer hbyz's first question (Q1), where we explained the new components in our algorithm design compared with single-task algorithms FLAMBE and REP-UCB. We hope that the reviewer will find those useful. Thanks again for your time and efforts during the review process!

---

### Meta-Review · Area_Chair_HXEP · 2022-08-29

**Recommendation:** Accept
**Confidence:** Less certain

**Metareview:**

The reviewers are largely in consensus that the questions posed in the paper are extremely relevant today and, while somewhat unsurprising, the paper provides solid value by establishing "provable benefit", including algorithms with novel components and numerous analysis tools that look attractive for re-use by future researchers in this emerging domain.

The AC is surprised that not even a toy set of experiments was deemed necessary to validate the predicted multi-task performance in practice. However, the reviewers have all believed in the value of the theory alone, and I appreciate the setup of a large enough multi-task set would require significant work, though I hope this is pursued soon and often in future work.

**Award:**

No

---

### Decision · Program_Chairs · 2022-09-14

Accept